# Uncertainty-based inference of a common cause for body ownership

**Marie Chancel[1]\*, H Henrik Ehrsson[1]†, Wei Ji Ma[2]†**

[1]Department of Neuroscience, Karolinska Institutet, Stockholm, Sweden; [2]Center for Neural Science and Department of Psychology, New York University, New York, United States

**Abstract** Many studies have investigated the contributions of vision, touch, and proprioception to body ownership, i.e., the multisensory perception of limbs and body parts as our own. However, the computational processes and principles that determine subjectively experienced body ownership remain unclear. To address this issue, we developed a detection-like psychophysics task based on the classic rubber hand illusion paradigm, where participants were asked to report whether the rubber hand felt like their own (the illusion) or not. We manipulated the asynchrony of visual and tactile stimuli delivered to the rubber hand and the hidden real hand under different levels of visual noise. We found that: (1) the probability of the emergence of the rubber hand illusion increased with visual noise and was well predicted by a causal inference model involving the observer computing the probability of the visual and tactile signals coming from a common source; (2) the causal inference model outperformed a non-Bayesian model involving the observer not taking into account sensory uncertainty; (3) by comparing body ownership and visuotactile synchrony detection, we found that the prior probability of inferring a common cause for the two types of multisensory percept was correlated but greater for ownership, which suggests that individual differences in rubber hand illusion can be explained at the computational level as differences in how priors are used in the multisensory integration process. These results imply that the same statistical principles determine the perception of the bodily self and the external world.

**\*For correspondence:**
marie.chancel@ki.se

†These authors contributed equally to this work

**Competing interest:** The authors declare that no competing interests exist.

## Editor's evaluation

In the rubber hand illusion, a rubber hand feels as if being part of one's body when stroked in synchrony with one's own occluded hand. By varying the temporal lags between the strokes to the rubber and to the real hand, and visual noise, the authors suggest that body ownership is governed by an active and probabilistic causal inference process that uses both prior knowledge and sensory uncertainty. The authors argue that probabilistic functions rather than fixed multisensory integration rules governs body ownership, thereby opening new venues for investigating its computational principles. The evidence is compelling, and the findings advanced our fundamental understanding of the computational principles of body ownership.

## Introduction

The body serves as an anchor point for experiencing the surrounding world. Humans and animals need to be able to perceive what constitutes their body at all times, i.e., which objects are part of their body and which are not, to effectively interact with objects and other individuals in the external environment and to protect their physical integrity through defensive action. This experience of the body as one's own, referred to as 'body ownership' (*Ehrsson, 2012*), is automatic and perceptual in nature and depends on integrating sensory signals from multiple sensory modalities, including vision,

touch, and proprioception. We thus experience our physical self as a blend of sensory impressions that are combined into a coherent unitary experience that is separable from the sensory impressions associated with external objects, events, and scenes in the environment. This perceptual distinction between the self and nonself is fundamental not only for perception and action but also for higher self-centered cognitive functions such as self-recognition, self-identity, autobiographical memory, and self-consciousness (*Banakou et al., 2013*; *Beaudoin et al., 2020*; *Bergouignan et al., 2014*; *Blanke et al., 2015*; *Maister and Tsakiris, 2014*; *Tacikowski et al., 2020*; *van der Hoort et al., 2017*). Body ownership is also an important topic in medicine and psychiatry, as disturbances in bodily self-perception are observed in various neurological (*Brugger and Lenggenhager, 2014*; *Jenkinson et al., 2018*) and psychiatric disorders (*Costantini et al., 2020*; *Keizer et al., 2014*; *Saetta et al., 2020*), and body ownership is a critical component of the embodiment of advanced prosthetic limbs (*Collins et al., 2017*; *Makin et al., 2017*; *Niedernhuber et al., 2018*; *Petrini et al., 2019*). Thus, understanding how body ownership is generated is an important goal in psychological and brain sciences.

The primary experimental paradigm for investigating the sense of body ownership has been the rubber hand illusion (*Botvinick and Cohen, 1998*). In the rubber hand illusion paradigm, participants watch a life-sized rubber hand being stroked in the same way and at the same time as strokes are delivered to their real passive hand, which is hidden from view behind a screen. After a period of repeated synchronized strokes, most participants start to feel the rubber hand as their own and sense the touches of the paintbrush on the rubber hand where they see the model hand being stroked. The illusion depends on the match between vision and somatosensation and is triggered when the observed strokes match the sensed strokes on the hidden real hand and when the two hands are placed sufficiently close and in similar positions. A large body of behavioral research has characterized the temporal (*Shimada et al., 2009*; *Shimada et al., 2014*), spatial (*Lloyd, 2007*; *Preston, 2013*), and other (e.g. form and texture; *Filippetti et al., 2019*; *Holmes et al., 2006*; *Lin and Jörg, 2016*; *Lira et al., 2017*; *Tieri et al., 2015*; *Ward et al., 2015*) rules that determine the elicitation of the rubber hand illusion and have found that these rules are reminiscent of the spatial and temporal congruence principles of multisensory integration (*Ehrsson, 2012*; *Kilteni et al., 2015*). Moreover, neuroimaging studies associate body ownership changes experienced under the rubber hand illusion with activations of multisensory brain regions (*Ehrsson et al., 2004*; *Guterstam et al., 2019a*; *Limanowski and Blankenburg, 2016*). However, we still know very little about the perceptual decision process that determines whether sensory signals should be combined into a coherent own-body representation or not, i.e., the multisensory binding problem that lays at the heart of body ownership and the distinction between the self and nonself.

The current study goes beyond the categorical comparisons of congruent and incongruent conditions that have dominated the body representation literature and introduces a quantitative model-based approach to investigate the computational principles that determine body ownership perception. Descriptive models (e.g. Gaussian fit) traditionally used in psychophysics experiments are useful to provide detailed statistical summaries of the data. These models describe 'what' perception emerges in response to stimulation without making assumptions about the underlying sensory processing. However, computational approaches using process models make quantitative assumptions on 'how' the final perception is generated from sensory stimulation. Among these types of models, Bayesian causal inference (BCI) models (*Körding et al., 2007*) have recently been used to explain the multisensory perception of external objects (*Cao et al., 2019*; *Kayser and Shams, 2015*; *Rohe et al., 2019*), including the integration of touch and vision (*Badde et al., 2020*). The interest in this type of model stems from the fact that it provides a formal solution to the problem of deciding which sensory signals should be bound together and which should be segregated in the process of experiencing coherent multisensory objects and events. In BCI models, the most likely causal structure of multiple sensory events is estimated based on spatiotemporal correspondence, sensory uncertainty, and prior perceptual experiences; this inferred causal structure then determines to what extent sensory signals should be integrated with respect to their relative reliability.

In recent years, it has been proposed that this probabilistic model could be extended to the sense of body ownership and the multisensory perception of one's own body (*Fang et al., 2019*; *Kilteni et al., 2015*; *Samad et al., 2015*). In the case of the rubber hand illusion, the causal inference principle predicts that the rubber hand should be perceived as part of the participant's own body if a common

cause is inferred for the visual, tactile, and proprioceptive signals, meaning that the real hand and rubber hand are perceived as the same. *Samad et al., 2015* developed a BCI model for the rubber hand illusion based on the spatiotemporal characteristics of visual and somatosensory stimulation but did not quantitatively test this model. These authors used congruent and incongruent conditions and compared questionnaire ratings and skin conductance responses obtained in a group of participants (group level) to the model simulations; however, they did not fit their model to individual responses, i.e., did not quantitatively test the model. *Fang et al., 2019* conducted quantitative model testing, but a limitation of their work is that they did not use body ownership perceptual data but an indirect behavioral proxy of the rubber hand illusion (reaching error) that could reflect processes other than body ownership (arm localization for motor control). More precisely, these authors developed a visuoproprioceptive rubber hand illusion based on the action of reaching for external visual targets. The error in the reaching task, induced by manipulating the spatial disparity between the image of the arm displayed on a screen and the subject's (a monkey or human) real unseen arm, was successfully described by a causal inference model. In this model, the spatial discrepancy between the seen and felt arms is taken into account to determine the causal structure of these sensory stimuli. The inferred causal structure determines to what extent vision and proprioception are integrated in the final percept of arm location; this arm location estimate influences the reaching movement by changing the planned action's starting point. Although such motor adjustments to perturbations in sensory feedback do not equate to the sense of body ownership, in the human participants, the model's outcome was significantly correlated with the participants' subjective ratings of the rubber hand illusion. While these findings are interesting (*Ehrsson and Chancel, 2019*), the evidence for a causal inference principle governing body ownership remains indirect, using the correlation between reaching performance and questionnaire ratings of the rubber hand illusion instead of a quantitative test of the model based on perceptual judgments of body ownership.

Thus, the present study's first goal was to test whether body ownership is determined by a Bayesian inference of a common cause. We developed a new psychophysics task based on the classical rubber hand illusion to allow for a trial-by-trial quantitative assessment of body ownership perception and then fitted a BCI model to the individual-level data. Participants performed a detection-like task focused on the ownership they felt over a rubber hand within a paradigm where the tactile stimulation they felt on their real hidden hand was synchronized with that of the rubber hand or systematically delayed or advanced in intervals of 0–500 ms. We calculated the percentage of trials in which participants felt the rubber hand as theirs for each degree of asynchrony. A Bayesian observer (or 'senser', as the rubber hand illusion creates a bodily illusion that one feels) would perceive the rubber hand as their own hand when the visual and somatosensory signals are inferred as coming from a common source, a single hand. In this BCI for body ownership model (which we refer to as the 'BCI model'), the causal structure is inferred by comparing the absolute value of the measured asynchrony between the participants' seen and felt touches to a criterion that depends on the prior probability of a common source for vision and somatosensation.

A second key aim was to test whether sensory uncertainty influences the inference of a common cause for the rubber hand illusion, which is a critical prediction of the BCI models not tested in earlier studies (*Fang et al., 2019*; *Samad et al., 2015*). Specifically, a Bayesian observer would take into account trial-to-trial fluctuations in sensory uncertainty when making perceptual decisions, changing their decision criterion in a specific way as a function of the sensory noise level of the current trial (*Keshvari et al., 2012*; *Körding et al., 2007*; *Magnotti et al., 2013*; *Qamar et al., 2013*; *Zhou et al., 2020*). Alternatively, the observer might incorrectly assume that sensory noise does not change or might ignore variations in sensory uncertainty. Such an observer would make a decision regarding whether the rubber hand is theirs or not based on a fixed criterion (FC) that does not depend on sensory uncertainty. Suboptimal but potentially 'easy-to-implement' observer models using a FC decision rule have often been used to challenge Bayesian models of perception (*Badde et al., 2020*; *Qamar et al., 2013*; *Rahnev et al., 2011*; *Stengård and van den Berg, 2019*; *Zhou et al., 2020*). To address whether humans optimally adjust the perceptual decision made to the level of sensory uncertainty when inferring a common cause for body ownership, we varied the level of sensory noise from trial to trial and determined how well was the data fit from our BCI model compared to a FC model.

Finally, we directly compared body ownership and a basic multisensory integration task within the same computational modeling framework. Multisensory synchrony judgment is a widely used

task to examine the integration versus segregation of signals from different sensory modalities (*Colonius and Diederich, 2020*), and such synchrony perception follows BCI principles (*Adam and Noppeney, 2014*; *Magnotti et al., 2013*; *Noel et al., 2018*; *Noppeney and Lee, 2018*; *Shams et al., 2005*). Thus, we reasoned that by comparing ownership and synchrony perceptions, we could directly test our assumption that both types of multisensory percepts follow similar probabilistic causal inference principles and identify differences that can advance our understanding of the relationships of the two (see further information below). To this end, we collected both visuo-tactile synchrony judgments and body ownership judgments of the same individuals under the same conditions; only instructions regarding which perceptual feature to detect – hand ownership or visuotactile synchrony – differed. Thus, we fit both datasets using our BCI model. We modeled shared sensory parameters and lapses for both tasks as we applied the same experimental stimulations to the same participants, and we compared having a shared prior for both tasks versus having separate priors for each task and expected the latter to improve the model fit (see below). Furthermore, we tested whether the estimates of prior probabilities for a common cause in the ownership and synchrony perceptions were correlated in line with earlier observations of correlations between descriptive measures of the rubber hand illusion and individual sensitivity to asynchrony (*Costantini et al., 2016*; *Shimada et al., 2014*). We also expected the prior probability of a common cause to be systematically higher for body ownership than for synchrony detection; this a priori greater tendency to integrate vision and touch for body ownership would explain how the rubber hand illusion could emerge despite the presence of noticeable visuotactile asynchrony (*Shimada et al., 2009*; *Shimada et al., 2014*). In the rubber hand illusion paradigm, the rubber hand's placement corresponds with an orientation and location highly probable for one's real hand, a position that we often adopt on a daily basis. Such previous experience likely facilitates the emergence of the rubber hand illusion we theorized (*Samad et al., 2015*) while not necessarily influencing visuotactile simultaneity judgments (*Smit et al., 2019*).

Our behavioral and modeling results support the predictions made for the three main aims described above. Thus, collectively, our findings establish the uncertainty-based inference of a common cause for multisensory integration as a computational principle for the sense of body ownership.

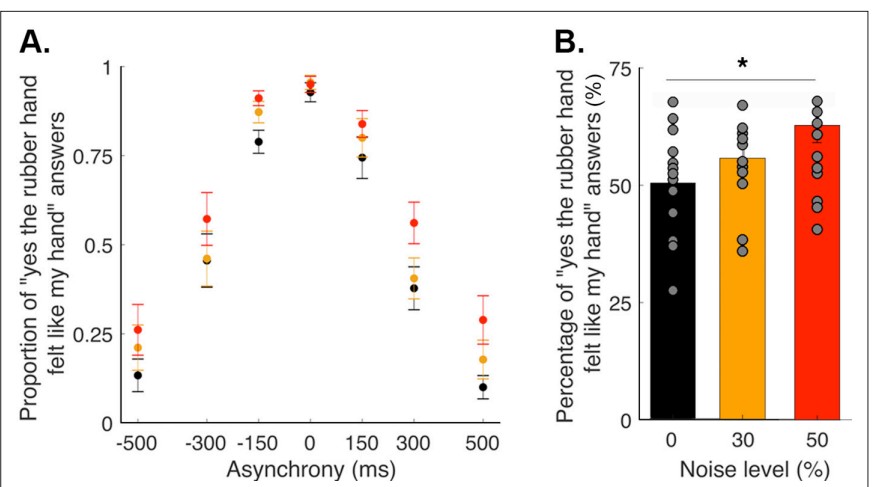

**Figure 1.** Elicited rubber hand illusion under different levels of visual noise. (**A**) Colored dots represent the mean reported proportion of elicited rubber hand illusions (± SEM) for each asynchrony for the 0 (black), 30 (orange), and 50% (red) noise conditions. (**B**) Bars represent how many times in the 84 trials the participants answered 'yes (the rubber hand felt like my own hand)' under the 0 (black), 30 (orange), and 50% (red) noise conditions; gray dots are individual data points. There was a significant increase in the number of 'yes' answers when the visual noise increased * p<0.001.

The online version of this article includes the following source data for figure 1:

**Source data 1.** Sum of "yes" answer for the different asynchrony and noise levels tested in the body ownership judgment task used in *Figure 1*.

# Results

## Behavioral results

In this study, participants performed a detection-like task on the ownership they felt toward a rubber hand; the tactile stimulation they felt on their hidden real hand (taps) was synchronized with the taps applied to the rubber hand that they saw or systematically delayed (negative asynchronies) or advanced (positive asynchronies) by 150, 300, or 500 ms. Participants were instructed to report if 'yes or no (the rubber hand felt like it was my hand)'. For each degree of asynchrony, the percentage of trials in which the participants felt like the rubber hand was theirs was determined (*Figure 1A*). Three different noise conditions were tested, corresponding to 0, 30, and 50% of visual noise being displayed via augmented reality glasses (see Materials and methods). The rubber hand illusion was successfully induced in the synchronous condition; indeed, the participants reported perceiving the rubber hand as their own hand in 94 ± 2% (mean ± SEM) of the 12 trials when the visual and tactile stimulations were synchronous; more precisely, 93 ± 3%, 96 ± 2%, and 95 ± 2% of responses were 'yes' responses for the conditions with 0, 30, and 50% visual noise, respectively. Moreover, for every participant, increasing the asynchrony between the seen and felt taps decreased the prevalence of the illusion. When the rubber hand was touched 500 ms before the real hand, the illusion was reported in only 20 ± 5% of the 12 trials (noise level 0: 13 ± 4%, noise level 30: 21 ± 5%, and noise level 50: 26 ± 7%); when the rubber hand was touched 500 ms after the real hand, the illusion was reported in only 19 ± 6% of the 12 trials (noise level 0: 10 ± 3%, noise level 30: 18 ± 5%, and noise level 50: 29 ± 6%; main effect of asynchrony: F[6, 84]=5.97, p<0.001; for the individuals' response plots, see *Figure 2—figure supplements 1–4*). Moreover, regardless of asynchrony, the participants perceived the illusion more often when the level of visual noise increased (*F*[2, 28]=22.35, p<0.001; Holmes' post hoc test: noise level 0 versus noise level 30: p=0.018, $d_{avg}$ = 0.4; noise level 30 versus noise level 50: p=0.005, $d_{avg}$ = 0.5; noise level 0 versus noise level 50: p<0.001, $d_{avg}$ = 1; *Figure 1B*). The next step was to examine whether these behavioral results can be accounted for by the BCI principles, including the increased emergence of the rubber hand illusion with visual noise.

## BCI model fit to body ownership

Our main causal inference model, the BCI model, assumes that the observer infers the causal structure of the visual and tactile signal to decide to what extent they should be merged into one coherent percept. In this model, the inference depends on the prior probability of the common cause and the trial-to-trial sensory uncertainty. Thus, this model has five free parameters: $p_{\mathrm{same}}$ is the prior probability of a common cause for vision and touch, independent of any sensory stimulation, $\sigma_0$, $\sigma_{30}$, $\sigma_{50}$ correspond to the noise impacting the measured visuotactile asynchrony in each of the three noise conditions, and $\lambda$ is the lapse rate to account for random guesses and unintended responses (see Materials and methods and Appendix 1 for more details). This BCI model fits the observed data well (*Figure 2A*). This finding supports our hypothesis that the sense of body ownership is based on an uncertainty-based inference of a common cause. Three further observations can be noted. First, the probability of a common cause for the visual and tactile stimuli $p_{\mathrm{same}}$ exceeded 0.5 (mean ± SEM: 0.80±0.05), meaning that in the context of body ownership, observers seemed to assume that vision and touch were more likely to come from one source than from different sources. This result broadly corroborates previous behavioral observations that the rubber hand illusion can emerge despite considerable sensory conflicts, for example, visuotactile asynchrony of up to 300 ms (*Shimada et al., 2009*). Second, the estimates for the sensory noise $\sigma$ increased with the level of visual white noise: 116±13 ms, 141±25 ms, and 178±33 ms for the 0, 30, and 50% visual noise conditions, respectively (mean ± SEM); this result echoes the increased sensory uncertainty induced by our experimental manipulation. Finally, the averaged lapse rate estimate $\lambda$ was rather low, 0.08±0.04, as expected for this sort of detection-like task, when participants were performing the task according to the instructions (see *Figure 2—figure supplement 1* for individual fit results).

## Comparing the BCI model to Bayesian and non-Bayesian alternative models

Next, we compared our BCI model to alternative models (see Materials and methods and Appendix 1). First, we observed that adding an additional parameter to account for observer-specific stimulation

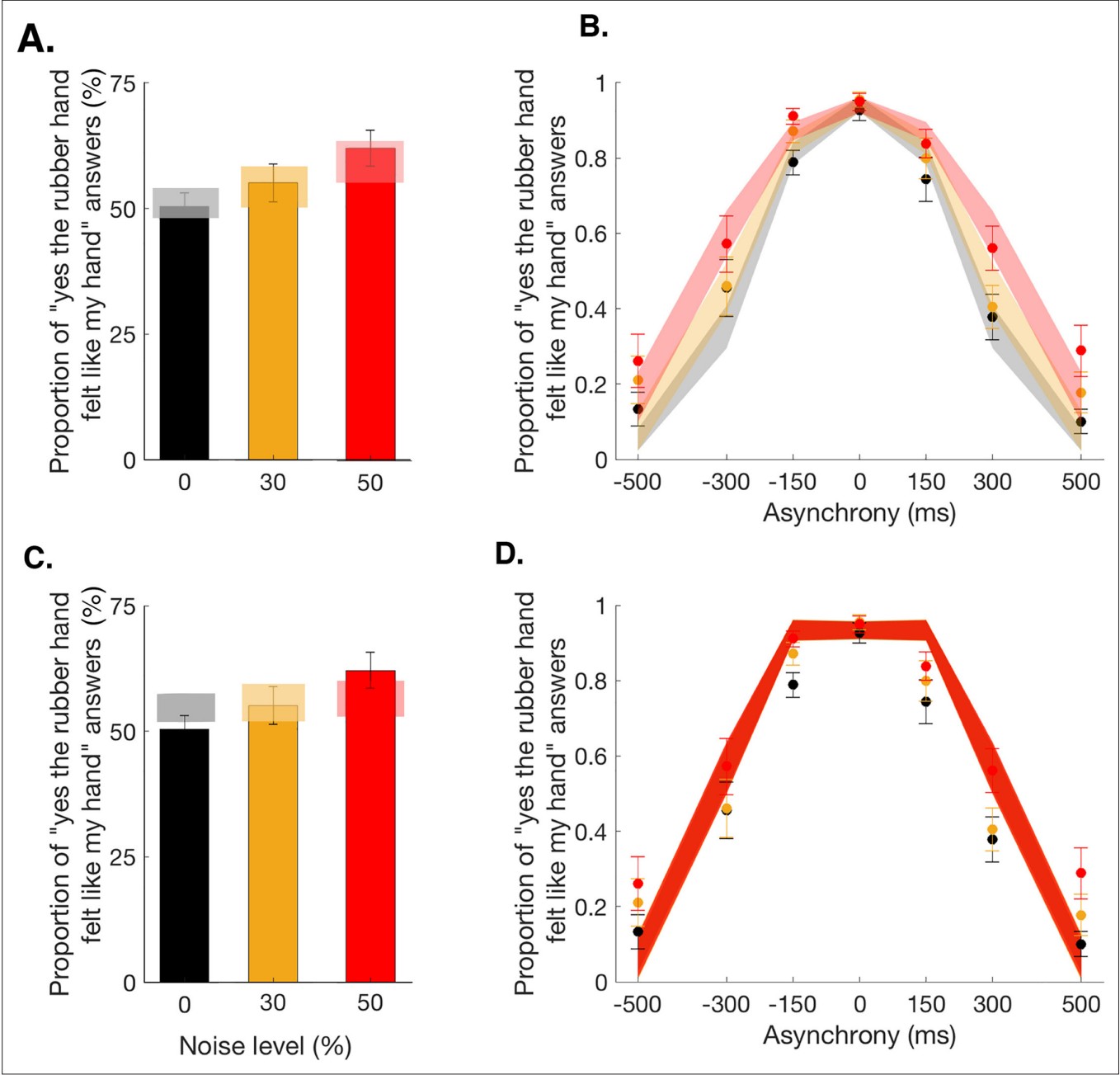

**Figure 2.** Observed and predicted detection responses for body ownership in the rubber hand illusion. Bars represent how many times across the 84 trials participants answered 'yes' in the 0 (black), 30 (orange), and 50% (red) noise conditions (mean ± SEM). Lighter polygons denote the Bayesian causal inference (BCI) model predictions (**A**) and fixed-criterion (FC) model predictions (**C**) for the different noise conditions. Observed data refer to 0 (black dots), 30 (orange dots), and 50% (red dots) visual noise and corresponding predictions (mean ± SEM; gray, yellow, and red shaded areas, respectively) for the BCI model (**B**) and FC model (**D**).

The online version of this article includes the following source data and figure supplement(s) for figure 2:

**Source data 1.** Parameter estimates for the BCI* model use in *Figure 2*.

**Figure supplement 1.** Individual data and BCI model fit.

**Figure supplement 2.** Individual data and FC model fit.

**Figure supplement 3.** Individual data and BCI* model fit.

**Figure supplement 4.** Individual data and BCIbias model fit.

**Figure supplement 5.** Predicted probabilty of emergence of the rubber hand illusion by the BCI model (upper table) and the FC model (lower table).

**Table 1.** Bootstrapped CIs (95% CI) of the Akaike information criterion (AIC) and Bayesian information criterion (BIC) differences between our main model Bayesian causal inference (BCI) and the BCI* (first line) and fixed criterion (FC; second line) models.

A negative value means that the BCI model is a better fit. Thus, the BCI model outperformed the other two.

| Model comparison | AIC (95% CI) | | | BIC (95% CI) | | |
|---|---|---|---|---|---|---|
| | Lower bound | Raw sum | Upper bound | Lower bound | Raw sum | Upper bound |
| BCI – BCI* | −28 | −25 | −21 | −81 | −77 | −74 |
| BCI – FC | −116 | −65 | −17 | −116 | −65 | −17 |

Finally, the pseudo-R2 were of the same magnitude for each model (mean ± SEM: BCI = 0.62 ± 0.04, BCI* = 0.62 ± 0.04, FC = 0.60 ± 0.05). However, the exceedance probability analysis confirmed the superiority of the Bayesian models over the fixed criterian one for the ownership data (family exceedance probability [EP]: Bayesian: 0.99, FC: 0.0006; when comparing our main model to the FC: protected-$EP_{FC}$ = 0.13, protected-$EP_{BCI}$ = 0.87, posterior probabilities: RFX: p[H1|y] = 0.740, null: p[H0|y] = 0.260).

uncertainty in the BCI* model did not improve the fit of the BCI model (*Table 1*, *Figure 2—figure supplement 3*). This observation suggests that assuming the observer's assumed stimulus distribution has the same SD as the true stimulus distribution was reasonable, i.e., allowing a participant-specific value for $\sigma_S$ did not improve the fit of our model enough to compensate for the loss of parsimony.

Second, an important alternative to the Bayesian model is a model that ignores variations in sensory uncertainty when judging if the rubber hand is one's own, for example, because the observer incorrectly assumes that sensory noise does not change. This second alternative model based on a fixed decisional criterion is the FC model. The goodness of fit of the BCI model was found to be higher than that of the FC model (*Figure 2*, *Table 1*, and *Figure 2—figure supplement 2*). This result shows that the BCI model provides a better explanation for the ownership data than the simpler FC model that does not take into account the sensory uncertainty in the decision process.

## Comparison of the body ownership and synchrony tasks

The final part of our study focused on the comparison of causal inferences of body ownership and visuotactile synchrony detection. In an additional task, participants were asked to decide whether the visual and tactile stimulation they received happened at the same time, i.e., whether the felt and seen touches were synchronous or not. The procedure was identical to the body ownership detection task apart from a critical difference in the instructions, which was now to detect if the visual and tactile stimulations were synchronous (instead of judging illusory rubber hand ownership).

## Extension analysis results (Table 2 and Figure 3 and Figure 3—figure supplement 1)

The BCI model fits the combined dataset from both ownership and synchrony tasks well (*Figure 3B and C* and *Figure 3—figure supplement 1*). Since the model used identical parameters (or identical parameters except for one), this observation supports the hypothesis that both the rubber hand illusion and visuotactile synchrony perception are determined by similar multisensory causal inference processes. However, in agreement with one of our other hypotheses, the goodness of fit of

**Table 2.** Bootstrapped CIs (95% CI) for the Akaike information criterion (AIC) and Bayesian information criterion (BIC) differences between shared and different $p_{same}$ values for the Bayesian causal inference (BCI) model in the extension analysis.

A negative value means that the model with different $p_{same}$ values is a better fit.

| Model comparison | AIC (95% CI) | | | BIC (95% CI) | | |
|---|---|---|---|---|---|---|
| | Lower bound | Raw sum | Upper bound | Lower bound | Raw sum | Upper bound |
| Different $p_{same}$ – shared parameters | −597 | −352 | −147 | −534 | −289 | −83 |

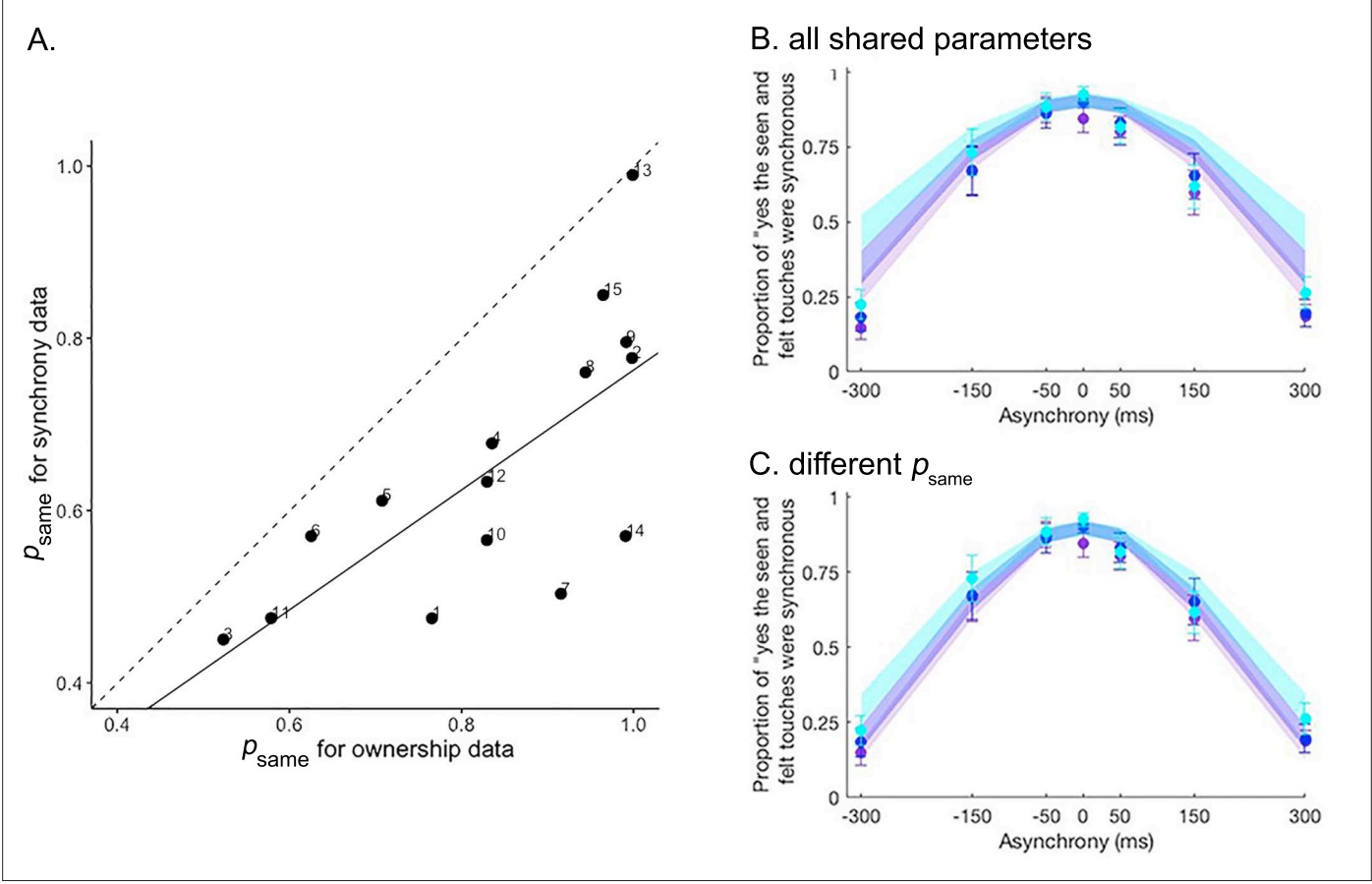

**Figure 3.** Extension analysis results. (**A**) Correlation between the prior probability of a common cause $p_{same}$ estimated for the ownership and synchrony tasks in the extension analysis. The $p_{same}$ estimate is significantly lower for the synchrony task than for the ownership task. The solid line represents the linear regression between the two estimates, and the dashed line represents the identity. Numbers denote the participants' numbers. (**B and C**) Colored dots represent the mean reported proportion of perceived synchrony for visual and tactile stimulation for each asynchrony under the 0 (purple), 30 (blue), and 50% (light blue) noise conditions (±SEM). Lighter shaded areas show the corresponding Bayesian causal inference (BCI) model predictions made when all parameters are shared between the ownership and synchrony data (**B**) and when $p_{same}$ is estimated separately for each dataset (**C**) for the different noise conditions (see also *Figure 3—figure supplement 1*).

The online version of this article includes the following source data and figure supplement(s) for figure 3:

**Source data 1.** Parameter estimates for the extension and transfer analysis and collected answers in the synchrony detection tasks used in *Figure 3*.

**Figure supplement 1.** Mean + SEM behavioural (dots) and model (shaded areas) results for body ownership (**A & C**) and synchrony detection (**B & D**) tasks in the extension analysis.

**Figure supplement 2.** Mean + SEM behavioural (dots) and model (shaded areas) results for body ownership (**A & C**) and synchrony detection (**B & D**) tasks in the transfer analysis.

**Figure supplement 3.** Perceived synchrony under different levels of visual noise.

the model improved greatly when the probability of a common cause ($p_{same}$) differed between the two tasks (*Table 2*). Importantly, $p_{same}$ was significantly lower for the synchrony judgment task (mean ± SEM: 0.65±0.04) than for the ownership judgment task (mean ± SEM: 0.83±0.04, paired t-test: $t=5.9141$, df = 14, and p<0.001). This relatively stronger a priori probability for a common cause for body ownership compared to visuotactile synchrony judgments supports the notion that body ownership and visuotactile event synchrony correspond to distinct multisensory perceptions, albeit being determined by similar causal probabilistic causal inference principles. Finally, in line with our hypothesis, we found that the $p_{same}$ values estimated separately for the two tasks were correlated (Pearson correlation: p=0.002, cor = 0.71; *Figure 3A*). That is, individuals who displayed a higher prior probability of combining the basic tactile and visual signals and perceiving the visuotactile synchrony of the events also showed a greater likelihood of combining multisensory signals in the ownership task

**Table 3.** Bootstrapped CIs (95% CIs) of the Akaike information criterion (AIC) and Bayesian information criterion (BIC) differences between the partial and full transfer analyses for the Bayesian causal inference (BCI) model.

'O to S' corresponds to the fitting of synchrony data by the BCI model estimates from ownership data. 'S to O' corresponds to the fitting of ownership data by the BCI model estimates from synchrony data. A negative value means that the partial transfer model is a better fit.

| | AIC (partial – full transfer, 95% CI) | | | BIC (partial – full transfer, 95% CI) | | |
|---|---|---|---|---|---|---|
| Transfer direction | Lower bound | Raw sum | Upper bound | Lower bound | Raw sum | Upper bound |
| O to S | –1837 | –1051 | –441 | –1784 | –998 | –388 |
| S to O | –1903 | –1110 | –448 | –1851 | –1057 | –394 |

and experiencing the rubber hand illusion. This observation corroborates the link between visuotactile synchrony detection and body ownership perception and provides a new computational understanding of how individual differences in multisensory integration can explain individual differences in the rubber hand illusion.

## Transfer analysis results (Table 3, Figure 3—figure supplement 2)

Finally, we compared the body ownership and synchrony tasks using what we call a transfer analysis. We used the parameters estimated for the ownership task to fit the synchrony task data (O to S) or the parameters estimated for the synchrony task to fit the ownership task data (S to O). Leaving $p_{same}$ as a free parameter always led to a much better fit of the data, as displayed in *Table 3* (see also *Figure 3—figure supplement 2*). Thus, this analysis leads us to the same conclusion as that of the extension analysis. The body ownership task and synchrony task involved different processing of the visual and somatosensory signals for the participants, and this difference in behavioral responses was well captured when two different a priori probabilities for a common cause were used to model each task.

Note that the exceedance probability analysis also confirmed the superiority of the Bayesian models over the FC one for the synchrony data when analyzed separately from the ownership data (family exceedance probability: Bayesian: 0.71, FC: 0.29; when comparing our main model to the FC: protected-EP$_{FC}$=0.46, protected-EP$_{BCI}$=0.54, posterior probabilities: RFX (random-effect analysis): p[H1|y]=0.860, null: p[H0|y]=0.140). Further details about the behavioral results for the synchrony judgment task can be found in the *Figure 3—figure supplement 3*.

## Discussion

The main finding of the present study is that body ownership perception can be described as a causal inference process that takes into account sensory uncertainty when determining whether an object is a part of one's own body or not. Participants performed a detection-like task on the ownership they felt over a rubber hand placed in full view in front of them in our version of the rubber hand illusion paradigm that involved the use of psychophysics, robotically controlled sensory stimulation, and augmented reality glasses (to manipulate visual noise); the tactile stimulation that the participants felt on their own hidden hand was synchronized with the taps applied to the rubber hand that they saw or systematically delayed or advanced. For each degree of asynchrony, the percentage of trials for which the participants felt like the rubber hand was theirs was determined. We found that the probability of the emergence of the rubber hand illusion was better predicted by a Bayesian model that takes into account the trial-by-trial level of sensory uncertainty to calculate the probability of a common cause for vision and touch given their relative onset time than by a non-Bayesian (FC) model that does not take into account sensory uncertainty. Furthermore, in comparing body ownership and visuotactile synchrony detection, we found interesting differences and similarities that advance our understanding of how the perception of multisensory synchrony and body ownership is related at the computational level and how individual differences in the rubber hand illusion can be explained as individual differences causal inference. Specifically, the prior probability of a common cause was found to be higher for ownership than for synchrony detection, and the two prior probabilities were found

to be correlated across individuals. We conclude that body ownership is a multisensory perception of one's own body determined by an uncertainty-based probabilistic inference of a common cause.

## Body ownership perception predicted by inference of a common cause

One of the strengths of the present study lies in its direct, individual-level testing of a causal inference model on body ownership perceptual data. This novel means to quantify the rubber hand illusion based on psychophysics is more appropriate for computational studies focused on body ownership than traditional measures such as questionnaires or changes in perceived hand position (proprioceptive drift). Previous attempts made to apply BCI to body ownership were conducted at the group level by the categorical comparison of experimental conditions (*Samad et al., 2015*); however, such a group-level approach does not properly challenge the proposed models as required according to standards in the field of computational behavioral studies. The only previous study that used quantitative Bayesian model testing analyzed target-reaching error in a virtual reality version of the rubber hand illusion (*Fang et al., 2019*), but reaching errors tend to be relatively small, and it is unclear how well the reaching errors correlate with the subjective perception of the illusion (*Heed et al., 2011*; *Kammers et al., 2009*; *Newport et al., 2010*; *Newport and Preston, 2011*; *Rossi Sebastiano et al., 2022*; *Zopf et al., 2011*). Thus, the present study contributes to our computational understanding of body ownership as the first direct fit of the BCI model to individual-level ownership sensations judged under the rubber hand illusion.

Computational approaches to body ownership can lead to a better understanding of the multisensory processing involved in this phenomenon than traditional descriptive approaches. The BCI framework informs us about how various sensory signals and prior information about body states are integrated at the computational level. Previous models of body ownership focus on temporal and spatial congruence rules and temporal and spatial 'windows of integration;' if visual and somatosensory signals occur within a particular time window (*Shimada et al., 2009*; *Costantini et al., 2016*) and within a certain spatial zone (*Lloyd, 2007*; *Brozzoli et al., 2012*), the signals will be combined, and the illusion will be elicited (*Ehrsson, 2012*; *Tsakiris, 2010*; *Makin et al., 2008*). However, these models do not detail how this happens at the computational level or explain how the relative contribution of different sensory signals and top-down prior information dynamically changes due to changes in uncertainty. Instead of occurring due to a sequence of categorical comparisons as proposed by *Tsakiris, 2010* or by a set of rigid temporal and spatial rules based on receptive field properties of multisensory neurons as implied by *Ehrsson, 2012* or *Makin et al., 2008*, body ownership under the rubber hand illusion arises as a consequence of a probabilistic computational process that infers the rubber hand as the common cause of vision and somatosensation by dynamically taking into account all available sensory evidence given their relative reliability and prior information. The causal inference model further has greater predictive power than classical descriptive models; in that, it makes quantitative predictions about how illusion perception will change across a wide range of temporal asynchronies and changes in sensory uncertainty. For example, the 'time window of integration' model – which is often used to describe the temporal constraint of multisensory integration (*Meredith et al., 1987*; *Stein and Meredith, 1993*) – only provides temporal thresholds (asynchrony between two sensory inputs) above which multisensory signals will not be integrated (*Colonius and Diederich, 2004*). In contrast, the present causal inference model explains how information from such asynchronies is used together with prior information and estimates of uncertainty to infer that the rubber hand is one's own or not. Even though the present study focuses on temporal visuotactile congruence, spatial congruence (*Fang et al., 2019*; *Samad et al., 2015*) and other types of multisensory congruences (e.g. *Ehrsson et al., 2005*; *Tsakiris et al., 2010*; *Ide, 2013*; *Crucianelli and Ehrsson, 2022*) would naturally fit within the same computational framework (*Körding et al., 2007*; *Sato et al., 2007*). Thus, in extending beyond descriptive models of body ownership, our study supports the idea that individuals use probabilistic representations of their surroundings and their own body that take into account information about sensory uncertainty to infer the causal structure of sensory signals and optimally process them to create a clear perceptual distinction between the self and nonself.

From a broader cognitive neuroscience perspective, causal inference models of body ownership can be used in future neuroimaging and neurophysiological studies to investigate the underlying neural mechanisms of the computational processes. For example, instead of simply identifying frontal, parietal, and subcortical structures that show higher activity in the illusion condition compared to

control conditions that violate temporal and spatial congruence rules (*Ehrsson et al., 2004*; *Ehrsson et al., 2005*; *Guterstam et al., 2013*; *Limanowski and Blankenburg, 2016*; *Guterstam et al., 2019a*; *Rao and Kayser, 2017*), one can test the hypothesis that activity in key multisensory areas closely follows the predictions of the BCI model and correlates with specific parameters of this model. Such a model-based imaging approach, recently successfully used in audiovisual paradigms (*Cao et al., 2019*; *Rohe and Noppeney, 2015*; *Rohe and Noppeney, 2016*; *Rohe et al., 2019*), can thus afford us a deeper understanding of the neural implementation of the causal inference for body ownership. From previous neuroimaging work (*Ehrsson et al., 2004*; *Guterstam et al., 2013*; *Limanowski and Blankenburg, 2016*; *Guterstam et al., 2019a*), anatomical and physiological considerations based on nonhuman primate studies (*Avillac et al., 2007*; *Graziano et al., 1997*; *Graziano et al., 2000*; *Fang et al., 2019*), and a recent model-based fMRI study on body ownership judgments (*Chancel et al., 2022*), we theorize that neuronal populations in the posterior parietal cortex and premotor cortex could implement the computational processes of the uncertainty-based inference of a common cause of body ownership.

## Observers take trial-to-trial sensory uncertainty into account in judging body ownership

The current study highlights the contribution of sensory uncertainty to body ownership by showing the superiority of a Bayesian model in predicting the emergence of the rubber hand illusion relative to a non-Bayesian model. Although BCI is an often-used model to describe multisensory processing from the behavioral to cerebral levels (*Badde et al., 2020*; *Cao et al., 2019*; *Dokka et al., 2019*; *Kayser and Shams, 2015*; *Körding et al., 2007*; *Rohe et al., 2019*; *Rohe and Noppeney, 2015*; *Wozny et al., 2010*), it is not uncommon to observe behaviors induced by sensory stimulation that diverge from strict Bayesian-optimal predictions (*Beck et al., 2012*). Some of these deviations from optimality can be explained by a contribution of sensory uncertainty to the perception that differs from that assumed under a Bayesian-optimal inference (*Drugowitsch et al., 2016*). Challenging the Bayesian-optimal assumption is thus a necessary good practice in computational studies (*Jones and Love, 2011*), and this is often done in studies of the perception of external sensory events, such as visual stimuli (*Qamar et al., 2013*; *Stengård and van den Berg, 2019*; *Zhou et al., 2020*). However, very few studies have investigated the role of sensory uncertainty in perceiving one's own limbs from a computational perspective. Such studies explore the perception of limb movement trajectory (*Reuschel et al., 2010*), limb movement illusion (*Chancel et al., 2016*), or perceived static limb position (*van Beers et al., 1999*; *van Beers et al., 2002*) but not the sense of body ownership or similar aspects of the embodiment of an object. These studies assume the full integration of visual and somatosensory signals and describe how sensory uncertainty is taken into account when computing a single-fused estimate of limb movement or limb position. However, none of these previous studies investigate inferences about a common cause. A comparison between Bayesian and non-Bayesian models was also missing from the above-described studies of the rubber hand illusion and causal inference (*Fang et al., 2019*; *Samad et al., 2015*). Thus, the current results reveal how uncertainty influences the automatic perceptual decision to combine or segregate bodily related signals from different sensory modalities and that this inference process better follows Bayesian principles than non-Bayesian principles. While we have argued that people take into account trial-to-trial uncertainty when making their body ownership and synchrony judgments, it is also possible that they learn a criterion at each noise level (*Ma and Jazayeri, 2014*), as one might predict in standard signal detection theory. However, we believe this is unlikely because we used multiple interleaved levels of noise while withholding any form of experimental feedback. Thus, more broadly, our results advance our understanding of the multisensory processes that support the perception of one's own body, as they serve as the first conclusive empirical demonstration of BCI in a bodily illusion. Such successful modeling of the multisensory information processing in body ownership is relevant for future computational work into bodily illusions and bodily self-awareness, for example, more extended frameworks that also include contributions of interoception (*Azzalini et al., 2019*; *Park and Blanke, 2019*), motor processes (*Burin et al., 2015*; *Burin et al., 2017*), pre-existing stored representations about what kind of objects that may or may not be part of one's body (*Tsakiris et al., 2010*), expectations (*Chancel et al., 2021*; *Guterstam et al., 2019b Ferri et al., 2013*), and high-level cognition (*Lush et al., 2020*; *Slater and Ehrsson, 2022*). Future quantitative computational studies like the present

one are needed to formally compare these different theories of body ownership and advance the corresponding theoretical framework.

In the present study, we compared the Bayesian hypothesis to a FC model. FC strategies are simple heuristics that could arise from limited sensory processing resources. Our body plays such a dominant and critical role in our experience of the world that one could easily imagine the benefits of an easy-to-implement heuristic strategy for detecting what belongs to our body and what does not. Our body is more stable than our ever-changing environment, so in principle, a resource-effective and straight-forward strategy for an observer could be to disregard, or not optimally compute, sensory uncertainty to determine whether an object in view is part of one's own body or not. However, our analysis shows that the BCI model outperforms such a model. Thus, observers seem to take into account trial-to-trial sensory uncertainty to respond regarding their body ownership perception. More visual noise, i.e., increased visual uncertainty, increases the probability of the rubber hand illusion, consistent with the predictions of Bayesian probabilistic theory. Intuitively, this makes sense, as it is easier to mistake one partner's hand for one's own under poor viewing conditions (e.g. in semidarkness) than when viewing conditions are excellent. However, this basic effect of sensory uncertainty on own-body perception is not explained by classical descriptive models of the rubber hand illusion (*Botvinick and Cohen, 1998*; *Tsakiris et al., 2010*; *Ehrsson, 2012*; *Makin et al., 2008*). Thus, the significant impact of sensory uncertainty on the rubber hand illusion revealed here advances our understanding of the computational principles of body ownership and of bodily illusions and multisensory bodily perception more generally.

## Relationship between body ownership and synchrony perception

The final part of our study focused on the comparison of causal inferences of body ownership and visuotactile synchrony detection. Previous studies have already demonstrated that audiovisual synchrony detection can be explained by BCI (*Adam and Noppeney, 2014*; *Magnotti et al., 2013*; *Noel et al., 2018*; *Noppeney and Lee, 2018*; *Shams et al., 2005*). We successfully extend this principle to visuotactile synchrony detection in the context of a rubber hand illusion paradigm. The results of our extension analysis using both ownership and synchrony data suggest that both multisensory perceptions follow similar computational principles in line with our expectations and previous literature. Whether the rubber hand illusion influences synchrony perception was not investigated in the present study, as the goal was to design ownership and synchrony tasks to be as identical as possible for the modeling. However, the results from the previous literature diverge regarding the potential influence of body ownership on synchrony judgment (*Ide and Hidaka, 2013*; *Maselli et al., 2016*; *Smit et al., 2019*), so this issue deserves further investigation in future studies.

Body ownership and synchrony perception were better predicted when modeling different priors instead of a single shared prior. The goodness of fit of the BCI model is greatly improved when the a priori probability of a common cause is different for each task, even when the loss of parsimony due to an additional parameter is taken into account. This result holds whether the two datasets are fitted together (extension analysis), or the parameters estimated for one task are used to fit the other (transfer analysis). Specifically, the estimates of the a priori probability of a common cause were found to be smaller for the synchrony judgment than for the ownership judgment. This means that the degree of asynchrony had to be lower for participants to perceive the seen and felt taps as occurring simultaneously compared to the relatively broader degree of visuotactile asynchrony that still resulted in the illusory ownership of the rubber hand. This result suggests that a common cause for vision and touch outcomes is a priori more likely to be inferred for body ownership than for visuotactile synchrony. We believe that this makes sense, as a single cause for visual and somatosensory impressions in the context of the ownership of a human-like hand in an anatomically matched position in sight is a priori a more probable scenario than a common cause for brief visual and tactile events that in principle could be coincidental and stem from visual events occurring far from the body. This observation is also consistent with previous studies reporting the induction of the rubber hand illusion for visuotactile asynchronies of as long as 300 ms (*Shimada et al., 2009*), which are perceptually noted. While it seems plausible that $p_{same}$ reflects the real-world prior probability of a common cause of the visual and somatosensory signals, it could also be influenced by experimental properties of the task, demand characteristics (participants forming beliefs based on cues present in a testing situation,

*Weber and Cook, 1972*; *Corneille and Lush, 2022*; *Slater and Ehrsson, 2022*), and other cognitive biases.

How the a priori probabilities of a common cause under different perceptive contexts are formed remains an open question. Many studies have shown the importance of experience in shaping the prior (*Adams et al., 2004*; *Chambers et al., 2017*; *Snyder et al., 2015*), and recent findings also seem to point toward the importance of effectors in sensorimotor priors (*Yin et al., 2019*) and dynamical adjustment during a task (*Prsa et al., 2015*). In addition, priors for own-body perception could be shaped early during development (*Bahrick and Watson, 1985*; *Bremner, 2016*; *Rochat, 1998*) and influenced by genetic and anatomical factors related to the organization of cortical and subcortical maps and pathways (*Makin and Bensmaia, 2017*; *Stein et al., 2014*).

The finding that prior probabilities for a common cause were correlated for the ownership and synchrony data suggests a shared probabilistic computational process between the two multisensory tasks. This result could account for the previously observed correlation at the behavioral level between individual susceptibility to the rubber hand illusion and individual temporal resolution ('temporal window of integration') in visuotactile synchrony perception (*Costantini et al., 2016*). It is not that having a narrower temporal window of integration makes one more prone to detect visuotactile temporal mismatches leading to a weaker rubber hand illusion as the traditional interpretation assumes. Instead, our behavioral modeling suggests that the individual differences in synchrony detection and the rubber hand illusion can be explained by individual differences in how prior information on the likelihood of a common cause is used in multisensory causal inference. This probabilistic computational explanation for individual differences in the rubber hand illusion emphasizes differences in how information from prior knowledge, bottom-up sensory correspondence, and sensory uncertainty is combined in a perceptual inferential process rather than there being 'hard-wired' differences in temporal windows of integration or trait differences in top-down cognitive processing (*Eshkevari et al., 2012*; *Germine et al., 2013*; *Marotta et al., 2016*). It should be noted that other multisensory factors not studied in the present study can also contribute to individual differences in the rubber hand illusion, notably as the relative reliability of proprioceptive signals from the upper limb (*Horváth et al., 2020*). The latter could be considered in future extensions of the current model that also consider the degree of spatial disparity between vision and proprioception and the role of visuoproprioceptive integration (*Samad et al., 2015*; *Fang et al., 2019*; *Kilteni et al., 2015*).

## Conclusion

BCI models have successfully described many aspects of perception, decision making, and motor control, including sensory and multisensory perception of external objects and events. The present study extends this probabilistic computational framework to the sense of body ownership, a core aspect of self-representation and self-consciousness. Specifically, the study presents direct and quantitative evidence that body ownership detection can be described at the individual level by the inference of a common cause for vision and somatosensation, taking into account trial-to-trial sensory uncertainty. The fact that the brain seems to use the same probabilistic approach to interpret the external world and the self is of interest to Bayesian theories of the human mind (*Ma and Jazayeri, 2014*; *Rahnev, 2019*) and suggests that even our core sense of conscious bodily self (*Blanke et al., 2015*; *Ehrsson, 2020*; *Tsakiris, 2017*; *de Vignemont, 2018*) is the result of an active inferential process making 'educated guesses' about what we are.

## Materials and methods

### Participants

18 healthy participants naïve to the conditions of the study were recruited for this experiment (six males, aged 25.2±4 years, right-handed; they were recruited from outside the department, never having taken part in a bodily illusion experiment before). Note that in computational studies such as the current one, the focus is on fitting and comparing models within participants, i.e., to rigorously quantify perception at the single-subject level, and not only rely on statistical results at the group level. All volunteers provided written informed consent prior to their participation. All participants received 600 SEK (Swedish krona) as compensation for their participation (150 SEK per hr). All experiments were approved by the Swedish Ethics Review Authority (Ethics number 2018/471-31/2).

## Inclusion test

In the main experiment, participants were asked to judge the ownership they felt toward the rubber hand. It was therefore necessary for them to be able to experience the basic rubber hand illusion. However, we know that approximately 20–25% of healthy participants do not report a clear and reliable rubber hand illusion (*Kalckert and Ehrsson, 2014*), and such participants are not able to make reliable ownership discriminations in psychophysics tasks (*Chancel and Ehrsson, 2020*), which were required for the current modeling study (they tended to respond randomly). Thus, all participants were first tested on a classical rubber hand illusion paradigm to ensure that they could experience the illusion. For this test, each participant sat with their right hand resting on a support beneath a small table. On this table, 15 cm above the hidden real hand, the participant viewed a life-sized cosmetic prosthetic male right hand (model 30,916 R, Fillauer, filled with plaster; a 'rubber hand') placed in the same position as the real hand. The participant kept their eyes fixed on the rubber hand while the experimenter used two small probes (firm plastic tubes, diameter: 7 mm) to stroke the rubber hand and the participant's hidden hand for 12 s, synchronizing the timing of the stroking as much as possible. Each stroke lasted 1 s and extended approximately 1 cm; the strokes were applied to five different points along the real and rubber index fingers at a frequency of 0.5 Hz. The characteristics of the strokes and the duration of the stimulation were designed to resemble the stimulation later applied by the robot during the discrimination task (see below). Then, the participant completed a questionnaire adapted from that used by *Botvinick and Cohen, 1998*, see also *Chancel and Ehrsson, 2020* and *Figure 4—figure supplement 1*. This questionnaire includes three items assessing the illusion and four control items to be rated with values between –3 ('I completely disagree with this item') and 3 ('I completely agree with this item'). Our inclusion criteria for a rubber hand illusion strong enough for participation in the main psychophysics experiment were as follows: (1) a mean score for the illusion statements (Q1, Q2, and Q3) of greater than 1 and (2) a difference between the mean score for the illusion items and the mean score for the control items of greater than 1. Three participants (two females) did not reach this threshold; therefore, 15 subjects participated in the main experiment (five males, aged 26.3±4 years, *Figure 4—figure supplement 2*). The inclusion test session lasted 30 min in total. After this inclusion phase, the participants were introduced to the setup used in the main experiment.

## Experimental setup

During the main experiment, the participant's right hand lay hidden, palm down, on a flat support surface beneath a table (30 cm lateral to the body midline), while on this table (15 cm above the real hand), a right rubber hand was placed in the same orientation as the real hand aligned with the participants' arm (*Figure 4A*). The participant's left hand rested on their lap. A chin rest and elbow rest (Ergorest Oy, Finland) ensured that the participant's head and arm remained in a steady and relaxed position throughout the experiments. Two robot arms (designed in our laboratory by Martti Mercurio and Marie Chancel, see *Chancel and Ehrsson, 2020* for more details) applied tactile stimuli (taps) to the index finger of the rubber hand and to the participant's hidden real index finger. Each robot arm was composed of three parts: two 17-cm-long, 3-cm-wide metal pieces and a metal slab (10×20 cm) as a support. The joint between the two metal pieces and that between the proximal piece and the support was powered by two HS-7950TH Ultra Torque servos that included 7.4 V optimized coreless motors (Hitec Multiplex, USA). The distal metal piece ended with a ring containing a plastic tube (diameter: 7 mm) that was used to touch the rubber hand and the participant's real hand.

During the experiment, the participants wore augmented reality glasses: a Meta2 VR headset with a 90° field of view, 2560×1440 high-dpi display, and 60 Hz refresh rate (Meta View Inc). Via this headset, the uncertainty of the visual scene could be manipulated: The probability of a pixel of the scene observed by the participant turning white from one frame to the other varied (frame rate: 30 images/s); when turning white, a pixel became opaque, losing its meaningful information (information on the rubber hand and robot arm touching the rubber hand) and therefore becoming irrelevant to the participant. The higher the probability of the pixels turning white becomes, the more uncertain the visual information becomes. During the experiment, the participants wore earphones playing white noise to cancel out any auditory information from the robots' movements that might have otherwise interfered with the behavioral task and with illusion induction (*Radziun and Ehrsson, 2018*).

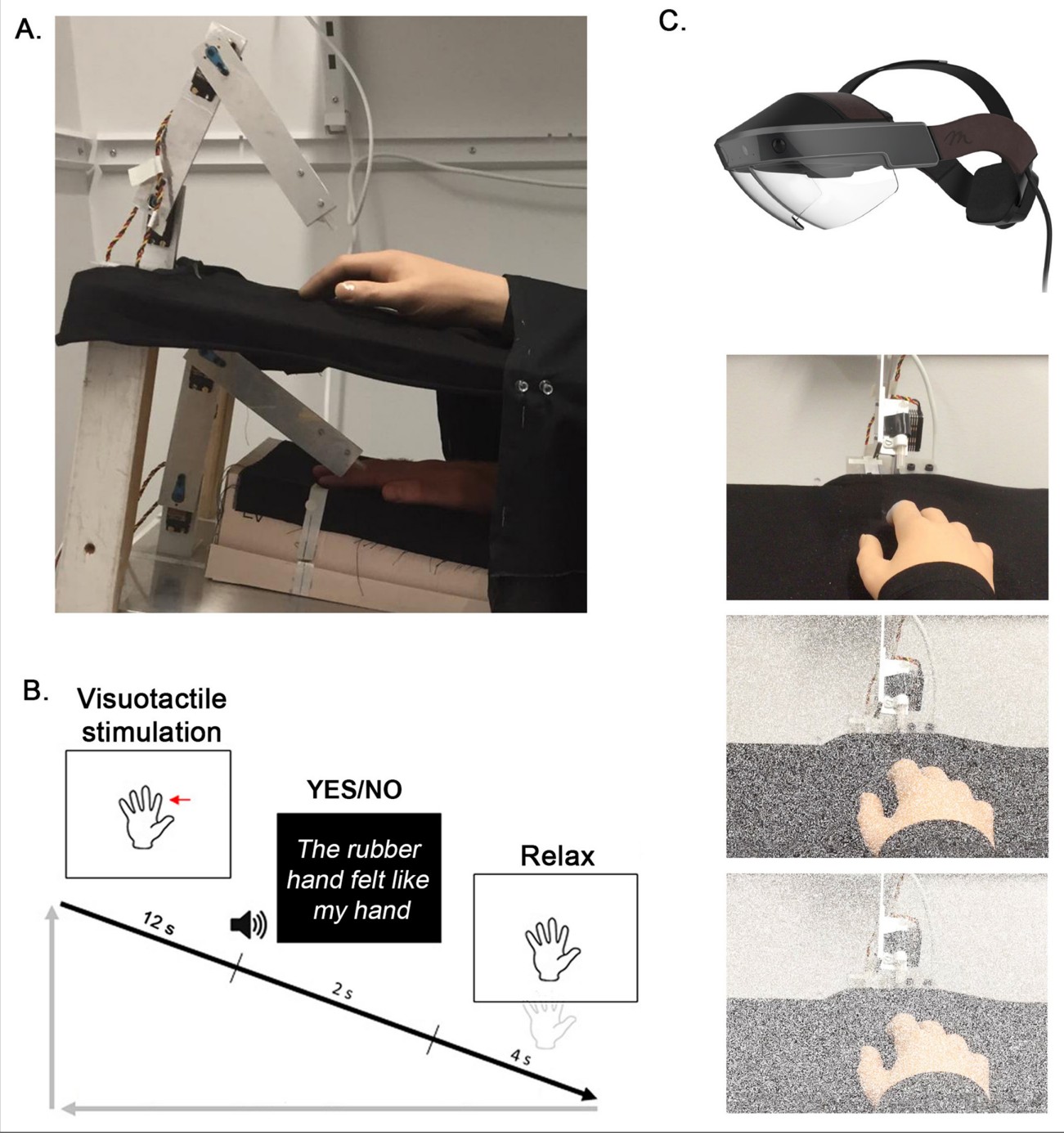

**Figure 4.** Experimental setup (**A**) and experimental procedure (**B and C**) for the ownership judgment task. A participant's real right hand is hidden under a table while they see a life-sized cosmetic prosthetic right hand (rubber hand) on the table (**A**). The rubber hand and real hand are touched by robots for periods of 12 s, either synchronously or with the rubber hand touched slightly earlier or later at a degree of asynchrony that is systematically manipulated (±150 ms, ±300 ms, or ± 500ms). The participant is then required to state whether the rubber hand felt like their own hand or not ('yes' or 'no' forced choice task) (**B**). Using the Meta2 headset, three noise conditions are tested: 0 (top picture), 30 (middle picture), and 50% (bottom picture) visual noise (**C**).

The online version of this article includes the following figure supplement(s) for figure 4:

**Figure supplement 1.** Questionnaire.

**Figure supplement 2.** Mean questionnaire results for the participants included in the main experiment.

## Procedure

The main experiment involved two tasks conducted in two different sessions: a body ownership judgment task and a synchrony judgment task. Both tasks were yes/no psychophysical detection tasks (*Figure 4B*).

## Body ownership judgment task

In each trial, the participant was asked to decide whether the rubber hand felt like their own hand, i.e., to determine whether they felt the key phenomenological aspect of the rubber hand illusion (*Botvinick and Cohen, 1998*; *Ehrsson et al., 2004*; *Longo et al., 2008*). Each trial followed the same sequence. The robots repeatedly tapped the index fingers of the rubber hand and the actual hand six times each for a total period of 12 s in five different locations in randomized order ('stimulation period'): immediately proximal to the nail on the distal phalanx, on the distal interphalangeal joint, on the middle phalanx, on the proximal interphalangeal joint, and on the proximal phalanx. All five locations were stimulated at least once in each 12 s trial, and the order of stimulation sites randomly varied from trial to trial. The participant was instructed to focus their gaze on the rubber hand. Then, the robots stopped while the participant heard a tone instructing them to verbally report whether the rubber hand felt like their own hand by saying 'yes' (the rubber hand felt like it was my hand) or 'no' (the rubber hand did not feel like it was my hand). This answer was registered by the experimenter. A period of 12 s was chosen in line with a previous rubber hand illusion psychophysics study (*Chancel and Ehrsson, 2020*), and because earlier studies with individuals susceptible to the illusion have shown that the illusion is reliably elicited in approximately 10 s (*Ehrsson et al., 2004*; *Guterstam et al., 2013*; *Lloyd, 2007*), different locations on the finger were chosen to prevent the irritation of the skin during the long psychophysics session and in line with earlier studies stimulating different parts of the hand and fingers to elicit the rubber hand illusion (e.g. *Guterstam et al., 2011*). During this period of stimulation, the participant was instructed to look at and focus on the rubber hand.

After the stimulation period and the body ownership judgment answer, the participant was asked to wiggle their right fingers to avoid any potential numbness or muscle stiffness from keeping their hand still and to eliminate possible carry-over effects to the next stimulation period by breaking the rubber hand illusion (moving the real hand while the rubber hand remained immobile eliminates the rubber hand illusion). The participant was also asked to relax their gaze by looking away from the rubber hand because fixating on the rubber hand for a whole session could have been uncomfortable. 5 s later, a second tone informed the participant that the next trial was about to start; the next trial started 1 s after this sound cue.

Two variables were manipulated in this experiment: (1) the synchronicity between the taps that seen and those felt by the participants (asynchrony condition) and (2) the level of visual white noise added to the visual scene (noise condition). Seven different asynchrony conditions were tested. The taps on the rubber hand could be synchronized with the taps on the participant's real hand (synchronous condition) or could be delayed or advanced by 150, 300, or 500 ms. In the rest of this article, negative values of asynchrony (−150,−300, and −500 ms) mean that the rubber hand was touched first, and positive values of asynchrony (+150,+300, and +500 ms) mean that the participant's hand was touched first. The seven levels of asynchrony appeared with equal frequencies in pseudorandom order so that no condition was repeated more than twice in a row. The participants did not know how many different asynchrony levels were tested (as revealed in unformal post-experiment interviews) and that no feedback was given on their task performance. Three different noise conditions were tested, corresponding to 0, 30, and 50% of visual noise being displayed, i.e., the pixels of the Meta2 headset screen could turn white from one frame to another with a probability of 0, 30, or 50% (*Figure 4C*). The three levels of noise also appeared with equal frequencies in pseudorandom order. During the experiment, the experimenter was blind to the noise level presented to the participants, and the experimenter sat out of the participants' sight.

## Visuotactile synchrony judgment task

During this task, the participant was asked to decide whether the visual and tactile stimulation they received happened at the same time, i.e., whether the felt and seen touches were synchronous or not. The procedure was identical to the body ownership detection task apart from a critical difference in the instructions, which was now to determine if the visual and tactile stimulations were synchronous

(instead of judging illusory rubber hand ownership). In each trial, a 12 s visuotactile stimulation period was followed by the yes/no verbal answer given by the participant and a 4 s break. The same two variables were manipulated in this experiment: the synchronicity between the seen and felt taps (asynchrony condition) and the level of visual white noise (noise condition). The asynchronies used in this synchrony judgment task were lesser than those of the ownership judgment task (±50, ±150, or ±300 ms instead of ±150, ±300, or ±500 ms) to maintain an equivalent difficulty level between the two tasks; this decision was made based on a pilot study involving 10 participants (three males, aged 27.0±4 years, different than the main experiment sample) who performed the ownership and synchrony tasks under 11 different levels of asynchrony (*Appendix 1—table 3* and *Figure 2*). The noise conditions were identical to those used for the ownership judgment task.

The ordering of the tasks was counterbalanced across the participants. Each condition was repeated 12 times, leading to a total of 252 judgments made per participant and task. The trials were randomly divided into three experimental blocks per task, each lasting 13 min.

## Modeling

As explained in the introduction, we assumed that the rubber hand illusion is driven by the integration of visual and tactile signals in the current paradigm. To describe this integration, we designed a model in which the observer performs BCI; we compare this model to a non-Bayesian model. We then extended the same models of the synchrony judgment task and examined whether the same model with the same parameters could describe a participant's behavior in both tasks.

## BCI model for body ownership

We first specify the BCI model for body ownership. A more detailed and step-by-step description of the modeling can be found in Appendix 1.

## Generative model

Bayesian inference is based on a generative model, which is a statistical model of the world that the observer believes to give rise to observations. By 'inverting' this model for a given set of observations, the observer can make an 'educated guess' about a hidden state. Therefore, we first must specify the generative model that captures both the statistical structure of the task as assumed by the observer and an assumption about measurement noise. In our case, the model contains three variables: the causal structure category $C$, the tested asynchrony $s$, and the measurement of this asynchrony by the participant $x$. Even though the true frequency of synchronous stimulation ($C$=1) is 1/7=0.14, we allow it to be a free parameter, which we denote as $p_{same}$. One can view this parameter as an incorrect belief, but it can equivalently be interpreted as a perceptual or decisional bias. Next, when $C$=1, the asynchrony $s$ is always 0; we assume that the observer knows this. When $C$=2, the true asynchrony takes one of several discrete values; we do not assume that the observer knows these values or their probabilities and instead assume that the observer assumes that asynchrony is normally distributed with the correct SD $\sigma_S$ of 348 ms (i.e. the true SD of the stimuli used in this experiment). In other words, $p\left(s|C = 2\right) = N\left(s; 0, \sigma_s^2\right)$. Next, we assume that the observer makes a noisy measurement $x$ of the asynchrony. We make the standard assumption (inspired by the central limit theorem) that this noise follows the below a normal distribution:

$$p\left(x|s\right) = N\left(x; s, \sigma^2\right)$$

where the variance depends on the sensory noise for a given trial. Finally, we assume that the observer has accurate knowledge of this part of the generative model.

## Inference

Now that we have specified the generative model, we can turn to inference. Visual and tactile inputs are to be integrated, leading to the emergence of the rubber hand illusion if the observer infers a common cause ($C = 1$) for both sensory inputs. On a given trial, the model observer uses $x$ to infer the category $C$. Specifically, the model observer computes the posterior probabilities of both categories, $p\left(C = 1|x\right)$ and $p\left(C = 2|x\right)$, i.e., the belief that the category was $C$. Then, the observer would report

'yes, it felt like the rubber hand was my own hand' if the former probability were higher, or in other words, when $d > 0$, where

$$d = \log \frac{p(C=1|x)}{p(C=2|x)} .$$

This equation can be written as a sum of the log prior ratio and the log-likelihood ratio:

$$d = log\left(\frac{p_{\text{same}}}{1-p_{\text{same}}}\right) + log\left(\frac{p(x_{\text{trial}}|C=1)}{p(x_{\text{trial}}|C=2)}\right) \ \#$$

The decision rule $d>0$ is thus equivalent to (see the Appendix 1)

$$|x| < k$$

where

$$k = \sqrt{K}$$

and

$$K = \frac{\sigma^2 \ (\sigma_s^2 + \sigma^2)}{\sigma_s^2} \left(2\log \frac{p_{\text{same}}}{1-p_{\text{same}}} + \log \frac{\sigma_s^2 + \sigma^2}{\sigma^2}\right)$$

where $\sigma$ is the sensory noise level of the trial under consideration. As a consequence, the decision criterion changes as a function of the sensory noise affecting the observer's measurement (*Figure 5*). This is a crucial property of BCI and indeed a property shared by Bayesian models used in previous work on multisensory synchrony judgments (*Magnotti et al., 2013*), audiavisual spatial localization (*Körding et al., 2007*), visual searching (*Stengård and van den Berg, 2019*), change detection (*Keshvari et al., 2012*), collinearity judgment (*Zhou et al., 2020*), and categorization (*Qamar et al., 2013*). The output of the BCI model is the probability of the observer reporting the visual and tactile inputs as emerging from the same source when presented with a specific asynchrony value $s$:

$$p\left(\hat{C} = 1|s\right) = 0.5\lambda + (1 - \lambda)\left(\Phi\left(s; \ k, \ \sigma^2\right) - \Phi\left(s; -k, \ \sigma^2\right)\right)$$

Here, the additional parameter $\lambda$ reflects the probability of the observer lapsing, i.e., randomly guessing. This equation is a prediction of the observer's response probabilities and can thus be fit to a participant's behavioral responses.

The BCI model has five free parameters: $p_{\text{same}}$: the prior probability of a common cause for vision and touch, independent of any sensory stimulation, $\sigma_0$, $\sigma_{30}$, $\sigma_{50}$ : the noise impacting the measurement $x$ specific to each noise condition, and $\lambda$: a lapse rate to account for random guesses and unintended responses. We assumed a value of 348 ms for $\sigma_S$ , i.e., $\sigma_S$ is equal to the actual SD of the asynchronies used in the experiment, but we challenged this assumption later. Moreover, in our experiment, the spatial parameters and the proprioceptive state of our participants are not manipulated or altered from one condition to the other. Thus, our model focuses on the temporal aspects of the visuotactile integration in the context of body ownership. In this, it differs from the model proposed by *Samad et al., 2015* in which both spatial and temporal aspects were modeled separately and then averaged to obtain an estimate of body ownership (that they then compared with questionnaire ratings of rubber hand illusion).

## Alternative models
### BCI model for body ownership with a free level of uncertainty impacting the stimulation (BCI*)
For the BCI model, we assumed that the observer's assumed stimulus distribution has the same SD $\sigma_S$ as the true stimulus distribution. We also tested a variant in which the assumed SD $\sigma_S$ is a free parameter. As a result, this model is less parsimonious than the BCI model. The model has six free parameters ($p_{\text{same}}$, $\sigma_0$, $\sigma_{30}$, $\sigma_{50}$, $\sigma_S$, and $\lambda$). Nevertheless, the decision rule remains the same as that of the BCI model.

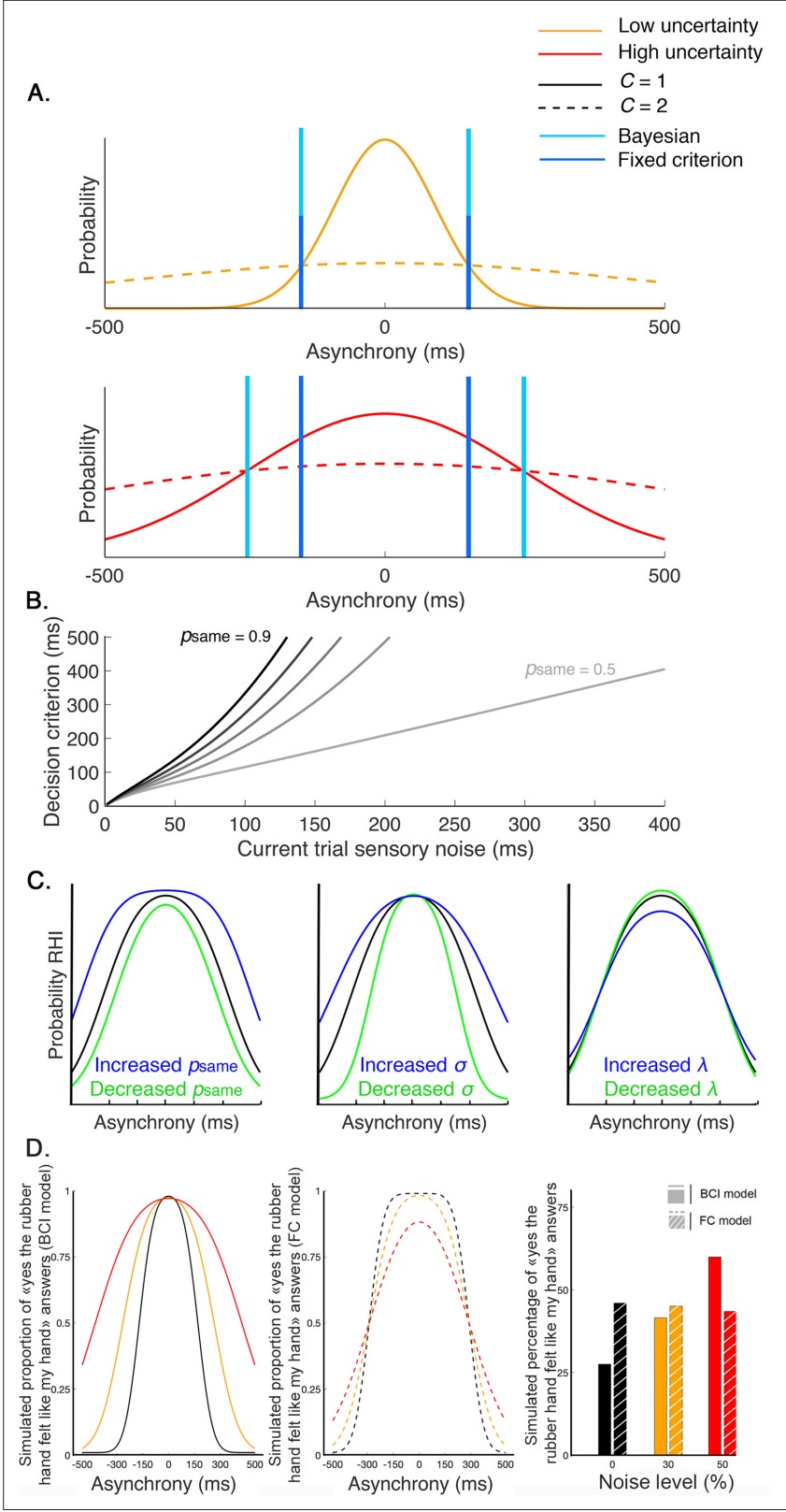

**Figure 5.** Decision process for the emergence of the rubber hand illusion (RHI) according to the Bayesian and fixed criterion observers. (**A**) The measured asynchrony between the visual and tactile events for the low (orange) or high (red) noise level conditions and the probability of the different causal scenarios: the visual and tactile events come from one source, the observer's body, or from two different sources. The probability of a

*Figure 5 continued on next page*

*Figure 5 continued*

common source is a narrow distribution (full curves), and the probability of two distinct sources is a broader distribution (dashed curve), both centered on synchronous stimulation (0 ms) such that when the stimuli are almost synchronous, it is likely that they come from the same source. When the variance of the measured stimulation increases from trial to trial, decision criteria may adjust optimally (Bayesian – light blue) or stay fixed (fixed – dark blue). The first assumption corresponds to the Bayesian causal inference (BCI) model, and the second corresponds to the fixed criterion (FC) model (see next paragraph for details). The displayed distributions are theoretical, and the BCI model's p~same~ is arbitrarily set at 0.5. (**B**) The decision criterion changes from trial to trial as a function of sensory uncertainty according to the optimal decision rule from the BCI model. Black curves represent this relationship for different p~same~ values of 0.4–0.9 (from lightest to darkest). (**C**) From left to right, these last plots illustrate how the BCI model-predicted outcome is shaped by $p_{same}$, $\sigma$, and $\lambda$, respectively. Left: $p_{same}$ = 0.8 (black), 0.6 (green), and 0.9 (blue). Middle: $\sigma$ = 150 ms (black), 100 ms (green), and 200 ms (blue). Right: $\lambda$ = 0.05 (black), 0.005 (green), and 0.2 (blue). (**D**) Finally, this last plot shows simulated outcomes predicted by the BCI model (in full lines and bars) and the FC model (in dashed lines and shredded bars). In this theoretical simulation, both models predict the same outcome distribution for one given level of sensory noise (0%); however, since the decision criterion of the BCI model is adjusted to the level of sensory uncertainty, an overall increase of the probability of emergence of the RHI is predicted by this Bayesian model. On the contrary, the FC model, which is a non-Bayesian model, predicts a neglectable effect of sensory uncertainty on the overall probability of emergence of the RHI.

## FC (non-Bayesian) model

An important alternative to the Bayesian model is a model that ignores variations in sensory uncertainty when judging if the rubber hand is one's own, for example, because the observer incorrectly assumes that sensory noise does not change. We refer to this as the FC model. The decision rule for the FC model then becomes the following:

$$|x| < k_0,$$

where $k_0$ corresponds to an FC for each participant, which does not vary with trial-to-trial sensory uncertainty. If the decisional stage is independent of the trial-to-trial sensory uncertainty, the encoding stage is still influenced by the level of sensory noise. Thus, the output of the FC model is the probability of the observer reporting the illusion when presented with a specific asynchrony value $s$:

$$p\left(illusion|s\right) = 0.5\lambda + \left(1 - \lambda\right)\left(\Phi\left(s;\ k_0,\ \sigma^2\right) - \Phi\left(s;\ -k_0,\ \sigma^2\right)\right)$$

Again, the additional parameter $\lambda$ reflects the probability of the observer lapsing, i.e., randomly guessing. This equation is a prediction of the observer's response probabilities and can thus be fitted to a participant's behavioral responses.

## Parameter estimation

All model fitting was performed using maximum-likelihood estimation implemented in MATLAB (Math-Works). We used both the built-in MATLAB function fmincon and the Bayesian adaptive directed search (BADS) algorithm (*Acerbi and Ma, 2017*), each using 100 different initial parameter combinations per participant. Fmincon is gradient based, while BADS is not. The best estimate from either of these two procedures was kept, i.e., the set of estimated parameters that corresponded to the maximal log-likelihood for the models. Fmincon and BADS produced the same log-likelihood for the BCI, BCI*, and FC models for 12, 13, and 14 of the 15 participants, respectively. For the remaining participants, the BADS algorithm performed better. Moreover, the fitting procedure run 100 times (with different initial parameter combinations) led to the same set of estimated parameters at least 31 times for all participants and models. To validate our procedure, we performed parameter recovery. For this procedure, data simulated from random parameters were fitted using the models we designed. Because the generating random parameters were recovered, i.e., are similar to the estimated parameters, we are confident that the parameter estimation applied for the fitting procedure used in the current study is reliable (*Appendix 1—figure 1* & *Appendix 1—table 2*).

## Model comparison

The Akaike information criterion (AIC; *Akaike, 1973*) and Bayesian information criterion (BIC; *Schwarz, 1978*) were used as measures of goodness of model fit. The lower the AIC or BIC, the better the fit. The BIC penalizes the number of free parameters more heavily than the AIC. We calculated AIC and BIC values for each model and participant according to the following equations:

$$AIC = 2n_{\text{par}} - 2\log L^*$$

$$BIC = n_{\text{par}} \log n_{\text{trials}} - 2\log L^*$$

where $L^*$ is the maximized value of the likelihood, $n_{\text{par}}$ the number of free parameters, and $n_{\text{trial}}$ the number of trials. We then calculated the AIC and BIC difference between models and summed across the participants. We estimated a CI using bootstrapping: 15 random AIC/BIC differences were drawn with replacement from the actual participants' AIC/BIC differences and summed; this procedure was repeated 10,000 times to compute the 95% CI.

As an additional assessment of the models, we compute the coefficient of determination $R^2$ (Nagelkerke, 1991) defined as

$$R^2 = 1 - \exp\left(-\tfrac{2}{n}\left(\log L\left(M\right) - \log L\left(M_0\right)\right)\right)$$

where $\log L\left(M\right)$ and $\log L\left(M_0\right)$ denote the log-likelihoods of the fitted and the null model, respectively, and n is the number of data points. For the null model, we assumed that an observer randomly chooses one of the two response options, i.e., we assumed a discrete uniform distribution with a probability of 0.5. As in our case the models' responses were discretized to relate them to the two discrete response options, the coefficient of determination was divided by the maximum coefficient (Nagelkerke, 1991) defined as

$$\max\left(R^2\right) = 1 - \exp\left(\tfrac{2}{n}\log L\left(M_0\right)\right).$$

We also performed Bayesian model selection (*Rigoux et al., 2014*) at the group level to obtain the exceedance probability for the candidate models (i.e. the probability that a given model is more likely than any other model given the data) using the VBA (Variational Bayesian Analysis) toolbox (*Rigoux et al., 2014*). With this analysis, we consider a certain degree of heterogeneity in the population instead of assuming that all participants follow the same model and assess the a posteriori probability of each model.

## Ownership and synchrony tasks

The experimental contexts of the ownership and synchrony judgment tasks only differed in the instructions given to the participants regarding which perceptual feature they were to detect (rubber hand ownership or visuotactile synchrony). Thus, the bottom-up processing of the sensory information is assumed to be the same. In particular, the uncertainty impacting each sensory signal is likely to be the same between the two tasks, since the sensory stimulation delivered to the observer is identical. The difference in the participants' synchrony and ownership perceptions should be reflected in the a priori probability of the causal structure. For our BCI model, this means that the $\sigma_0$, $\sigma_{30}$, and $\sigma_{50}$ parameters are assumed to be the same for the two tasks. The same applies for the lapse rate $\lambda$ that depends on the observer and not on the task. In contrast, the prior probability for a common cause $p_{\text{same}}$ could change when a different judgment (ownership or synchrony) is assessed.

We used two complementary approaches to test whether people show different prior probabilities of a common cause for body ownership and synchrony perceptions: an extension analysis and a transfer analysis. In the extension analysis, we applied our BCI model to both sets of data and compared the fit of the model with all parameters ($p_{\text{same}}$, $\sigma_0$, $\sigma_{30}$, $\sigma_{50}$, $\sigma_S$, and $\lambda$) shared between tasks to a version of the model with one probability of a common cause $p_{\text{same, ownership}}$ for the body ownership task only and one probability of a common cause $p_{\text{same, synchrony}}$ for the synchrony task only. In the transfer analysis, we used the estimated parameters for one task (ownership or synchrony) to predict the data from the other task (synchrony or ownership). We compared a full transfer, in which all previously estimated parameters were used, to a partial transfer, in which $p_{\text{same}}$ was left as a free parameter. We again used the AIC and BIC to compare the different models.

## Acknowledgements

We would like to thank Martti Mercurio for his help in building the robots and writing the program to control them. We also thank Pius Kern and Birgit Hasenack for their help with data acquisition during the pilot phase of this study. We thank the reviewers for their constructive feedback during the reviewing process that helped us improve the article.

This research was funded by the Swedish Research Council, the Göran Gustafssons Foundation, and the European Research Council under the European Union's Horizon 2020 research and innovation programme (grant 787386 SELF-UNITY). MC was funded by a postdoctoral grant from the Wenner-Gren Foundation.

## Additional information

### Funding

| Funder | Grant reference number | Author |
|---|---|---|
| European Research Council | 787386 | H Henrik Ehrsson |
| Wenner-Gren Foundation | | Marie Chancel |

The funders had no role in study design, data collection and interpretation, or the decision to submit the work for publication.

### Author contributions

Marie Chancel, Conceptualization, Data curation, Formal analysis, Investigation, Visualization, Methodology, Writing – original draft, Project administration, Writing – review and editing; H Henrik Ehrsson, Conceptualization, Resources, Supervision, Funding acquisition, Validation, Writing – original draft, Writing – review and editing; Wei Ji Ma, Conceptualization, Supervision, Validation, Methodology, Writing – original draft, Writing – review and editing

### Author ORCIDs

Marie Chancel ⓘ http://orcid.org/0000-0002-3052-5268
H Henrik Ehrsson ⓘ http://orcid.org/0000-0003-2333-345X
Wei Ji Ma ⓘ http://orcid.org/0000-0002-9835-9083

### Ethics

Human subjects: All volunteers provided written informed consent prior to their participation. All experiments were approved by the Swedish Ethics Review Authority (Ethics number 2018/471-31/2).

### Decision letter and Author response

Decision letter https://doi.org/10.7554/eLife.77221.sa1
Author response https://doi.org/10.7554/eLife.77221.sa2

## Additional files

### Supplementary files

• Transparent reporting form

### Data availability

Figure 1—source data 1, Figure 2—source data 1 and Figure 3—source data 1 contain the numerical data used to generate the figures and their supplements. These files have also been made available: https://osf.io/zu2h6/.

The following dataset was generated:

| Author(s) | Year | Dataset title | Dataset URL | Database and Identifier |
|---|---|---|---|---|
| Chancel M, Ehrsson HH, Ma WJ | 2021 | Uncertainty-based inference of a common cause for body ownership | https://osf.io/n7atw/ | Open Science Framework, n7atw |

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

## Appendix 1

### 1. Bayesian causal inference model for body ownership

Bayesian models typically require three steps: first, specification of the generative model, which represents the statistics of the variables and their relationships, as believed by the observer; second, specification of the actual inference process, in which the observer uses a particular observation and 'inverts' the generative model to build a posterior distribution over the world state of interest; and third, specification of the predicted response distribution, which can be directly related to data. Below, we lay out these three steps for the body ownership task, in which the observer judges whether the rubber hand is theirs or not. For synchrony detection task, everything is the same except for the interpretation of the category variable $C$.

### Step 1: generative model

We first need to specify the generative model, which captures the statistical structure of both the task and the measurement noise, as assumed by the observer. It contains three variables: the category, $C$, the physical visuotactile asynchrony, $s$, and the noisy measurement of this asynchrony, $x$. The variable $C$ represents the high-level scenario:

$C = 1$: only one common source, hence the rubber hand is my hand.

$C = 2$: two different sources, hence the rubber hand is not my hand.

The a priori probability of a common cause, before any sensory stimulation is delivered to the observer is expressed as:

$$p\left(C = 1\right) = p_{\text{same}}$$

Next, we assume that the observer correctly assumes that the asynchrony $s$ is always zero when $C = 1$, and incorrectly assumes that the asynchrony follows a Gaussian distribution with standard deviation $\sigma_s$ when $C = 2$:

$$p\left(s|C = 1\right) = \delta\left(s\right) \tag{1}$$

$$p\left(s|C = 2\right) = N\left(s; 0, \sigma_s^2\right) \tag{2}$$

Note that the distribution $p\left(s|C = 2\right)$ is not the experimental asynchrony distribution that would be a mixture of delta functions, because in the $C = 2$ condition, we presented a discrete set of asynchronies (±500 ms, ±300 ms, ±150 ms, and 0 ms). Why do we assume that the observer's assumed asynchrony distribution for $C = 2$ is different from the experimental one? We reasoned that it is unlikely that our participants were aware of the discrete nature of the experimental distribution, and that it is more likely that they assumed the distribution to be continuous. We use a Gaussian distribution because, in view of its simplicity and frequent occurrence, this seems to be a distribution that participants could plausibly assume. We tested both a model in which the SD of the Gaussian is equal to the experimental SD, and one in which it is not necessarily so (and therefore fitted as a free parameter).

Finally, we assume that the observer assumes that the measured asynchrony $x$ is affected by a Gaussian noise $\sigma$:

$$p\left(x|s\right) = N\left(x; s, \sigma^2\right) \tag{3}$$

This assumption is standard and loosely motivated by the central limit theorem.

### Step 2: inference

We now move to the inference performed by the observer. Visual and tactile inputs are to be integrated, thus leading to the emergence of the rubber hand illusion if the observer inferred a common cause ($C = 1$) for both sensory inputs. On a given trial, the observer receives a particular measured asynchrony $x_{\text{trial}}$ (simply a number) and infers the category $C$ by computing the posterior probabilities $p\left(C = 1|x_{\text{trial}}\right)$ and $p\left(C = 2|x_{\text{trial}}\right)$. These probabilities are conveniently combined into the log posterior ratio $d$:

$$d = log\left(\frac{p\left(C=1|x_{\text{trial}}\right)}{p\left(C=2|x_{\text{trial}}\right)}\right) \tag{4}$$

The observer would report 'yes, it felt like the rubber hand was my own hand' if $d$ is positive. *Equation (4)* can be written as a sum of the log prior ratio and the log-likelihood ratio:

$$d = log\left(\frac{p_{\text{same}}}{1-p_{\text{same}}}\right) + log\left(\frac{p\left(x_{\text{trial}}|C=1\right)}{p\left(x_{\text{trial}}|C=2\right)}\right) \tag{5}$$

Further evaluation of this expression requires us to calculate two likelihoods. The likelihood of $C = 1$ is

$$p\left(x_{\text{trial}}|C = 1\right) = p\left(x_{\text{trial}}|s = 0\right)$$

$$= N\left(x_{\text{trial}}; 0, \sigma^2\right)$$

where we used *Equations (1) and (3)*. The likelihood of $C = 2$ is

$$p\left(x|C = 2\right) = \int p\left(x_{\text{trial}}|s\right) p\left(s|C = 2\right) ds$$

$$= N\left(x_{\text{trial}}; 0, \sigma^2 + \sigma_s^2\right)$$

where we used *Equations (2) and (3)*. Substituting both likelihoods into *Equation (5)*, we can now calculate $d$:

$$d = log\left(\frac{p_{\text{same}}}{1-p_{\text{same}}}\right) + log\left(\frac{N\left(x_{\text{trial}};0, \sigma^2\right)}{N\left(x_{\text{trial}};0, \sigma^2+\sigma_s^2\right)}\right) \tag{6}$$

$$= log\left(\frac{p_{\text{same}}}{1-p_{\text{same}}}\right) + \frac{1}{2}log\left(\frac{\sigma^2+\sigma_s^2}{\sigma^2}\right) - \frac{x_{\text{trial}}^2}{2}\left(\frac{1}{\sigma^2} - \frac{1}{\sigma^2+\sigma_s^2}\right) \tag{7}$$

As mentioned above, we assume that the observer reports 'yes, the rubber hand felt like my own hand' if $d > 0$. Using *Equation (7)*, we can now rewrite this condition in terms of $x_{\text{trial}}$.

$$\frac{x_{\text{trial}}^2}{2}\left(\frac{1}{\sigma^2} - \frac{1}{\sigma^2+\sigma_s^2}\right) < log\left(\frac{p_{\text{same}}}{1-p_{\text{same}}}\right) + \frac{1}{2}log\left(\frac{\sigma^2+\sigma_s^2}{\sigma^2}\right)$$

$$x_{\text{trial}}^2 < \frac{\sigma^2\left(\sigma^2+\sigma_s^2\right)}{\sigma_s^2}\left(2log\left(\frac{p_{\text{same}}}{1-p_{\text{same}}}\right) + log\left(\frac{\sigma^2+\sigma_s^2}{\sigma^2}\right)\right)$$

Then, we define

$$K = \frac{\sigma^2\left(\sigma^2+\sigma_s^2\right)}{\sigma_s^2}\left(2log\frac{p_{\text{same}}}{1-p_{\text{same}}} + log\frac{\sigma^2+\sigma_s^2}{\sigma^2}\right)$$

If $K < 0$, which can theoretically happen when $p_{\text{same}}$ is very small, then the condition $d > 0$ is never satisfied, regardless of the value of $x_{\text{trial}}$. This corresponds to the (unrealistic) case that it is so a priori improbable that there is a common cause that no amount of sensory evidence can override that belief. If $K < 0$, the condition $d > 0$ is satisfied when this condition is equivalent to

$$\left|x_{\text{trial}}\right| < k$$

where we call $k = \sqrt{K}$ the decision criterion. Notice that $k$ takes into account both $p_{\text{same}}$ and the sensory uncertainty. This concludes our specification of the Bayesian inference performed by our model observer.

## Step 3: response probability

We complete the model by calculating the probability that our model observer responds 'I felt like the rubber hand was my hand' (which we denote by $C=1$) for the visuotactile asynchrony $s_{\text{trial}}$ experimentally presented on a given trial. The first case to consider is $K < 0$. Then,

$$p\left(\hat{C} = 1|s_{\text{trial}}\right) = 0$$

Otherwise,

$$
\begin{aligned}
p\left(\hat{C} = 1|s_{\text{trial}}\right) &= \Pr_{x_{\text{trial}}|s_{\text{trial}}}\left(\left|x_{\text{trial}}\right| < k\right) \\
&= \Phi\left(k; s_{\text{trial}}, \sigma^2\right) - \Phi\left(-k; s_{\text{trial}}, \sigma^2\right) \\
&= \Phi\left(k; s_{\text{trial}}, \sigma^2\right) - \Phi\left(-k; s_{\text{trial}}, \sigma^2\right)
\end{aligned}
$$

where $\Phi$ denotes the cumulative normal distribution. Finally, we introduce a lapse rate, which is the probability of making a random response (which we assume to be yes or no [the rubber hand felt like my hand] with equal probability). Then, the overall response probability becomes

$$p_{\text{with lapse}}\left(\hat{C} = 1|s_{\text{trial}}\right) = 0.5\lambda + (1 - \lambda)\left(\Phi\left(k; s_{\text{trial}}, \sigma^2\right) - \Phi\left(-k; s_{\text{trial}}, \sigma^2\right)\right).$$

It is this outcome probability that we want to fit to our data. Five free parameters need to be fitted: $\theta = \left[p_{\text{same}}, \sigma_0, \sigma_{30}, \sigma_{50}, \lambda\right]$. In the basic model, the source noise $\sigma_s$ is fixed, its value corresponding to the real SD of the asynchronies used in the experiment (348 ms).

## 2. Alternative models

### BCI model with free source noise: BCI*

This model shares the generative model and decision rule of the Bayesian causal inference (BCI) model (*Equation 7*). However, the level of noise impacting the stimulation $\sigma_s$ is considered as a free parameter instead of being fixed. Thus, six parameters need to be fitted: $\theta = \left[p_{\text{same}}, \sigma_0, \sigma_{30}, \sigma_{50}, \sigma_s, \lambda\right]$.

### BCI model with a minimal asynchrony different from 0: BCI_bias

We also designed a model that did not assume that the observer treats an asynchrony of 0 as minimal. In this alternative model, the decision criterion is the same as in the BCI model (*Equation 7*); however, a parameter μ (representing the mean of the distribution of asynchrony) is taken into account when computing the predicted answer in the following step:

$$p_{\text{with lapse}}\left(\hat{C} = 1|s_{\text{trial}}\right) = 0.5\lambda + (1 - \lambda)\left(\Phi\left(k + \mu; s_{\text{trial}}, \sigma^2\right) - \Phi\left(-k + \mu; s_{\text{trial}}, \sigma^2\right)\right)$$

Thus, six parameters need to be fitted: $\theta = \left[p_{\text{same}}, \sigma_0, \sigma_{30}, \sigma_{50}, \mu, \lambda\right]$.

### Fixed-criterion model: FC

This model shares the generative model with the BCI models, but the variations of the level of sensory uncertainty from trial to trial are not taken into account in the decision rule (*Equation 7*). Because $p_{\text{same}}$ remains constant in our experiment, the decision rule is equivalent to reporting 'yes, the rubber hand felt like my hand' if the measured asynchrony is smaller than a constant $k_0$ :

$$\left|x_{\text{trial}}\right| < k_0.$$

Five free parameters need to be fitted: $\theta = \left[k_0, \sigma_0, \sigma_{30}, \sigma_{50}, \lambda\right]$.

Note that if the decisional stage in the FC model is independent of the trial-to-trial sensory uncertainty, the encoding stage is still influenced by the level of sensory noise. Thus, the output of the FC model is the probability of the observer reporting the illusion when presented with a specific asynchrony value $s$:

$$p_{\text{with lapse}}\left(\hat{C} = 1|s_{\text{trial}}\right) = 0.5\lambda + (1 - \lambda)\left(\Phi\left(k_0; s_{\text{trial}}, \sigma^2\right) - \Phi\left(-k_0; s_{\text{trial}}, \sigma^2\right)\right)$$

As in the main BCI model, the additional parameter $\lambda$ reflects the probability of the observer lapsing, i.e., randomly guessing. This equation is a prediction of the observer's response probabilities and can thus be fit to a participant's behavioral responses.

## 3. Model fitting and comparison

### Model fitting

For each model, we want to find the combination of parameters that best describe our data $D$, i.e., the yes/no responses to the presented asynchronies. We use maximum-likelihood estimation to estimate the model parameters, which for a given model, we collectively denote by $\theta$. The likelihood of $\theta$ is the probability of the data $D$ given $\theta$:

$$L(\theta) = p(D|\theta).$$

We next assume that the trials are conditionally independent so that the likelihood becomes a product over trials:

$$L(\theta) = \prod_{\text{trial } t} p\left(\hat{C}_t | s_t,\ \sigma_t,\ \theta\right)$$

where $s_t$ and $\sigma_t$ are the asynchrony and the noise level on the t$^{\text{th}}$ trial, respectively. It is convenient to maximize the logarithm of the likelihood, which is

$$logL(\theta) = \sum_{\text{trial } t} \log p\left(\hat{C}_t | s_t,\ \sigma_t,\ \theta\right) \tag{8}$$

We now switch notation and group trials by noise condition (labeled $i$ and corresponding to the three noise levels) and stimulus condition (labeled $j$ and corresponding to the seven asynchronies). Then, we can compactly denote the observed data by $n_{1ij}$ and $n_{0ij}$, which are the numbers of times the participant reported 'yes' and 'no,' respectively, in the $(i, j)^{th}$ condition. Then, **Equation 8** simplifies to

$$logL(\theta) = \sum_{i,\ j}\left[n_{1ij}\log p\left(\hat{C} = 1|s_j,\ \theta\right) + n_{0ij}\log\left(1 - p\left(\hat{C} = 1|s_j,\ \theta\right)\right)\right]$$

The hard and plausible bounds used in the optimization algorithms can be found in the **Appendix 1—table 1**.

**Appendix 1—table 1.** Bounds used in the optimization algorithms.

| Parameter | Type | Hard bound | Plausible bound |
|---|---|---|---|
| $p_{\text{same}}$ | Probability | (0, 1) | (0.3, 0.7) |
| $\sigma$ | Sensory noise (log) | (−Inf, +Inf) | (−3, 9) |
| $\lambda$ | Lapse | (0, 1) | (eps, 0.2) |
| $k_0$ | Asynchrony (log) | (−Inf, +Inf) | (−3, 9) |

### Parameter recovery

In order to qualitatively assess our fitting process, we performed parameter recovery. We used random sets of parameters $\theta = [p_{\text{same}}, \sigma_0, \sigma_{30},\ \sigma_{50},\ \sigma_s,\ \lambda]$ to generate data from the BCI model, then fitted the BCI model to these simulated data. We then did three assessments: (1) the log likelihoods of the fitted parameters were higher than of the generating parameters Negative log-likelihood: NLL (Minitial)=920 ± 78; NLL (Mrecovered)=812 ± 79 and than of an alternative model NLL (FC)=948 ± 89; (2) the model fits to the simulated data looked excellent (**Appendix 1—figure 1**); (3) the generating parameters were roughly recovered after this procedure. Thus, parameter recovery was successful (**Appendix 1—table 1**).

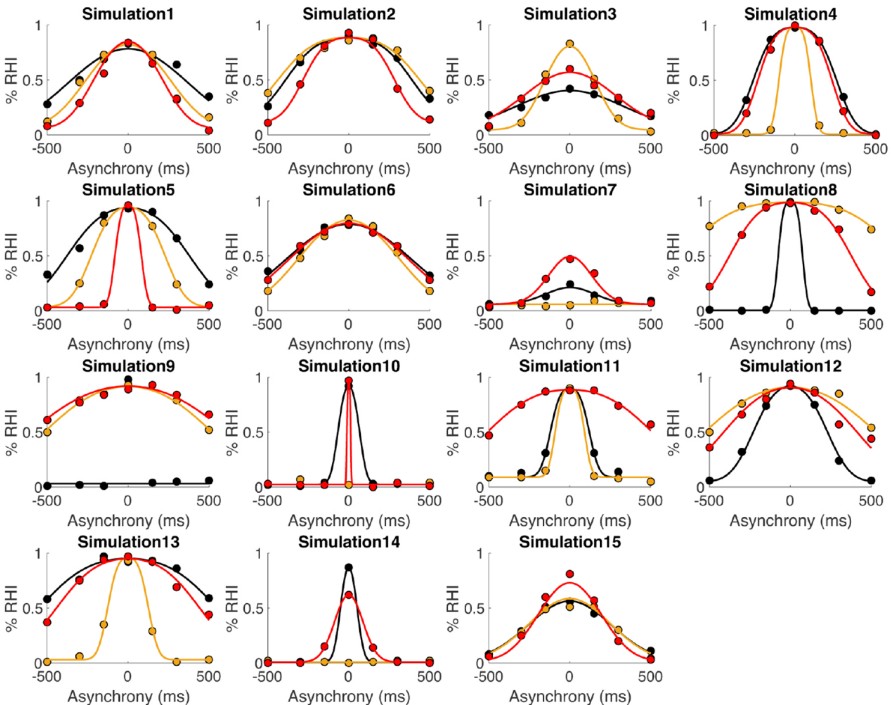

**Appendix 1—figure 1.** The figure displays simulated 'yes (the rubber hand felt like my own hand)' answers as a function of visuotactile asynchrony (dots) and corresponding Bayesian causal inference (BCI) model fit (curves). As in the main text, black, orange, and red correspond to the 0, 30, and 50% noise levels, respectively.

**Appendix 1—table 2.** Initial parameters used to generate the simulations and recovered parameters.

| Participant | Initial | | | | | Recovered | | | | |
|---|---|---|---|---|---|---|---|---|---|---|
| | $p_{same}$ | $\sigma_0$ | $\sigma_{30}$ | $\sigma_{50}$ | $\lambda$ | $p_{same}$ | $\sigma_0$ | $\sigma_{30}$ | $\sigma_{50}$ | $\lambda$ |
| S1 | 0.53 | 246 | 164 | 129 | 0.09 | 0.51 | 264 | 176 | 133 | 0.11 |
| S2 | 0.74 | 183 | 204 | 130 | 0.15 | 0.86 | 152 | 171 | 109 | 0.21 |
| S3 | 0.39 | 281 | 96 | 223 | 0.15 | 0.41 | 313 | 111 | 251 | 0.09 |
| S4 | 0.90 | 97 | 32 | 85 | 0.02 | 0.89 | 94 | 33 | 83 | 0.02 |
| S5 | 0.73 | 185 | 96 | 29 | 0.07 | 0.74 | 176 | 101 | 31 | 0.07 |
| S6 | 0.54 | 238 | 198 | 215 | 0.19 | 0.50 | 294 | 221 | 275 | 0.00 |
| S7 | 0.26 | 138 | 275 | 110 | 0.12 | 0.27 | 151 | 17,803 | 123 | 0.12 |
| S8 | 0.90 | 1 | 240 | 141 | 0.01 | 0.87 | 25 | 256 | 146 | 0.01 |
| S9 | 0.69 | 7 | 265 | 296 | 0.08 | 0.66 | 0 | 274 | 316 | 0.06 |
| S10 | 0.19 | 10 | 142 | 12 | 0.05 | 0.36 | 36 | 4776 | 4 | 0.05 |
| S11 | 0.75 | 50 | 3 | 213 | 0.16 | 0.76 | 47 | 34 | 230 | 0.18 |
| S12 | 0.69 | 108 | 270 | 191 | 0.10 | 0.67 | 111 | 272 | 213 | 0.09 |
| S13 | 0.81 | 224 | 46 | 181 | 0.08 | 0.79 | 237 | 48 | 193 | 0.06 |
| S14 | 0.22 | 22 | 203 | 83 | 0.01 | 0.22 | 34 | 232 | 76 | 0.02 |
| S15 | 0.40 | 215 | 247 | 156 | 0.05 | 0.39 | 232 | 223 | 157 | 0.03 |

## Model comparison

We used the Akaike information criterion (AIC) and the Bayesian information criterion (BIC) to compare models. These quantities are calculated for each model and each participant:

$$AIC = 2n_{par} - 2\log L^*$$

$$BIC = n_{trial}\log n_{par} - 2\log L^*$$

where $L^*$ is the maximized value of the likelihood, $n_{par}$ the number of free parameters, and $n_{trial}$ the number of trials. To compare two models, we calculated the difference in AIC between the two models per participant and summed the differences across the 15 participants. We obtained CIs through bootstrapping: we drew 15 random AIC differences with replacement from the actual participants' AIC differences, then summed those. This procedure was repeated 10,000 times to compute the 95% CI. The same analysis was also conducted for the BIC results.

## 4. Pilot experiment and asynchrony sample adjustment

We chose to match qualitatively difficulty by adjusting the degree of asynchrony in the synchrony judgment task after analyzing the results from 10 participants (six women, 26±4 years) in a pilot study. We only used the zero-noise condition in this pilot and tested identical asynchronies in the two tasks (from −500 ms to +500 ms), otherwise, the procedure was identical to the main experiment. As shown in the table below, in the ±500 ms and the ±300 ms conditions, the number of trials for which the visuotactile stimulation was perceived as synchronous was consistently very low or never happened (zeros) in many cases. This observation suggests that the synchrony task was too easy and that it would not produce behavioral data that would be useful for model fitting or testing the BCI model. Thus, we adjusted the asynchrony conditions in the synchrony task to make this task more challenging and more comparable to the ownership judgment task. Note that we could not change the asynchronies in the ownership task to match the synchrony task because we need the longer 300 ms and 500 ms asynchronies to break the illusion effectively.

**Appendix 1—table 3.** Pilot data.

Number of 'yes' (the visual and tactile stimulation were synchronous) answers in the synchrony judgment task and of 'yes' (the rubber hand felt like it was my own hand) answers in the body ownership task (total number of trials per condition: 12).

| Participant | Synchrony judgment | | | | | | | Ownership judgment | | | | | | |
|---|---|---|---|---|---|---|---|---|---|---|---|---|---|---|
| | −500 | −300 | −150 | 0 | 150 | 300 | 500 | −500 | −300 | −150 | 0 | 150 | 300 | 500 |
| P1 | 0 | 0 | 5 | 11 | 4 | 0 | 0 | 0 | 1 | 6 | 7 | 3 | 4 | 0 |
| P2 | 0 | 0 | 2 | 12 | 3 | 0 | 0 | 9 | 12 | 12 | 12 | 12 | 10 | 0 |
| P3 | 0 | 0 | 1 | 12 | 2 | 0 | 0 | 0 | 2 | 11 | 12 | 12 | 9 | 0 |
| P4 | 0 | 0 | 1 | 12 | 1 | 1 | 0 | 4 | 6 | 9 | 11 | 11 | 11 | 8 |
| P5 | 0 | 1 | 3 | 11 | 1 | 0 | 0 | 0 | 3 | 7 | 12 | 6 | 2 | 0 |
| P6 | 0 | 0 | 0 | 0 | 0 | 0 | 0 | 11 | 12 | 12 | 12 | 11 | 9 | 7 |
| P7 | 0 | 0 | 1 | 9 | 2 | 0 | 0 | 0 | 8 | 12 | 12 | 12 | 2 | 0 |
| P8 | 0 | 0 | 2 | 10 | 0 | 1 | 0 | 5 | 6 | 8 | 11 | 8 | 4 | 2 |
| P9 | 1 | 0 | 1 | 12 | 3 | 0 | 0 | 3 | 7 | 10 | 12 | 3 | 2 | 0 |
| P10 | 0 | 0 | 3 | 12 | 2 | 0 | 0 | 0 | 4 | 10 | 12 | 5 | 2 | 0 |

To assess if this change in asynchrony range between tasks may explain the lower prior probability for a common cause in the synchrony detection task, we applied our extension analysis to the pilot data to test the BCI model on tasks with identical asynchronies. The pilot study did not manipulate the level of sensory noise (only the 0% noise level was included). The *Appendix 1—figure 2* shows the key results regarding the estimated $p_{same}$. The same trend was observed as in the main experiment: the estimated a priori probability for a common cause for synchrony judgment was lower than for body ownership. However, for more than half of our pilot participants, $p_{same}$ for body ownership reaches the extremum ($p_{same}$=1). This ceiling effect probably is because the synchrony task was too easy when using asynchronies of 300 ms and 500 ms as in the ownership task; it lacked challenging stimulation conditions required to assess the participants' perception as a gradual function finely. This observation convinced us further that we needed to make the synchrony judgment task more difficult by reducing the longer asynchronies to obtain high-quality behavioral data that would allow

us to test the subtle effects of sensory noise, compare different models, and compare with the ownership judgment task in a meaningful way. From a more general perspective, different tasks may interact differently with sensory factors, but we argue that such task differences is most likely reflected in a change in prior. Even if our model cannot rule out some task-related influences on sensory processing, our interpretation that the priors are genuinely different between the two tasks is consistent with previous studies that examined the relationship between synchrony perception and body ownership (*Costantini et al., 2016*; *Chancel and Ehrsson, 2020*; *Maselli et al., 2016*; see introduction).

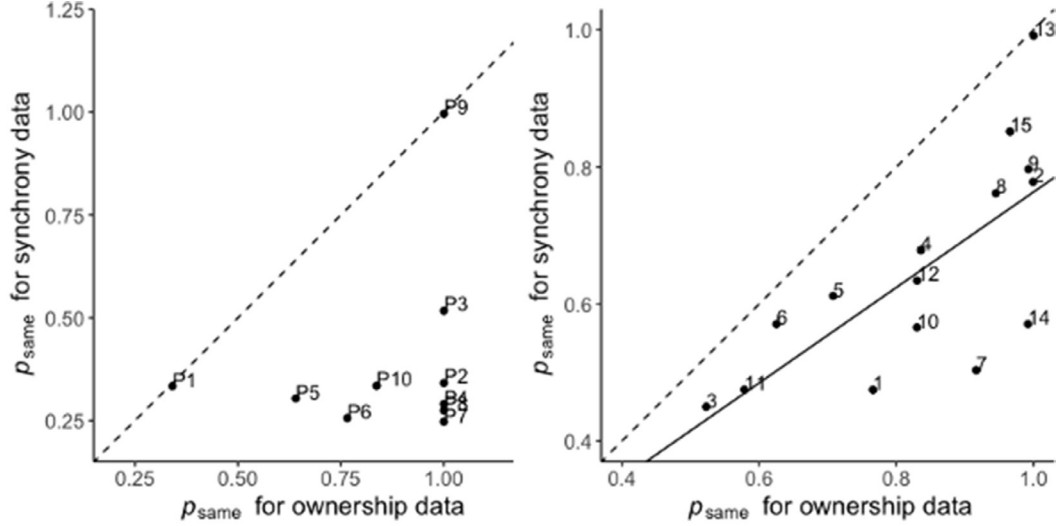

**Appendix 1—figure 2.** Correlation between the prior probability of a common cause p$_{same}$ estimated for the ownership and synchrony tasks in the extension analysis in the pilot study (left) and the main study (right). The solid line represents the linear regression between the two estimates, and the dashed line represents the identity function (x=f[x]).

