## [Editor Report]

In the rubber hand illusion, a rubber hand feels as if being part of one's body when stroked in synchrony with one's own occluded hand. By varying the temporal lags between the strokes to the rubber and to the real hand, and visual noise, the authors suggest that body ownership is governed by an active and probabilistic causal inference process that uses both prior knowledge and sensory uncertainty. The authors argue that probabilistic functions rather than fixed multisensory integration rules governs body ownership, thereby opening new venues for investigating its computational principles. The evidence is compelling, and the findings advanced our fundamental understanding of the computational principles of body ownership.

---

## [Decision Letter]

**Decision letter after peer review:**

Thank you for submitting your article "Uncertainty-based inference of a common cause for body ownership" for consideration by *eLife*. Your article has been reviewed by 3 peer reviewers, and the evaluation has been overseen by a Reviewing Editor and Tamar Makin as the Senior Editor. The following individuals involved in the review of your submission have agreed to reveal their identity: Zoltan Dienes (Reviewer #1); Liping Wang (Reviewer #2); Mate Aller (Reviewer #3).

Essential revisions:

(1) A first reviewer highlights a major interpretation problem based on the 12 s asynchrony, contrasting a domain-general reasoning module with a body ownership one. The reviewer makes suggestions on how to accommodate and clearly refine the claims (e.g. explaining that a main observation is about a shift in criterion).

(2) Consistent with the above issue, a second reviewer highlights several empirical limitations and inconsistencies in the authors' analytical approach (points 2 to 5) with a lack of clear comparisons between the two tasks. These points need to be addressed as listed.

(3) The issue of inter-individual differences is being raised by two reviewers and several suggestions are made to provide a more transparent report of the BCI model and their consistency with authors' interpretations.

*Reviewer #1 (Recommendations for the authors):*

The authors explore the basis of the "rubber hand illusion" in which people come to feel a rubber hand as their own when it is stroked or tapped synchronously with their real hand. They show that when participants are exposed to lags in the tapping of the rubber and real hand, under conditions of different visual noise, people adjust their sense of ownership according to the manipulated variables. The degree of adjustment can be fitted by a Bayesian model, i.e. one in which evidence for ownership is construed as strength of evidence for the stroking being synchronous, and this is used to update the prior probability of there being a common cause for the sight and feeling of stroking.

The paper shows a lot of care in writing and study methodology. Setting up the equipment, the robotic device and the VR, required a lot of effort; the modeling and analysis have been conscientiously performed. The weakness for me is in the background framing of what is going on: Naturally, as is the authors' prerogative, they have taken a particular stance. However, they also say that these data may help distinguish different theories and approaches from their own preferred one. But no argument is made for how the data distinguish their interpretative approach from different ones. This is left as a promissory note. In fact, I think there is a rather different way of viewing the data.

The authors frame their work in terms of a mechanism for establishing bodily ownership. On the other hand, people may infer how feelings of ownership should vary given what is manipulated in the experiment. That is, if asynchrony is manipulated we know people view this as something that is meant to change their feeling of ownership (e.g. Lush 2020 Collabra). That is, the results may be based on general inference rather than any special ownership module. Consistently, reasoning in medical diagnosis and legal decisions can be fit by Bayesian models as well (e.g. Kitzis et al., Journal Math. Psych.; Shengelia and Lagnado, 2021, Frontiers in Psych). That is, the study could be brought under the research programme of showing people often reason in approximately Bayesian ways in all sorts of inference tasks, especially when concrete information is given trial by trial (e.g. Oaksford and Chater).

The results support the bare conclusions I stated, but what those conclusions mean is still up for grabs. I always welcome attempts to model theoretical assumptions, and the approach is more rigorous than many other experiments in that field. Hopefully it will set an example.

Modeling how inferences are formed when presented with different sources of information is an important task, whether or not those inferences reflect the general ability of people to reason, or else the specific processes of a particular model. For example, Fodor claimed that general reasoning was beyond the scope of science, but the fact that across many different inference tasks similar principles arise – roughly though not exactly Bayesian – indicates Fodor was wrong!

Specific comments:

i) The model can be described as Bayesian, but how different is it from a signal detection theory model, with adjustable vs fixed criteria, and a criteria offset for the RHI and asynchrony judgment task? In other words, rather than the generative model being an explicit model for the system, instead different levels of asynchrony simply produce certain levels of evidence for a time difference, as in an STD model. Then set up criteria to judge whether there was or was not a time difference. Adjust criteria according to prior probabilities, as is often done in SDT. That's it. Is this just a verbal rephrasing?

One thing on the framing of the model: Surely a constant offset over several taps is just as good evidence for a common cause no matter whether the offset is 0 or something else? But if the prior probability is specifically that common cause is the same hand is involved (which requires an offset close to 0), surely that prior probability is essentially zero? So how should the assumption of C1 be properly framed?

ii) lines 896-897"…temporally correlated visuotactile signals are a key driving factor behind the emergence of the rubber hand illusion Chancel and Ehrsson,202, …"

Cite findings that need some twisting to fit in with this view, e.g. the finding that imagining a hand is one's own creates SCRs to its being cut; and perhaps more easy to deal with but I think rather telling, no visual input of a hand is needed (Guterstan et al., 2013) and laser lights instead of brushes work just about as well (Durgin et al., 2013) as does stroking the air (Guterstam et al., 2016), making magnetic sensations akin to the magnetic hands suggestion in hypnosis. It seems the simplest explanation is that participants respond to what they perceive as what is meant to be relevant in the study manipulations. Suitable responses are constructed, based on genuine experience or otherwise, in accordance with the needs of the context. The very way the question is framed determines the answer according to principles of general inference (e.g. Lush and Seth, 2022, Nat Coms; Corneille and Lush, https://psyarxiv.com/jqyvx/).

iii) Provide means and SE's for conditions for synchrony judgment tasks.

iv) Discuss the alternative views of how the study could be interpreted as I have indicated above.

*Reviewer #2 (Recommendations for the authors):*

Using rubber hand illusion in humans, the study investigated the computational processes in self-body representation. The authors found that the subjects' behavior can be well captured by the Bayesian causal inference model, which is widely used and well described in multisensory integration. The key point in this study is that the body ownership perception was well predicted by the posterior probability of the visual and tactile signals coming from a common source. Although this notion was investigated before in humans and monkeys, the results are still novel:

1. This study directly measures illusions with the alternative forced-choice report instead of objective behavioral measurements (e.g., proprioceptive drift).

2. The visual sensory uncertainty was changed trial by trial to examine the contribution of sensory uncertainty in multisensory body perception. Combined with the model comparison results, these results support the superiority of a Bayesian model in predicting the emergence of the rubber hand illusion relative to the non-Bayesian model.

3. The authors investigated the asynchrony task in the same experiment to compare the computational processing in the RHI task and visual-tactile synchrony detection task. They found that the computational mechanisms are shared between these tasks, while the prior of common source is different.

In general, the conclusions are well supported, and the findings advanced our understanding of the computational principles of body ownership.

Main comments:

1. One of the critical points in this study is the comparison between the BCI model which takes the sensory uncertainty into account and the non-Bayesian (fixed-criterion) model. Therefore, I suggest the authors show the prediction results of both the BCI and non-Bayesian model in Figure 2 and compare the key hypothesis and the prediction results.

2. This study has two tasks: the ownership task and the asynchrony task. As the temporal disparity is the key factor, the criteria for determining the time disparities are important. The author claim that the time disparities used in the two tasks were determined based on the pilot experiments to maintain an equivalent difficulty level between the two tasks. If I understand correctly, the subjects were asked to report whether the rubber hand felt like my hand or not in the ownership task. Thus, there are no objective criteria of right or wrong. The authors should clarify how they define the difficulty of the two tasks and how to make sure the difficulty is equal. I think this is important because the time disparities in these two tasks were different, and the comparison of Psame in these tasks may be affected by the time disparities. Furthermore, the authors claimed that the ownership and visual-tactile synchrony perception had distinct multisensory processing according to the different Psame in the two tasks. Thus the authors should show further evidence to exclude the difference of Psame results from the chosen time disparities.

3. Related to the question above, the authors found that the same BCI model can reasonably predict the behavioral results in these two tasks with the same parameters (or only different Psame). They claimed that these two tasks shared similar results in multisensory causal inference. While in the following, they argued that there was a distinct multisensory perception in these two tasks because the Psame were different. If these tasks shared the same BCI computational approaches and used the posterior probability of common source to determine ownership and synchrony judgment, what is the difference between them?

4. The extension analysis showed that the Psame values from the two tasks were correlated across subjects. This is very interesting. However, since the uncertainty of timing perception (the sensory uncertainty of perceived time of the tactile stimuli on real hand and fake hand) was taken into account to estimate the posterior probability of common source in this study, the variance across subjects in ownership and synchrony task can only be interpreted by the Psame. In fact, the timing perception was considered as a Gaussian distribution in a modulated version of the BCI model for the agency (temporal binding) (R. Legaspi, 2019) and ownership (Samad, 2015). It will be more persuasive if the authors exclude the possibility that the individual difference of timing uncertainty cannot explain the variance across subjects.

5. Please include the results of single-subject behavior in the asynchrony task. It is helpful to compare the behavioral pattern between these two tasks. The authors compared the Psame between ownership and asynchrony tasks. Still, they did not directly compare the behavioral results (e.g., the reported proportion of elicited rubber hand illusions and the reported proportion of perceived synchrony).

6. The analysis of model fitting seems to lack some details. If I understood it correctly, the authors repeated the fitting procedure 100 times (line 502), then averaged all repeats as the final results of each parameter? It is reported that "the same set of estimated parameters at least 31 times for all participants and models". What does this sentence mean? Can the authors show more details about the repeats of the model result?

7. In Figure 3A, was there an interaction effect between the time disparity and visual noise level?

8. Line 624, the model comparison results suggested that the subjects have the same standard deviation as the true stimulus distribution. I encourage the authors to directly compare the BCI* model predicted uncertainty (σ s) to the true stimulus uncertainty, which will make this conclusion more convincing.

9. How did the authors fit the BCI model to the combined dataset from both ownership and synchrony tasks? What is the cost function when fitting the combined dataset?

10. As shown in the supplementary figures, the variations of ownership between the three visual noise levels varied widely among subjects and the predicted visual sensory sigmas (Appendix 1 – Table 2). The ownership in the three visual noise levels correlated with the individual difference of visual uncertainty?

11. The statements of the supplementary figures are confusing. For example, it is hard to determine which one is the "Supplementary File 1. A" in line 249?

12. Line 1040, it is hard to follow how to arrive at this equation from the previous formulas. Please give some more details and explanations.

*Reviewer #3 (Recommendations for the authors):*

This study investigated the computational mechanisms underlying the rubber hand illusion. Combining a detection-like task with the rubber hand illusion paradigm and Bayesian modelling, the authors show that human behaviour regarding body ownership can be best explained by a model based on Bayesian causal inference which takes into account the trial-by-trial fluctuations in sensory evidence and adjusts its predictions accordingly. This is in contrast with previous models which use a fixed criterion and do not take trial-by-trial fluctuations in sensory evidence into account.

The main goal of the study was to test whether body ownership is governed by a probabilistic process based on Bayesian causal inference (BCI) of a common cause. The secondary aim was to compare the body ownership task with a more traditional multisensory synchrony judgement task within the same probabilistic framework.

The objective and main question of the study is timely and interesting. The authors developed a new version of the rubber hand illusion task in which participants reported their perceived body ownership over the rubber hand on each trial. With the manipulation of visual uncertainty through augmented reality glasses they were able to assess whether trial-by-trial fluctuation in sensory uncertainty affects body ownership – a key prediction of the BCI model.

This behavioural paradigm opens up the intriguing possibility of testing the BCI model for body ownership at a neural level with fMRI or EEG (e.g., as in Rohe and Noppeney (2015, 2016) and Aller and Noppeney (2019)).

I was impressed by the methodological rigour, modelling and statistical methods of the paper. I was especially glad to see the modelling code validated by parameter recovery. This greatly increases one's confidence that good coding practices were followed. It would be even more reassuring if the analysis code were made publicly available.

The data and analyses presented in the paper support the key claims. The results represent a relevant contribution to our understanding of the computational mechanisms of body ownership. The results are adequately discussed in light of a broader body of literature. Figures are well designed and informative.

Main points:

1. Line 298: It is not clear if all 5 locations were stimulated in each 12 s stimulation phase or they were changed only between stimulation phases. Please clarify.

2. Line 331: "The 7 levels of asynchrony appeared with equal frequencies in pseudorandom order". I assume this was also true to the noise conditions, i.e., they also appeared in pseudorandom order with equal frequencies and not e.g., blocked. Could you please make this explicit here?

3. Line 348: Was the pilot study based on an independent sample of participants from the main experiment? Please also include standard demographics data (mean+/-SD age, sex) from the pilot study.

4. Line 406: From the standpoint of understanding the inference process at a high level, the crucial step of how prior probabilities are combined with sensory evidence to compute posterior probabilities is missing from the equations. More precisely it is not exactly missing, but it is buried inside the definition of K (line 416) if I understand correctly. I think it would make it easier for non-experts to follow the thought process if Equation 5 from Supplementary material would be included here.

5. Line 511: There are different formulations of BIC, could you please state explicitly the formula you used to compute it? Please also state the formula for AIC.

6. Line 512: "Badness of fit": Interesting choice of word, I completely understand why it is chosen here, however perhaps I would use "goodness of fit" instead to avoid confusion and for the sake of consistency with the rest of the paper.

7. Figure 4: I think the title could be improved here, e.g., "Model predictions of behavioural results for body ownership" or something similar. Details in the current title (mean +/- sem etc.) could go in the figure legend text.

I am a bit confused about why the shaded overlays from the model fits are shaped as oblique polygons? This depiction hints that there is a continuous increase in the proportion of "yes" answers in the neighbourhood of each noise level. Aren't these model predictions based on a single noise level value?

The mean model predictions are not indicated in the figure only the +/- SEM ranges marked by the shaded areas.

Line 261: Given that participants' right hand was used consistently, I am wondering if it makes any difference if the dominant or non-dominant hand is used to elicit the rubber hand illusion? If this is a potential source of variability it would be useful to include information on the handedness of the participants in the methods section.

Line 314: Please use either stimulation "phase" or "period" consistently across the manuscript.

Line 422: The study by Körding et al., (2007) is about audiovisual spatial localization, not synchrony judgment as is referenced currently. Please correct.

Line 529: Perhaps the better paper to cite here would be Rigoux et al., (2014) as this is an improvement over Stephan et al., (2009) and also this is the paper that introduces the protected exceedance probability which is used here.

Figure 3 and potentially elsewhere: Please overlay individual data points on bar graphs. There is plenty of space to include these on bar graphs and would provide valuable additional information on the distribution of data.

Figure 5A: Please consider increasing the size of markers and their labels for better visibility (i.e., similar size as in panels B and C).

Line 608,611, 639-640, and potentially elsewhere: Please indicate what values are stated in the form XX +/- YY. I assume they represent mean +/- SEM, but this must be indicated consistently throughout the manuscript.

*[Editors’ note: further revisions were suggested prior to acceptance, as described below.]*

Thank you for resubmitting your work entitled "Uncertainty-based inference of a common cause for body ownership" for further consideration by *eLife*. Your revised article has been evaluated by Tamar Makin (Senior Editor) and a Reviewing Editor.

The manuscript has been improved but there are some remaining issues that need to be addressed, as outlined below:

In particular, there was a consensus that the potential contribution of demand characteristics should be discussed, rather than dismissed. We also ask that you discuss the potential utility of a more simple model (signal detection theory). Please see more details below.

1) The authors added a paragraph detailing why the minimised (in their opinion) the contributions of demand characteristics. They argue that the theory that subjects respond to demand characteristics "cannot explain the specifically shaped psychometric curves with respect to the subtle stepwise asynchrony manipulations and the widening of this curve by the noise manipulation as predicted by the causal inference model." We were not fully convinced by this argument and wonder why you would want to categorically rule this possibility out.

Reviewer 1 wrote back: This claim is false. The authors should spell out the alternative theory in terms of subjects using general purpose Bayesian inference to infer what is required. Subjects do not need to know how general purpose inference works; they just need to use it. Does the fact that information was delivered trial by trial over many trials rule out general purpose Bayesian inference? On the contrary, it supports it. Bayesian updating is particularly suited to trial by trial situations (e.g. see general learning and reasoning models by Cristoph Mathys), as illustrated by the author's own model. The authors' model could be a general purpose model of Bayesian inference, shown by its applicability to the asynchrony judgment task. The fact that the number of asynchrony levels may not have been noticed by subjects is likewise irrelevant; subjects do not need to know this. Indeed, the authors point out that the asynchrony judgment task was easier than the RHI task, so that the authors needed to use a smaller asynchrony range for this task than the RHI one. That is, subjects' ability to discriminate asynchronies is shown by the authors' data to be more than enough to allow task-appropriate RHI judgments. (Even if successive differences between asynchrony levels were below a jnd, one would still of course still get a well formed psychophysical function over an appropriate span of asynchronies; so subjects could still use asynchrony in lawful ways as part of general reasoning, i.e. apart from any specific self module.)

(In their cover letter the authors bring up other points that are covered by the fact that subjects do not need to know how e.g. inference and imagination work in order to use them. It is well established that response to imaginative suggestions involves the neurophysiology underlying the corresponding subjective experience (e.g https://psyarxiv.com/4zw6g/ for review); imagining a state of affairs will create appropriate fMRI or SCR responses without the subject knowing how either fMRI or SCRs work.) In sum, none of the arguments raised by the authors actually count against the alternative theory of general inference in response to demand characteristics (followed by imaginative absorption), which remains a simple alternative explanation.

Following a discussion we ask that to address the Reviewer's perspective, you acknowledge the possibility that demand characteristics are contributing to participants performance.

2) "How the a priori probabilities of a common cause under different perceptive contexts are formed remains an open question."

Reviewer 1: One plausible possibility is that in a task emphasizing discrimination (asynchrony task) vs emphasizing integration (RHI) subjects infer different criteria are appropriate; namely in the former, one says "same" less readily.

and

3) "temporally correlated visuotactile signals are a key driving factor behind the emergence of the rubber hand illusion"

Reviewer 1: On the theory that the RH effect is produced by whatever manipulation appears relevant to subjects, of course in this study, where asynchrony was clearly manipulated, asynchrony would come up as relevant. So the question is, are there studies where visual-tactile asynchrony is not manipulated, but something else is, so subjects become responsive to something else? And the answer is yes. Guterstam et al., obtained a clear RH ownership effect and proprioceptive drift for brushes stroking the air i.e. not touching the rubber hand Durgin et al., obtained a RH effect with laser pointers, i.e. no touch involved either. The authors may think the latter effect will not replicate; but potentially challenging results still need to be cited.

Here we do not ask that you provide an extensive literature review. Instead, we simply ask you that you acknowledge in the discussion that task differences might influence participants performance (similar to our request above).

4) On noise effects.

Reviewer 1: If visual noise increases until 100% of pixels are turned white, the ratio of likelihoods for C=1 vs C=2 must go to 1 (as there is no evidence for degree of asynchrony) so the probability of saying "yes" goes to p_same, no matter the actual asynchrony (which by assumption cannot be detected at all in this case). p-same is estimated as.8 in the RHI condition. Yet as noise increases, p(yes) actually increases higher than 0.8 in the -150 to +150 asynchrony range (Figure 2). Could an explanation be given of why noise increases p(yes) other than the apparent explanation I just gave (that p(yes) moves towards p(same) as the relative evidence for the different causal processes reduces)?

The deeper issue is that it seems as visual noise increases, the probability that subjects say the rubber hand is their own increases and becomes less sensitive to asynchrony. In the limit it means if one had very little visual information, one just knew a rubber hand had been placed near you on the table, you would be very likely to say it feels like your own hand (maybe around the level of p-same), just because you felt some stroking on your own hand. But if the reported feeling of the rubber hand were the output of a special self processing system, the prior probability of a rubber hand slapped down on the table being self must be close to 0; and it must remain close to zero even if you felt some stroking of your hand and saw visual noise. But if the tendency to say "it felt like my own hand" was an experience constructed by realizing the paradigm called for this, then a high baseline probability of saying a rubber hand is self could well be high – even in the presence of a lot of visual noise.

Please consider that this effect may bear on the two different explanations.

5) The authors reject an STD model in the cover letter on the grounds subjects would not know where to place their criterion on a trial by trial basis taking into account sensory uncertainty.

Reviewer 1: Why could not subjects attempt to roughly keep p(yes) the same across uncertainties? If the authors say this is asking subjects to keep track of too much, note in the Bayesian model the subjects need an estimate of a variance for each uncertainty to work out the corresponding K. That seems to be asking for even more from subjects. The authors should acknowledge the possibility of a simple STD model and what if anything hangs on using these different modelling frameworks.

We feel that a brief mention of this possibility will benefit the community when considering how to leverage your interesting work in future studies.

6) A few proofing notes that have been picked up by Reviewer 3 (these are not comprehensive, so please read over the manuscript again more carefully):

1. Main points 1 and 3: The changes in response to these points as indicated in the response to reviewers are not exactly incorporated in the main manuscript file. Could you please correct?

2. Main point 4: in the main manuscript file there is an unnecessary '#' symbol at the end of the equation, please remove.

3. Main point 7: the title for figure 2 in the updated manuscript does not match the title indicated in the response to reviewers. I think the latter would be a better choice.

4. Supplements for figures 2, 3, 4: It seems that after re-numbering these figures, the figure legends for their supplement figures have not been updated and they still show the original numbering. Could you please update?

---

## [Author Response]

Essential revisions:(1) A first reviewer highlights a major interpretation problem based on the 12 s asynchrony, contrasting a domain-general reasoning module with a body ownership one. The reviewer makes suggestions on how to accommodate and clearly refine the claims (e.g. explaining that a main observation is about a shift in criterion).

We have considered the first reviewer’s comments very carefully and provided detailed responses to all his comments; we also explicitly discuss these issues in the new version of the manuscript.

However, we disagree with the comment that there is a major interpretational problem with our experimental asynchrony manipulation. We are also puzzled by the recommendation that we need to “refine the claims” and that “the main observation is about a shift in criterion” because all three reviewers stated that our results support our main conclusions, and none of the reviewers claimed that our *main finding* is about a shift in criterion, as far as we can see: reviewer 2 discusses some issues related to the comparisons of the two tasks and inter-individual differences that we replied to, and Reviewer 1 wants us to consider a couple of alternative interpretations. We addressed the latter concern and detailed why we disagree with reviewer 1’s suggestion that demand characteristics and domain-general reasoning is a reasonable alternative explanation for our key findings.

Our main findings are the specifically shaped psychometric curves with respect to the subtle stepwise asynchrony manipulations and the widening of this curve by the noise manipulation as predicted by the causal inference model. We do not see how this finding can be explained by demand characteristics or a domain general cognitive reasoning strategy. In brief, the “computational” hypothesis is hidden from the naïve participants and can probably not be figured out spontaneously by them in the current detection task with over 500 randomized trials. Note further that we analyze and model each individual subject’s illusion detections and we find very good replication of our key modeling results between individual participants. There are also many features of the task design and the experimental procedures that reduce the risk of demand characteristics, e.g., use of robots and the experimenter being blind to the noise manipulation. Critically, the shortest asynchrony we used (150 ms) was short enough that most participants did not perceive it reliably as asynchronous, indeed previous work in the literature identified the crossmodal threshold to detect visuotactile asynchrony around 200 ms, which was confirmed by the analysis of our own asynchrony detection task data showing that participants did not detect the +/- 150 ms asynchrony above chance level, and the participants did not know how many different asynchrony levels were tested (as revealed in post-experiment interviews). Another crucial point is that no feedback was given to the participants about task performance, so they could not learn to map their responses to the different experimental manipulations. Cognitive bias might, of course, influence our data as in any perceptual decision paradigm, but our critical argument is that in the current study, such effects will most likely affect the conditions globally – across conditions – and very unlikely to explain specific changes in the psychometric curves that fitted our causal inference model’s predictions.

We also disagree with some of reviewer 1’s more general theoretical comments about the rubber hand illusion, where he seems to imply that the illusion might be nothing more than a combination of demand characteristics and domain-general reasoning, perhaps supplemented perhaps by hypnotic suggestibility. Such strong claims are not supported by the previous rubber hand illusion literature or the current study’s results. We have criticized Peter Lush’s and reviewer 1’s controversial claims about the rubber hand illusion in other publications (Ehrsson et al., 2022 Nature Communications; Slater and Ehrsson. 2022 Frontiers in Human Neuroscience), and in the current response letter we revisit and further clarify some of this debate with respect to the current study’s specific findings and present results from additional analyses. We are also attaching materials from two other ongoing studies that further support the current study’s main conclusion. In one model-based fMRI study, using the current rubber hand illusion detection task, we showed that the hand ownership detection decisions are associated with increased activity in specific multisensory brain areas. And in one behavioral study, we used signal detection theory in a rubber hand illusion discrimination task which supports the perceptual nature of the decision processes.

(2) Consistent with the above issue, a second reviewer highlights several empirical limitations and inconsistencies in the authors' analytical approach (points 2 to 5) with a lack of clear comparisons between the two tasks. These points need to be addressed as listed.

We have addressed all the empirical issues raised by the second reviewer in our point-by-point responses. Thanks to these constructive remarks, we think our analytical approach is now more clearly presented. We also justified in more detail our computational approach and explained the strengths and limitations of comparing the two tasks at the computational level.

(3) The issue of inter-individual differences is being raised by two reviewers and several suggestions are made to provide a more transparent report of the BCI model and their consistency with authors' interpretations.

We addressed all the concerns of reviewers related to inter-individual differences and used many of their recommendations to make changes in the manuscript to make the reporting of the models and the consistency of the result with respect to our conclusions more transparent. We also present several additional analyses and figures in the point-to-point response letter to further clarify these points. We hope that all our results regarding the BCI model and the issues related to inter-individual differences are now presented clearly and transparently.

We thank the reviewers and the editor for the valuable feedback and exciting discussions, and we are confident that the manuscript has been improved by taking into account all various concerns, suggestions, and positive feedback.

Reviewer #1 (Recommendations for the authors):The authors explore the basis of the "rubber hand illusion" in which people come to feel a rubber hand as their own when it is stroked or tapped synchronously with their real hand. They show that when participants are exposed to lags in the tapping of the rubber and real hand, under conditions of different visual noise, people adjust their sense of ownership according to the manipulated variables. The degree of adjustment can be fitted by a Bayesian model, i.e. one in which evidence for ownership is construed as strength of evidence for the stroking being synchronous, and this is used to update the prior probability of there being a common cause for the sight and feeling of stroking.The paper shows a lot of care in writing and study methodology. Setting up the equipment, the robotic device and the VR, required a lot of effort; the modeling and analysis have been conscientiously performed. The weakness for me is in the background framing of what is going on: Naturally, as is the authors' prerogative, they have taken a particular stance. However, they also say that these data may help distinguish different theories and approaches from their own preferred one. But no argument is made for how the data distinguish their interpretative approach from different ones. This is left as a promissory note. In fact, I think there is a rather different way of viewing the data.The authors frame their work in terms of a mechanism for establishing bodily ownership. On the other hand, people may infer how feelings of ownership should vary given what is manipulated in the experiment. That is, if asynchrony is manipulated we know people view this as something that is meant to change their feeling of ownership (e.g. Lush 2020 Collabra). That is, the results may be based on general inference rather than any special ownership module. Consistently, reasoning in medical diagnosis and legal decisions can be fit by Bayesian models as well (e.g. Kitzis et al., Journal Math. Psych.; Shengelia and Lagnado, 2021, Frontiers in Psych). That is, the study could be brought under the research programme of showing people often reason in approximately Bayesian ways in all sorts of inference tasks, especially when concrete information is given trial by trial (e.g. Oaksford and Chater).The results support the bare conclusions I stated, but what those conclusions mean is still up for grabs. I always welcome attempts to model theoretical assumptions, and the approach is more rigorous than many other experiments in that field. Hopefully it will set an example.Modeling how inferences are formed when presented with different sources of information is an important task, whether or not those inferences reflect the general ability of people to reason, or else the specific processes of a particular model. For example, Fodor claimed that general reasoning was beyond the scope of science, but the fact that across many different inference tasks similar principles arise – roughly though not exactly Bayesian – indicates Fodor was wrong!

We thank the reviewer for his positive remarks on the methods, modeling results, and the writing of our paper. We are more than happy to discuss the alternative ways of interpreting the data that the reviewer is bringing up in this report. We agree with the reviewer that we conducted our study to test a particular hypothesis and computational model about the rubber hand illusion, and that our study was not designed to try to distinguish between different theories. So in the manuscript, we will stay focused on the conclusions that are directly supported by our results and avoid unnecessary theoretical speculation.

The reviewer’s central critical point is that our results might stem from ‘demand characteristics’, i.e., that participants may infer how feelings of ownership should vary given the experimental manipulations and then use a general reasoning strategy to generate the behavioral responses to meet these expectancies. In other words, rather than performing perceptual decisions based on a genuine illusion experience, the participants may simply be reporting as they think the researchers want them to report. In our response below, we will first analyze this concern in detail, and we will argue that it is very unlikely that demand characteristics and a general reasoning strategy can explain our findings. Next, we will consider some of the more general theoretical points raised by the reviewer, including the recent studies by Lush and colleagues (Lush, 2020; Lush et al., 2020). Here we will argue that the current study’s main findings are well-protected against the kind of cognitive effects reported in Lush’s articles. Despite some theoretical disagreements about rubber hand illusion (Ehrsson et al., 2022; Slater and Ehrsson, 2022), we think our positions converge on the importance of developing new methods to extend existing computational frameworks to several domains of human cognition and perception.

Demand characteristics and a general cognitive reasoning strategy

The reviewer’s central concern is that the current data emerge from a general cognitive inference process driven by cognitive expectancies that the participants develop about the different conditions. But what would these expectations be, more precisely? We used seven steps of visuotactile asynchrony (0 ms, +/- 150 ms, +/- 300 ms and +/- 500ms) and these differed in small temporal intervals. This means that it is difficult for the participants to separate them and keep track of which stimuli that belongs to which trial type among the hundreds of fully randomized trials in our psychophysics experiments. Critically, our hypotheses are “hidden” and correspond to specific psychometric curves that change in an unintuitive way by the noise manipulation. Thus, it is very difficult for the participants to develop meaningful understanding about what the study is supposed to demonstrate and what they should feel on any given trial.

Importantly, even asynchrony as short as 150 ms that leads to significantly different detection of the rubber hand illusion (- 150 ms versus 0 ms conditions: t = -3.31, df = 14, p = 0.0052, 95% CI = [-2.75; -0.59]; + 150 ms versus 0 ms conditions: t = -3.251, df = 14, p = 0.0059, 95% CI = [-3.65; -0.75]). However, the perceptual threshold for visuotactile asynchrony is above 200 ms (e.g., 211 ± 59.9 ms (mean ± SD) in Costantini et al., 2016; 302 ± 35 ms (mean ± SD) in Shimada et al., 2014; in our own synchrony detection task, participants did not detect the +/-150 ms asynchrony above chance level: 50% detection threshold = 179 +/- 47 ms (mean ± SD)). Thus, in the +/- 150 ms trials the participants are exposed to very subtle manipulations of asynchrony that most participants will not reliably perceive as asynchronous. Consequently, in our view it is implausible that participants form different expectations for different asynchrony trials that they do not even experience as different. Note further that the participants were never informed about how many levels of asynchrony that was used in the current study or told anything about the temporal intervals. Indeed, informal interviews at the end of the experimental sessions suggest the participants never realized that seven levels of asynchrony were used (most participants guessed three or four). So, we think it is very unlikely that they could develop the kind of precise cognitive expectations that would be required in order to even be able to generate the hypothesized curves of behavioral responses based using cognitive reasoning strategy.

The concern with demand characteristics becomes even more implausible when the visual noise conditions are taken into consideration. The causal inference model predicts that increasing sensory uncertainty by increasing visual noise should lead to a specific widening of the psychometric curves so that greater asynchronies are tolerated in the illusion; and the data fit the causal inference model’s predictions well (and better than a fixed-criteria model). We think that it is very implausible that the naïve participants in our study could figure out this hidden computational hypothesis by themselves. In informal interviews after the experiments, some participants spontaneously reported that they thought the noisier visual information should degrade the rubber hand illusion, but most participants had no idea what the noise was supposed to do. Even experts in our field rarely correctly predicted the impact of the visual noise: this work has been presented in two conferences (*Body Representation Network – 2021; ASSC24 – 2021*) and during two invited seminars (*LICÉA – Paris Nanterre, 2022; LPNC – Grenoble Alpes University, 2022*); when academic colleagues were first introduced to the task at these meetings most of them expected the visual noise to work against the illusion, i.e., lead to “a weaker illusion”, which is opposite of our results and does not capture the graded effect. Furthermore, it is critical to again note that participants *never receive any feedback* about their behavior; they are never told if they are right or wrong. Thus, with the trial-to-trial random variation in sensory noise and visuotactile asynchrony in the absence of feedback, it is impossible for the participants to learn to map their answers onto the specific predictions of the model.

It is also relevant to point out here that the experimenter was always out-of-sight of the participant and that all visuotactile stimulation was produced by two robots. Thus, the participant could not pick up putative subtle social cues from the experimenter that could potentially serve as an implicit source of information about the performance or the hypotheses. Moreover, the experimenter was blind to the visual noise condition.

If we look a bit more closely at the psychophysics task itself, please note that it is based on a huge number of trials presented in a fully randomized order, and with relatively little time after each trial to give the classification response. This experimental design makes the use of a reasoning strategy difficult. It is improbable the participants could keep track of the 12 randomized repetitions of the 42 conditions tested in the current experiment, which sums up to over 500 trials in a single experiment, which is required in order to generate response patterns based on a cognitive reasoning strategy to “simulate” the causal inference model’s fit. In addition, such a cognitive strategy based on domain-general reasoning would put great demands on working memory, long-term memory, analytical reasoning skills, as well as knowledge about probabilistic principles of perception. In addition to thinking about each trial, they would simultaneously have to remember the approximate number of yes/no responses for each asynchrony level in their previous history of responses and how the frequencies of responses on each of 7 different asynchrony conditions and three levels of noise should change according to Bayesian probabilistic principals of causal inference. We are not sure that this is even theoretically possible with the current paradigm and recall that our participants were naive participants recruited from outside the department who had nothing to gain from even attempting to perform such an unpleasantly demanding cognitive task. It is much easier for the participants just to follow the instruction and base their response on each trial on the rubber hand illusion feeling.

In addition to the above arguments, also note that we analyze and model our data on each individual participant individually. Thus, even if we assume that there were a few participants who could figure out the specific hypotheses and that also had the motivation, determination, theoretical knowledge, and cognitive capacity to simulate behavior to meet our computational predictions *against* task instruction (a super version of “the good subject”), it cannot explain that we observed good model fits in the large majority of participants (pseudoR2 above 0.60 for 11 out 15 participants). Importantly, when comparing our different models, we added to our AIC/BIC analysis a protected exceedance probability analysis (Rigoux et al., 2014). This type of analysis includes participant has a random factor, i.e., it takes into account the possibility that the goodness of fit of one model would be mostly due to a few “perfect” subjects while other participants followed another model or a random distribution, by computing the posterior probability that one model occurs more frequently than any other model in the set, above and beyond chance. The results of this type of analyses highlighted the relevance and dominance of our main model across our whole sample. In addition to these specific results, most participants in experimental behavioral studies like the current one simply want to follow task instructions and “the good subject” seems to be relatively rare: in reviewing the classic literature on demand characteristics in social psychology experiments, Weber and Cook (1972) concluded that evidence for demand characteristics in experimental psychological studies was weak and ambiguous in most cases and convincing evidence for instances where “the good subject” explained the results were lacking. The current analysis approach based on individual participants’ perception also means that our results cannot stem from weak demand characteristics or cognitive reasoning effects occurring in individual subjects that are then aggregated into a “false” group-level effect.

So, in sum, we do not see how expectations, demand characteristics, and high-level domain-general reasoning can explain the current study’s main results or constitute a plausible alternative interpretation for the conclusion that body ownership in the rubber hand illusion is governed by Bayesian causal inference of sensory evidence. Although cognitive bias might lead to “global” changes in decision criteria, as we will discuss further below, such possible effects cannot, in our view, explain the specific shapes of the psychometrics curves related to the subtle asynchrony manipulation or the changes in these curves occurring when sensory uncertainty is manipulated.

The fact that reasoning at a “high cognitive level” (e.g., medical and judicial decisions) can be described as a near-Bayesian general inferential process does not exclude the existence of specific perceptual modules. Similar probabilistic computational decision principles may govern cognition and perception (e.g., Shams and Beierholm, 2022). However, this does not mean that if a perceptual decision task follows Bayesian probabilistic principles, it must be based on high-level cognition. The idea that automatic perceptual decisions related to the multisensory binding problem are implemented in the brain’s perceptual systems and operate according to Bayesian causal inference principles is rather well established in the literature on multisensory perception (Aller and Noppeney, 2019; Körding et al., 2007; Rohe et al., 2019; Rohe and Noppeney, 2016, 2015; Shams and Beierholm, 2010). In the current study, we extend this principle that is relevant for illusory and non-illusory audio-visual perceptual effects to the case of a multisensory bodily illusion.

Lush’s studies

The reviewer states that “if asynchrony is manipulated we know people view this as something that is meant to change their feeling of ownership” and cite Lush 2020 (Collabra). We disagree with this statement because we think it is too strong and too generalizing, and it is not clear how Lush’s findings relate to the current study’s results since Lush 2020 did not test the rubber hand illusion. In fact, we still know quite little about how conceptual knowledge about the rubber hand illusion influences subjective ratings of the illusion in actual experiments when naïve participants are instructed to report on their illusion feeling.

Lush (2020) has several limitations (Slater and Ehrsson, 2022), which we need to point out because it is relevant for the current discussion. First, Lush 2020 never tested the participants on the actual rubber hand illusion, so we do not know how the participant’s knowledge and expectations might have influenced their subjective ratings of illusion strength in an actual rubber hand illusion experiment. Second, the information and instructions provided to the participants in Lush 2020 are different from a typical rubber hand illusion experiment. In Lush 2020 the participants, who were psychology undergraduates, studied extensive written and video material about the rubber hand procedures (on their own laptops), including detailed information about the synchronous and asynchronous stimulation conditions, videos of the hidden real hand receiving the different types of tactile stimulation, and explicit information that the purpose of the rubber hand illusion was to “generate changes in experience”. The participants' task was to try to guess what people experience in the rubber hand illusion when shown the questionnaire statements that are used to quantify the illusion. In typical rubber hand illusion experiments, naïve participants are given minimal information about the procedures, and they are just instructed to report what they experience. Thus, these differences in task instruction (metacognitive evaluation versus rate a perceptual sensation) and the differences in the information about the rubber hand illusion may have created demand characteristics and expectations that are specific to Lush’s study and not representative of rubber hand illusion experiments in general. In other words, regardless of what the psychology undergraduates may or may not have thought about how the illusion works, this may not have substantially influenced their ratings of the illusion in a real rubber hand illusion experiment. Third, when the participants in Lush 2020 are given a rubber hand illusion questionnaire and asked to fill them out according to their expectancies, they report affirmative mean illusion ratings in both the synchronous and the asynchronous conditions. Noteworthy, this is qualitatively different from real rubber hand illusion experiments, where the illusion is typically clearly rejected in the asynchronous condition (negative scores in the order of mean -1 to -2; e.g., Kalckert and Ehrsson, 2014; Reader et al., 2021), suggesting that the participants’ guesses about the RHI were vague when it come to the specific effect of the asynchrony manipulation.

Interestingly, in Lush et al., (2020), the participants' expectations about what they were expected to experience in the synchronous and asynchronous conditions were registered before the rubber hand illusion (together with their trait suggestibility), and then the rubber hand illusion was tested and quantified with questionnaires (and proprioceptive drift). Importantly, expectations about what they expected to feel in the synchronous and the asynchronous conditions only had a very small effect on the questionnaire ratings from the actual rubber hand illusion experiment, only affecting the synchronous condition’s ratings a little and apparently not at all affecting the asynchronous condition (Slater and Ehrsson, 2022). Critically, the contribution was so small that it could effectively be ignored compared to the contribution of the visuotactile synchrony-asynchrony manipulation and trait suggestibility (Slater and Ehrsson, 2020). Furthermore, the contribution of visuotactile synchrony-asynchrony was two-to-three times more important than expectations and trait suggestibility combined. Thus, potential expectancies are unlikely to explain the current study’s main behavioural results. In addition to this, Lush et al., 2020, as well as recently reanalyzes of the same dataset (Ehrsson et al., 2022; Slater and Ehrsson, 2022), clearly show that the differences in illusion ratings between the synchronous and asynchronous conditions are unrelated to trait suggestibility (Lush et al., 2020; Ehrsson et al., 2022; Slater and Ehrsson, 2022). Thus, possible differences in trait suggestibility cannot explain the current study's main findings since these are based on asynchrony manipulation and differences between conditions, and such difference measures do not correlate with trait suggestibility (Lush et al., 2020; Ehrsson et al., 2022; Slater and Ehrsson, 2022). We think it was relevant to clarify this point here because suggestibility is a key concept in Lush’s articles.

To us, Lush and colleagues’ work are interesting because they inform us about *individual differences* in how trait suggestibility and cognitive expectancies may influence subjective illusion reports. There is no incompatibility between the arguments that the rubber hand illusion is a bodily illusion driven to a large extent by multisensory correlations, and at the same time, subjective rubber hand illusion reports can be modulated by top-down cognitive processes (Slater and Ehrsson, 2022) and show individual differences in illusion reports that are modulated by cognition. In the current study, we focus on the former multisensory perceptual aspects of the illusion at the level of individual subjects. We did not test Lush and colleagues’ hypotheses about demand characteristics, although the current computational modeling and psychophysics approach could be used for this purpose in future studies.

Cognitive bias

To be clear, we are not arguing that participants’ cognitive expectations cannot influence rubber hand illusion reports at all, or that the current dataset is completely free from cognitive biases, or that there could be no changes in decision criteria between some of the conditions in some of the participants. Our main argument is that such effects can probably not explain our main computational modeling results. It is possible that some participants adopt a more conservative decision criterion than other individuals in the detection task, while others use a more liberal criterion. Such differences in decision criteria could stem from many different postperceptual processes, including individual differences in trait suggestibility; but also differences in perceptual bias that can capture key perceptual aspects of perceptual illusions (Morgan et al., 1990). The effect of biases (cognitive or perceptual) could be accounted for by changes in the prior in our models. As such, a cognitive bias effect related to demand characteristics and expectations would most likely manifest itself as a global influence on all or some of conditions (Lush et al., 2020; Slater and Ehrsson, 2022), and not explain the specific patterns of trial-to-trial results we observed that fit with the causal inference model’s predictions.

Furthermore, a hypothetical change in decision criterion related to perceived synchrony (e.g., 0 ms and +/- 150 ms trials) or asynchrony (e.g., +/- 300 ms and +/- 500 ms) would not lead to the specific changes predicted by the causal inference model. Actually, such possible effects would correspond to a “fixed criteria” strategy rather than a Bayesian causal inference one. But the current study argues against such a fixed criteria strategy in our data because we formally compared the fit of the Bayesian causal inference model to our Fixed-Criterion (FC) model, and the causal inference model outperformed the FC model. In addition, and as already said, we have used a randomized trial design where all asynchrony conditions and the noise conditions are presented in random trial order. Such a design is considered to reduce the risk of adaptation and changes in decision criteria compared to designs where different conditions are presented in separate runs, and this was something we took into account when designing the study.

In conclusion, and after considering all the reviewer concerns, we still think that our conclusion that the rubber hand illusion is governed by Bayesian causal inference based on sensory evidence and sensory uncertainty is well-supported by the results. Our findings support this conclusion, and our results and conclusions are in line with our hypothesis and theoretical framework of the rubber hand as a multisensory bodily illusion, as well as the broader empirical and theoretical literature on causal inference in multisensory perception. We think reviewer 1’s suggestion that demand characteristic and a domain general cognitive strategy is an unlikely explanation for the current study’s main modeling findings, but we have made changes in the revised version of the manuscript to explicitly discuss this issue.

Specific comments:(i) The model can be described as Bayesian, but how different is it from a signal detection theory model, with adjustable vs fixed criteria, and a criteria offset for the RHI and asynchrony judgment task? In other words, rather than the generative model being an explicit model for the system, instead different levels of asynchrony simply produce certain levels of evidence for a time difference, as in an STD model. Then set up criteria to judge whether there was or was not a time difference. Adjust criteria according to prior probabilities, as is often done in SDT. That's it. Is this just a verbal rephrasing?

The reviewer points toward an interesting discussion on the differences and potential overlap between two different computational frameworks and correctly points out that SDT and BCI sometimes lead to the same predictions at the behavioral level. However, we would argue that the scope of these two frameworks diverges and that the BCI framework is more relevant for the current study’s aims.

The SDT allows the estimation of decisional thresholds given one or several sensory inputs. From this standpoint, SDT can describe the thresholds used by the participants but does not explain how they are established. Under this approach, one could speculate that thresholds are quantitatively adjusted in two different tasks (e.g., under two different priors), which would require complexifying the initial SDT assumptions. However, it would be unrealistic to assume that the participants learn to adjust their SDT threshold from trial-to-trial, taking into account the level of sensory uncertainty.

On the contrary, the BCI framework is designed to be a more comprehensive, process-based approach: this framework explains how the perceptual thresholds are computed. This is not just “verbal rephrasing” but a fundamentally different approach to examine hypotheses about underlying computational principles. As a result, our BCI model efficiently captures the observed behavioral effect of visual noise on body ownership perception, including the variations in rubber hand illusion detection on a trial-to-trial basis.

Both approaches could be used in future psychophysics studies on bodily illusions, but with different purposes. For example, we are currently working on an SDT rubber hand illusion study based on a 2-AFC hand ownership task that we have developed in our lab (Chancel et al., 2021; Chancel and Ehrsson, 2020). Among this study's many interesting observations, relevant to mention here is that hand ownership sensitivity (d’) is significantly above zero for stimulation asynchrony of 50 ms, 100 ms and 200 ms, i.e., small asynchronies in a similar range as used in the current experiment. Note also significant ownership sensitivity for delays of 50 ms, which are too brief to be consciously perceived and therefore should not produce cognitive expectancies (Lanfranco et al., 2022 in preparation). Since hand ownership sensitivity measures (d’) and bias-free estimates of the participants’ rubber hand illusion discrimination behavior, this finding is in line with the current investigation’s Bayesian modeling results and interpretation.

One thing on the framing of the model: Surely a constant offset over several taps is just as good evidence for a common cause no matter whether the offset is 0 or something else? But if the prior probability is specifically that common cause is the same hand is involved (which requires an offset close to 0), surely that prior probability is essentially zero? So how should the assumption of C1 be properly framed?

The reviewer is correct that the temporal correlation between the visual and tactile stimulation in our experiment could promote multisensory integration due to the constant offset over several taps regardless of the visuotactile asynchrony (Parise and Ernst, 2016). Yet, the visuotactile delay also matters for multisensory perception (as discussed in Parise and Ernst, 2016), and that is the parameter that we manipulated in the current study keeping all other factors constant (except for the noise manipulation). Thus, both the correlations and the asynchrony matter, and the greater the asynchrony, the less information in favor of the rubber hand as being one’s own despite the visuotactile correlation.

Importantly, the inference of a common cause in models such as the one we are using takes into account the discrepancy between the sensory signals; thus, the magnitude of the offset matters. Especially since in the model we use, a common cause for vision and touch in our model (C = 1) means the same hand is involved as the source of visual and tactile inputs. And indeed, it requires the offset to be close to 0; that’s why we assume that the asynchrony s is always zero when C=1 (see appendix 1 for a detailed presentation, page 36 lines 1056 to 1059). A similar phrasing of the different causal scenarios was already proposed by Samad et al., (2015). In the future, more complex models could be developed that incorporate both temporal correlations and visuotactile delays, but this is beyond the aim of the current study.

(ii) lines 896-897"…temporally correlated visuotactile signals are a key driving factor behind the emergence of the rubber hand illusion Chancel and Ehrsson,202, …"Cite findings that need some twisting to fit in with this view, e.g. the finding that imagining a hand is one's own creates SCRs to its being cut; and perhaps more easy to deal with but I think rather telling, no visual input of a hand is needed (Guterstan et al., 2013) and laser lights instead of brushes work just about as well (Durgin et al., 2013) as does stroking the air (Guterstam et al., 2016), making magnetic sensations akin to the magnetic hands suggestion in hypnosis. It seems the simplest explanation is that participants respond to what they perceive as what is meant to be relevant in the study manipulations. Suitable responses are constructed, based on genuine experience or otherwise, in accordance with the needs of the context. The very way the question is framed determines the answer according to principles of general inference (e.g. Lush and Seth, 2022, Nat Coms; Corneille and Lush, https://psyarxiv.com/jqyvx/).

The statement that “temporally correlated visuotactile signals are an important factor driving the emergence of the rubber hand illusion” is supported by a very large previous literature, as well as the current study’s findings. Note that we are not saying this is the only factor that drives the illusion (e.g., spatial congruence factors also contribute), only that it is an important factor.

We cited Chancel and Ehrsson (2020) because it is a highly relevant rubber hand illusion study given the similarities in paradigm and the manipulation of fine-grained temporal asynchronies of visual and tactile signals. Chancel and Ehrsson (2020) used seven levels of visuotactile asynchrony shorter than or equal to 200 ms, which means that the different visuotactile asynchronies were not clearly perceived by the participants and yet the collected data showed a significant relationship between the degree of visuotactile asynchrony and illusory rubber hand illusion as quantified in a discrimination task. Moreover, the illusory hand ownership discriminations were influenced by a manipulation of the distance between the participants’ hand and the rubber hand (5 cm change in lateral axis) in line with the spatial congruence principle of multisensory integration even though the participants did not notice this distance manipulation (which occurred between runs and out-of-sight of the participants, and confirmed in post-experimental interviews), and thus very unlikely to form high level cognitive expectations about the specific hypothesis related to this orthogonal and small spatial manipulation. Our statement that temporally correlated visuotactile signals are a key driving factor behind the rubber hand illusion comes from these observations, but as we said, also from a very large previous literature (e.g., Blanke et al., 2015; Botvinick and Cohen, 1998; Ehrsson, 2020; Ehrsson et al., 2004; Kilteni et al., 2015; Slater and Ehrsson, 2022; Tsakiris, 2010; Tsakiris and Haggard, 2005).

The statement under discussion is also supported by our own data. To illustrate this further we ran a new posthoc analysis. A mixed-effect logistic regression with participant as random effect confirms a significant effect of asynchrony (p <2e-16), as expected (see Author response image 1). A clear effect of asynchrony is seen in every participant, which is in line with our claim that this is key factor that drives the emergence of the rubber hand illusion.

**Author response image 1. sa2fig1:** Mixed-effect logistic regression with participant as random effect. Dots represents individual responses, the curves are the regression fit, the shaded areas the 95% confidence interval.

We could very well have cited Guterstam et al., (2013) here because it is a well-controlled study with many experiments that support the conclusion that temporally correlated visuotactile signals play a critical role in bodily illusions such as the rubber hand illusion. In this study, Arvid Guterstam presented evidence from 10 separate experiments conducted on different groups of naive participants; 234 subjects in total (one explorative pilot experiment and nine experiments that would nowadays be called hypothesis testing experiments). Nine of the experiments included synchronous and asynchronous conditions, and all findings support an important role for synchronous visuotactile correlations in driving this version of the rubber hand illusion. Several additional control conditions were included in this study, such as various spatial manipulations related to the spatial rule of multisensory integration and a control condition involving a block of wood that eliminates the illusion due to shape incongruence and top-down factors. The outcome measures were questionnaire ratings, increases in SCR triggered by physical threats towards the illusory hand, changes in hand position sense towards the illusory hand (proprioceptive drift), as well as one functional magnetic resonance imaging experiment. Critically the synchronous illusion condition leads to greater illusion questionnaire ratings, greater threat-evoked SCR, greater proprioceptive drift, and greater BOLD signals in key areas related to multisensory integration of bodily signals (such as the posterior parietal cortex, premotor cortex, and cerebellum) than the asynchronous condition and the other control conditions. Thus, the findings from Guterstam et al., (2013) support the sentence under discussion above and are in line with the current study’s findings and conclusions.

Guterstam et al., (2016) examine a somewhat different perceptual phenomenon of a perceived causal connection (similar to a magnetic force field) between visual and tactile events close to one’s own hand in peripersonal space. Again, this study is well controlled and contains many experiments, controls and measures, and the study finds that synchronous visuotactile stimulation and a limited spatial extent of peripersonal space around the hand are critical factors for this visuotactile illusion effect to arise. This study’s findings are in line with the rubber hand illusion literature and the literature on multisensory integration in peripersonal space. The “similarity” to “magnetic hands suggestions” that the reviewer mentions makes no sense to us. Such hypnotic suggestions do not obey temporal and spatial rules of multisensory integration and do not depend on correlated visuotactile signals. Also, the procedures and instructions are very different.

We can’t say so much about the Durgin et al., 2007 study because we have not tried to replicate it. However, in an ongoing study, we have used a control condition where participants just look at a rubber hand while we shine a laser light on it without any tactile or other somatosensory stimulation delivered to the hidden real hand. We find low questionnaire ratings in this condition, significantly lower than typically observed in rubber hand illusion conditions with synchronous visuotactile stimulation and more similar to the control condition when participants just look at a rubber hand without any stimulation. Synchronous visuotactile stimulation leads to a significantly stronger rubber hand illusion than this latter condition when participants are just looking at the rubber hand without any visuotactile stimulation (e.g., Guterstam et al., 2019). We think more studies are needed before we can draw any strong conclusions from Durgin et al., (2007).

It is well-known in psychological research that SCR is an unspecific measure, and that different perceptual, emotional, and cognitive processes can influence SCR. Thus, just as in psychophysiological research in general, it is very important to have adequate control conditions and adopt a hypothesis-driven approach when using the SCR in bodily illusion research. SCR responses triggered by physical threats are ecologically valid because it probes basic emotional defense reactions triggered by bodily threats (Ehrsson et al., 2007; Graziano and Cooke, 2006). In our bodily illusion studies with threat-evoked SCR, we always use control conditions (e.g., Gentile et al., 2013; Guterstam et al., 2015) and only focus on illusion-condition specific increases in SCR triggered by the threat-stimulus compared to when identical threats are presented in the asynchronous and other control conditions; in addition, we sometimes use control stimuli like non-threatening objects (e.g., Guterstam et al., 2015; Petkova and Ehrsson, 2008). In addition, bear in mind that most naïve participants do not understand how SCR works. Thus, in our view it is very unlikely that naïve participants can voluntarily control their evoked SCR responses in a condition-specific manner to simulate physiological responses in experimental designs like the ones described above. Also note that physical threats toward the rubber hand elicit specific fMRI responses in areas related to pain anticipation and fear, such as the anterior insular cortex (Ehrsson et al., 2007; Gentile et al., 2013) and amygdala (Guterstam et al., 2015), and the stronger the illusion in the synchronous condition compared to the asynchronous (and other controls), the stronger these threat-evoked BOLD responses. This suggest that the threat-evoked SCR reflects centrally mediated emotional defense reactions that are triggered by perceived threats towards one’s own body.

Note that in the recent study by Lush and colleagues (Lush et al., 2021) no SCR recordings were conducted so the claims about possible links between cognitive expectancies and changes in SCR remain speculative. This study also suffers from the same limitations as the questionnaire study discussed above (Lush 2020) in that psychology students were asked to guess which of the two conditions – the synchronous conditions or the asynchronous conditions – they thought should produce the strongest SCR after reading and learning about the rubber hand illusion. But as said, SCR was not registered, and the rubber hand illusion was not tested. Thus, it is unclear how metacognitive guesses about the rubber hand illusion relate to condition-specific differences in threat-evoked SCR during an actual rubber hand illusion experiment in genuinely naïve participants. To the best of our knowledge, the potential influence of cognitive expectations on SCR has yet not been tested in a controlled bodily illusion experiment.

The reviewer brought up the topic of indirect measures of the rubber hand here, so we would like to make a couple of further clarifications. In rubber hand illusion studies, it is common to supplement the results from questionnaires and rating scales with more objective tests such as proprioceptive drift, the cross-modal congruence task, threat-evoked SRC, fMRI, and electrophysiology (EEG/ECoG)(see Slater and Ehrsson 2022 for a recent review). The results of these studies support the hypothesis that the rubber hand illusion is a multisensory body illusion and underscore the importance of visual and somatosensory signals. Take fMRI, for example. Numerous studies have found differences between the synchronous and asynchronous conditions in specific premotor and posterior parietal areas that are known to be involved in the integration of visual, tactile, and proprioceptive bodily signals (Brozzoli et al., 2012; Ehrsson et al., 2004; Gentile et al., 2013; Limanowski and Blankenburg, 2016). Moreover, the stronger the illusion-condition-specific differences in BOLD signals in these areas, the stronger the illusion as rated in questionnaires (illusion ratings synchronous minus asynchronous).

Extremely relevant to the present study, in a recent imaging study, which is currently under revision in a leading neuroscience journal, we used fMRI to scan 30 participants as they performed the current rubber hand illusion detection task (based on the same stepwise asynchrony manipulation) and fitted the responses to the current BCI model (Chancel et al., see attached abstract for a poster at the OHBM conference in Glasgow – June 2022). As expected, BOLD activity in the premotor and posterior parietal cortices was related to illusion detection at the level of individual participants and trials, and activity in the posterior parietal cortex reflected the Bayesian causal inference model’s predicted probability of illusion emergence based on each participant’s behavioral response-profile. These findings corroborate the current behavioral study’s findings and conclusions and suggest that the rubber hand illusion detection decisions involve activity in multisensory areas related to integration versus segregation of bodily multisensory information.

That mental imagery of one’s hand being cut may modulate SCR is unsurprising. Mental imagery can influence emotional processes and lead to changes in SCR recordings. Without knowing which study you refer to, and what control conditions were used, it is difficult for us to say much more. For example, Hägni et al., (2008) did not elicit a bodily illusion but compared passive viewing arms playing a ball game versus a mental imagery condition so the changes in SCR reported in this study could relate to different cognitive factors.

Last and not least, we disagree with the reviewer’s statement that “the simplest explanation (for the rubber hand illusion, our addition) is that participants respond to what they perceive as what is meant to be relevant in the study manipulation”. Given all the arguments we presented above and in our first response, we fail to see how this “simple” interpretation can explain our key findings while another simple explanation that can do this well is that participants simply report what they feel when they experience the illusion.

References:

Aller M, Noppeney U. 2019. To integrate or not to integrate: Temporal dynamics of hierarchical Bayesian causal inference. *PLOS Biology* 17:e3000210. doi:10.1371/journal.pbio.3000210

Blanke O, Slater M, Serino A. 2015. Behavioral, Neural, and Computational Principles of Bodily Self-Consciousness. *Neuron* 88:145–166. doi:10.1016/j.neuron.2015.09.029

Botvinick M, Cohen J. 1998. Rubber hands “feel” touch that eyes see. *Nature* 391:756. doi:10.1038/35784

Brozzoli C, Gentile G, Ehrsson HH. 2012. That’s near my hand! Parietal and premotor coding of hand-centered space contributes to localization and self-attribution of the hand. *J Neurosci* 32:14573–14582. doi:10.1523/JNEUROSCI.2660-12.2012

Chancel M, Ehrsson HH. 2020. Which hand is mine? Discriminating body ownership perception in a two-alternative forced-choice task. *Atten Percept Psychophys*. doi:10.3758/s13414-020-02107-x

Chancel M, Hasenack B, Ehrsson HH. 2021. Integration of predictions and afferent signals in body ownership. *Cognition* 212:104722. doi:10.1016/j.cognition.2021.104722

Costantini M, Robinson J, Migliorati D, Donno B, Ferri F, Northoff G. 2016. Temporal limits on rubber hand illusion reflect individuals’ temporal resolution in multisensory perception. *Cognition* 157:39–48. doi:10.1016/j.cognition.2016.08.010

Durgin FH, Evans L, Dunphy N, Klostermann S, Simmons K. 2007. Rubber Hands Feel the Touch of Light. *Psychological Science* 18:152–157. doi:10.1111/j.1467-9280.2007.01865.x

Ehrsson HH. 2020. Multisensory processes in body ownershipMultisensory Perception. Elsevier. pp. 179–200. doi:10.1016/B978-0-12-812492-5.00008-5

Ehrsson HH, Spence C, Passingham RE. 2004. That’s My Hand! Activity in Premotor Cortex Reflects Feeling of Ownership of a Limb. *Science* 305:875–877. doi:10.1126/science.1097011

Ehrsson HH, Wiech K, Weiskopf N, Dolan RJ, Passingham RE. 2007. Threatening a rubber hand that you feel is yours elicits a cortical anxiety response. *Proc Natl Acad Sci USA* 104:9828–9833. doi:10.1073/pnas.0610011104

Gentile G, Guterstam A, Brozzoli C, Ehrsson HH. 2013. Disintegration of multisensory signals from the real hand reduces default limb self-attribution: an fMRI study. *J Neurosci* 33:13350–13366. doi:10.1523/JNEUROSCI.1363-13.2013

Graziano MSA, Cooke DF. 2006. Parieto-frontal interactions, personal space, and defensive behavior. *Neuropsychologia* 44:845–859. doi:10.1016/j.neuropsychologia.2005.09.009

Guterstam A, Björnsdotter M, Gentile G, Ehrsson HH. 2015. Posterior cingulate cortex integrates the senses of self-location and body ownership. *Curr Biol* 25:1416–1425. doi:10.1016/j.cub.2015.03.059

Guterstam A, Gentile G, Ehrsson HH. 2013. The invisible hand illusion: multisensory integration leads to the embodiment of a discrete volume of empty space. *J Cogn Neurosci* 25:1078–1099. doi:10.1162/jocn_a_00393

Guterstam A, Larsson DEO, Zeberg H, Ehrsson HH. 2019. Multisensory correlations—Not tactile expectations—Determine the sense of body ownership. *PLOS ONE* 14:e0213265. doi:10.1371/journal.pone.0213265

Guterstam A, Zeberg H, Özçiftci VM, Ehrsson HH. 2016. The magnetic touch illusion: A perceptual correlate of visuo-tactile integration in peripersonal space. *Cognition* 155:44–56. doi:10.1016/j.cognition.2016.06.004

Hägni K, Eng K, Hepp-Reymond M-C, Holper L, Keisker B, Siekierka E, Kiper DC. 2008. Observing Virtual Arms that You Imagine Are Yours Increases the Galvanic Skin Response to an Unexpected Threat. *PLOS ONE* 3:e3082. doi:10.1371/journal.pone.0003082

Kalckert A, Ehrsson HH. 2014. The moving rubber hand illusion revisited: comparing movements and visuotactile stimulation to induce illusory ownership. *Conscious Cogn* 26:117–132. doi:10.1016/j.concog.2014.02.003

Kilteni K, Maselli A, Kording KP, Slater M. 2015. Over my fake body: body ownership illusions for studying the multisensory basis of own-body perception. *Front Hum Neurosci* 9:141. doi:10.3389/fnhum.2015.00141

Körding KP, Beierholm U, Ma WJ, Quartz S, Tenenbaum JB, Shams L. 2007. Causal inference in multisensory perception. *PLoS ONE* 2:e943. doi:10.1371/journal.pone.0000943

Limanowski J, Blankenburg F. 2016. Integration of Visual and Proprioceptive Limb Position Information in Human Posterior Parietal, Premotor, and Extrastriate Cortex. *J Neurosci* 36:2582–2589. doi:10.1523/JNEUROSCI.3987-15.2016

Lush P. 2020. Demand Characteristics Confound the Rubber Hand Illusion. *Collabra: Psychology* 6:22. doi:10.1525/collabra.325

Lush P, Botan V, Scott RB, Seth AK, Ward J, Dienes Z. 2020. Trait phenomenological control predicts experience of mirror synaesthesia and the rubber hand illusion. *Nat Commun* 11:4853. doi:10.1038/s41467-020-18591-6

Lush P, Seth AK. 2022. Reply to: No specific relationship between hypnotic suggestibility and the rubber hand illusion. *Nat Commun* 13:563. doi:10.1038/s41467-022-28178-y

Lush P, Seth AK, Dienes Z. n.d. Hypothesis awareness confounds asynchronous control conditions in indirect measures of the rubber hand illusion. *Royal Society Open Science* 8:210911. doi:10.1098/rsos.210911

Morgan MJ, Hole GJ, Glennerster A. 1990. Biases and sensitivities in geometrical illusions. *Vision Research*, Optics Physiology and Vision 30:1793–1810. doi:10.1016/0042-6989(90)90160-M

Parise CV, Ernst MO. 2016. Correlation detection as a general mechanism for multisensory integration. *Nat Commun* 7:11543. doi:10.1038/ncomms11543

Petkova VI, Ehrsson HH. 2008. If I Were You: Perceptual Illusion of Body Swapping. *PLOS ONE* 3:e3832. doi:10.1371/journal.pone.0003832

Reader AT, Trifonova VS, Ehrsson HH. 2021. The Relationship Between Referral of Touch and the Feeling of Ownership in the Rubber Hand Illusion. *Front Psychol* 12:629590. doi:10.3389/fpsyg.2021.629590

Rigoux L, Stephan KE, Friston KJ, Daunizeau J. 2014. Bayesian model selection for group studies — Revisited. *NeuroImage* 84:971–985. doi:10.1016/j.neuroimage.2013.08.065

Rohe T, Ehlis A-C, Noppeney U. 2019. The neural dynamics of hierarchical Bayesian causal inference in multisensory perception. *Nat Commun* 10:1–17. doi:10.1038/s41467-019-09664-2

Rohe T, Noppeney U. 2016. Distinct Computational Principles Govern Multisensory Integration in Primary Sensory and Association Cortices. *Curr Biol* 26:509–514. doi:10.1016/j.cub.2015.12.056

Rohe T, Noppeney U. 2015. Cortical Hierarchies Perform Bayesian Causal Inference in Multisensory Perception. *PLOS Biology* 13:e1002073. doi:10.1371/journal.pbio.1002073

Samad M, Chung AJ, Shams L. 2015. Perception of body ownership is driven by Bayesian sensory inference. *PLoS ONE* 10:e0117178. doi:10.1371/journal.pone.0117178

Shams L, Beierholm U. 2022. Bayesian causal inference: A unifying neuroscience theory. *Neurosci Biobehav Rev* 137:104619. doi:10.1016/j.neubiorev.2022.104619

Shams L, Beierholm UR. 2010. Causal inference in perception. *Trends Cogn Sci* 14:425–432. doi:10.1016/j.tics.2010.07.001

Shimada S, Suzuki T, Yoda N, Hayashi T. 2014. Relationship between sensitivity to visuotactile temporal discrepancy and the rubber hand illusion. *Neuroscience Research* 85:33–38. doi:10.1016/j.neures.2014.04.009

Slater M, Ehrsson HH. 2022. Multisensory Integration Dominates Hypnotisability and Expectations in the Rubber Hand Illusion. Frontiers in Human Neuroscience 16.

Tsakiris M. 2010. My body in the brain: a neurocognitive model of body-ownership. *Neuropsychologia* 48:703–712. doi:10.1016/j.neuropsychologia.2009.09.034

Tsakiris M, Haggard P. 2005. The Rubber Hand Illusion Revisited: Visuotactile Integration and Self-Attribution. *Journal of Experimental Psychology: Human Perception and Performance* 31:80–91. doi:10.1037/0096-1523.31.1.80

Weber SJ, Cook TD. 1972. Subject effects in laboratory research: An examination of subject roles, demand characteristics, and valid inference. *Psychological Bulletin* 77:273–295. doi:10.1037/h0032351

(iII) Provide means and SE's for conditions for synchrony judgment tasks.

Table has been added to the Figure3_Supplement3 (former Figure5_Supplement3).

(iv) Discuss the alternative views of how the study could be interpreted as I have indicated above.

The revised manuscript includes a discussion of the alternative views suggested by the reviewer and we make the key arguments leading to our main conclusions more explicit for the reader. We think this study’s main results are well protected against demand characteristics and suggestibility and that it is unplausible that participants can solve the current rubber hand illusion task using a domain-general reasoning strategy. We agree with the reviewer that it is best to discuss these issues in the text.

The reviewer is correct that we conducted our study to test a particular hypothesis and computational model about the rubber hand illusion and that our study was not designed to try to distinguish between different theories. We are happy to stay more focused on the conclusions that directly relate to our modelling results and hypotheses and avoid unnecessary broader theoretical speculations. So, we have removed the original sentence we had where we cited Lush et al., 2020, which was uninformative and vague.

Note, that the current manuscript is already very long, and the other reviewers have also asked us to include more technical details (*eLife* does not allow supplementary material with supplementary discussion). We also do not want to replicate previous more general discussions about demand characteristics and suggestibility in the rubber hand illusion, which has already been well covered in other publications (e.g.,Lush and Seth, 2022; Ehrsson et al., 2022; Slater and Ehrssson, 2022). Thus, we are focusing the new discussion sentences on how the issues raised by the reviewer relates to the current study’s experimental design main modelling results and key conclusions.

Old version:

“Such successful modeling of the multisensory processing driving body ownership is especially relevant as several alternative models of body ownership emphasize interoception (Azzalini et al., 2019, Park and Blanke, 2019), motor processes (Burin et al., 2015, 2017), hierarchical cognitive and perceptual models (Tsakiris et al., 2010), or high-level cognition and expectations (Lush et al., 2020). Therefore, quantitative computational studies like the present one are needed to formally compare these different theories of body ownership and advance the corresponding theoretical framework.”

New version:

“Such successful modeling of the multisensory information processing in body ownership is relevant for future computational work into bodily illusions and bodily self-awareness, for example, more extended frameworks that also include contributions of interoception (Azzalini et al., 2019, Park and Blanke, 2019), motor processes (Burin et al., 2015, 2017), pre-existing stored representations about what kind of objects that may or may not be part of one’s body (Tsakiris et al., 2010), and high-level cognition and expectations (Lush et al., 2020; Lush 2019). Quantitative computational studies like the present one are needed to formally compare these different theories of body ownership and advance the corresponding theoretical framework.”

One of the reviewers raised the question of to what extent demand characteristics and cognitive expectations could influence the current findings, citing the recent debate on this topic (Lush 2020; Lush et al., 2020; Ehrsson et al., 2022; Lush and Seth 2022; Slater and Ehrsson 2022). Although demand characteristics and trait suggestibility is likely to modulate illusion responses across all levels of asynchrony in the current study (Lush 2020; Slater and Ehrsson 2022), and may thus explain some of the differences in the overall tendency to give ‘yes’ responses across participants, such effects cannot explain the specifically shaped psychometric curves with respect to the subtle stepwise asynchrony manipulations and the widening of this curve by the noise manipulation as predicted by the causal inference model. Note that there are many features of the task design and the experimental procedures that reduce the risk of demand characteristics or changes in decision criterion: a large number of trials were presented in a fully randomized order, robots delivered the stimuli, the experimenter was blind to the noise manipulation and out-of-view of the participants, the data was analyzed and modeled in individual participants, and the BCI model’s specific predictions are “hidden” and very difficult to figure out spontaneously; note further the shortest asynchrony we used (150 ms) was so brief that most participants did not perceive it reliably as asynchronous (Costantini et al., 2016; Shimada et al., 2009), and the participants did not know how many different asynchrony levels were tested (as revealed in post-experiment interviews). Another crucial point is that no feedback was given to the participants about task performance, so they could not learn to map their responses to the different experimental manipulations. Thus, our main results reflect the effect of the causal experimental manipulations – the level of asynchrony and visual noise – on the RHI detection responses.

We also made some additions to the method section to highlight the features of the task design and the experimental procedures that reduce the risk of demand characteristics:

– Page 26, lines 790 – 792: During the experiment, the experimenter was blind to the noise level presented to the participants, and the experimenter sat out of the participants’ sight.

– Page 26, lines 785 – 787: The participants did not know how many different asynchrony levels were tested (as revealed in unformal post-experiment interviews) and that no feedback was given on their task performance

– Page 23, lines 672 – 673: Eighteen healthy participants naïve to the conditions of the study were recruited for this experiment (6 males, aged 25.2 ± 4 years, right-handed; they were recruited from outside the department, never having taken part in a bodily illusion experiment before).

Reviewer #2 (Recommendations for the authors):Using rubber hand illusion in humans, the study investigated the computational processes in self-body representation. The authors found that the subjects' behavior can be well captured by the Bayesian causal inference model, which is widely used and well described in multisensory integration. The key point in this study is that the body ownership perception was well predicted by the posterior probability of the visual and tactile signals coming from a common source. Although this notion was investigated before in humans and monkeys, the results are still novel:1. This study directly measures illusions with the alternative forced-choice report instead of objective behavioral measurements (e.g., proprioceptive drift).2. The visual sensory uncertainty was changed trial by trial to examine the contribution of sensory uncertainty in multisensory body perception. Combined with the model comparison results, these results support the superiority of a Bayesian model in predicting the emergence of the rubber hand illusion relative to the non-Bayesian model.3. The authors investigated the asynchrony task in the same experiment to compare the computational processing in the RHI task and visual-tactile synchrony detection task. They found that the computational mechanisms are shared between these tasks, while the prior of common source is different.In general, the conclusions are well supported, and the findings advanced our understanding of the computational principles of body ownership.

Thank you for taking the time to assess our manuscript and carefully review it. We are delighted to read that the reviewer thinks our conclusions are well supported and that our study advances our understanding of the computational principles of body ownership.

Main comments:1. One of the critical points in this study is the comparison between the BCI model which takes the sensory uncertainty into account and the non-Bayesian (fixed-criterion) model. Therefore, I suggest the authors show the prediction results of both the BCI and non-Bayesian model in Figure 2 and compare the key hypothesis and the prediction results.

This is an excellent suggestion on how to illustrate the key difference between the BCI and FC models. Following this suggestion, we propose to add Author response image 2 as a panel D to Figure 5 (previously Figure 2).

**Author response image 2. sa2fig2:** (D) Finally, this last plot shows simulated outcomes predicted by the Bayesian Causal Inference model (BCI in full lines and bars) and the fixed criterion model (FC in dashed lines and shredded bars). In this theoretical simulation, both models predict the same outcome distribution for one given level of sensory noise (0%), however, since the decision criterion of the BCI model is adjusted to the level of sensory uncertainty, an overall increase of the probability of emergence of the rubber hand illusion is predicted by this Bayesian model. On the contrary, the FC model, which is a non- model, FC, predicts a neglectable effect of sensory uncertainty on the overall probability of emergence of the rubber hand illusion.

2. This study has two tasks: the ownership task and the asynchrony task. As the temporal disparity is the key factor, the criteria for determining the time disparities are important. The author claim that the time disparities used in the two tasks were determined based on the pilot experiments to maintain an equivalent difficulty level between the two tasks. If I understand correctly, the subjects were asked to report whether the rubber hand felt like my hand or not in the ownership task. Thus, there are no objective criteria of right or wrong. The authors should clarify how they define the difficulty of the two tasks and how to make sure the difficulty is equal.

The reviewer rightfully pointed out that we did not use an objective external criterion of right or wrong answer in our rubber hand illusion detection task thus we should not say that the difficulty is equal in both tasks. Our choice to reduce the range of tested asynchronies in the synchrony detection task was made after collecting the pilot data presented in Supplementary File 2A. In the +/- 500 and +/- 300 ms asynchrony conditions the number of trials for which the visuo-tactile stimulation was perceived as synchronous was consistently very low or never happened (zeros) in many cases. This observation suggests that the synchrony task was too easy with such relatively longer asynchronies and that it would not produce behavioral data that would be useful for model fitting or testing the BCI model. Thus, we adjusted the asynchrony conditions in the synchrony task to make this task more challenging and more comparable to the ownership judgment task in terms of modeling. Note that we could not change the asynchronies in the ownership task to match the synchrony task because we need the longer 300 ms and 500 ms asynchronies to suppress the illusion effectively. We agree that the term “difficulty” was misleading, thus we replaced it by “sensitivity”.

Please note that the key purpose of including two tasks was that we wanted to be able to show that both follow Bayesian causal inference, which is a finding that strengthens our main conclusions about multisensory causal inference in body ownership. We can now conclude from our own data that the rubber hand illusion obeys similar causal inference principles based on sensory evidence and sensory uncertainty as more basic forms multisensory integration. So having the two tasks in the current study has several advantages. Still, one should think of the synchrony task as a control task in the classic sense of the word because it is a different task than the rubber hand illusion detection task, so there will always be differences between them. The fact that illusory hand ownership can be elicited with longer asynchronies is one such these differences, that was already pointed out in earlier work studies (Shimiada et al., 2009).

I think this is important because the time disparities in these two tasks were different, and the comparison of Psame in these tasks may be affected by the time disparities. Furthermore, the authors claimed that the ownership and visual-tactile synchrony perception had distinct multisensory processing according to the different Psame in the two tasks. Thus the authors should show further evidence to exclude the difference of Psame results from the chosen time disparities.

The reviewer raises an interesting point. The fact that the tested asynchronies are different for the two tasks can at a first glance be seen at first as a limitation. However, as explained above, the decision to have shorter asynchronies in the synchrony task was taken after careful consideration and analysis of the pilot data. As mentioned in the discussion, we cannot exclude this methodological choice might have influenced the observed differences in the causal prior in the two tasks (i.e., higher prior probability of a common cause for the smaller range of asynchrony) instead of reflecting differences in causal inference of hand ownership versus visuotactile simultaneity perception as we suggest based on theoretical considerations and the previous literature.

However, to further examine the reviewer’s concern we here report the results from an additional analysis. We applied our extension analysis to the pilot data to test the BCI model on tasks with identical asynchronies as shown in Supplementary File 2B. Note that the pilot study did not manipulate the level of sensory noise (only the 0% noise level was included). As described above, this pilot study included ten naïve participants and used the same setup and procedures as the main experiment, with the only exception being that asynchronies in the synchrony judgment task now were identical to those in the ownership judgment task (and no noise manipulation as just said).

The Author response image 3 shows the key results regarding the estimated p_same_. Note that the *same trend is observed as in the main experiment*: the estimated a priori probability for a common cause for synchrony judgment is *lower* than for body ownership judgement. However, in this pilot experiment p_same_ for body ownership and synchrony are not correlated (spearman correlation: S=150, p = 0.81, rho = 0.09), and for more than half of our pilot participants, p_same_ for body ownership reaches the extremum (psame = 1). This ceiling effect might be because the synchrony task was too “easy” when using asynchronies of 300 ms and 500 ms; it lacked challenging stimulation conditions required to assess the participants’ perception as a finely gradual function, and thus these data was unsuitable for our modeling purposes. This observation further convinced us that we needed to make the synchrony judgment task more difficult by reducing the longer asynchronies to obtain high-quality behavioral data that would allow us to test the subtle effects of sensory noise, compare different models, and compare with the ownership judgment task in a meaningful way.

**Author response image 3. sa2fig3:** Correlation between the prior probability of a common cause psame estimated for the ownership and synchrony tasks in the extension analysis in the pilot study (left) and the main study (right). The psame estimate is significantly lower for the synchrony task than for the ownership task. The solid line represents the linear regression between the two estimates, and the dashed line represents the identity function (x=f(x)).

3. Related to the question above, the authors found that the same BCI model can reasonably predict the behavioral results in these two tasks with the same parameters (or only different Psame). They claimed that these two tasks shared similar results in multisensory causal inference. While in the following, they argued that there was a distinct multisensory perception in these two tasks because the Psame were different. If these tasks shared the same BCI computational approaches and used the posterior probability of common source to determine ownership and synchrony judgment, what is the difference between them?

We thank the reviewer for this opportunity to clarify our claim. We think it makes good sense that the degree of asynchrony of the visual and tactile stimuli drives both the synchrony judgments (for obvious reasons) and the rubber hand illusion judgments (since the temporal congruence of the visuotactile correlations is a key factor factor that drives the emergence of the illusion). Both follow multisensory causal inference, which we can demonstrate with our own data, and which support our main claim about multisensory body ownership. But of course, the tasks are different in that the perception of simultaneous brief visuotactile events and the perception of the rubber hand as one’s own are different percepts (and the latter also involve other sources of sensory evidence that we kept constant in the current study such as visuo-proprioceptive spatial congruence, the number of correlated stimuli, the prior state of body representation), so it is expected that p_same_ should be different we think.

What our modeling analysis shows is that both tasks use the same probabilistic principles for sensory decision making: the same strategy to combine sensory information and take into account sensory uncertainty, but the difference lays in what type of prior information that is combined and what the causal inference is about. The sensory input signals to be considered are the same, since the visuotactile stimulations are the same, as well as the visual impressions of the rubber hand and the proprioceptive feedback from the hidden hand. However, the prior information used in the causal inference process is different. Hence, we conclude that the two tasks use similar computational principles but call upon different prior representations in line with that the tasks probes two distinct types of perception. The prior is an umbrella term that can reflect different types of information. Nevertheless, the finding of different priors in the two tasks is in line with our conclusion of differences in computational processing between the two tasks. However, we agree that the comparison of the p_same_ across the two tasks is the least conclusive finding in the current study, and that is why we also discuss this finding critically in the Discussion section. Note, however, that this finding is independent of our main finding that hand-ownership feeling in the rubber hand illusion follows Bayesian causal inference based on sensory information and sensory uncertainty.

4. The extension analysis showed that the Psame values from the two tasks were correlated across subjects. This is very interesting. However, since the uncertainty of timing perception (the sensory uncertainty of perceived time of the tactile stimuli on real hand and fake hand) was taken into account to estimate the posterior probability of common source in this study, the variance across subjects in ownership and synchrony task can only be interpreted by the Psame. In fact, the timing perception was considered as a Gaussian distribution in a modulated version of the BCI model for the agency (temporal binding) (R. Legaspi, 2019) and ownership (Samad, 2015). It will be more persuasive if the authors exclude the possibility that the individual difference of timing uncertainty cannot explain the variance across subjects.

We are happy to read that the reviewer found this result interesting. Let us clarify our approach: in our study, the noise parameters (i.e., the timing uncertainty) are fitted individually, just like the p_same_ parameters, therefore they reflect parts of the variability across participants. Thus, this does not go against our interpretation of the results: the observed differences between the two tasks reflect the use of different priors between the two types perception and not an individual difference in the measured timing uncertainty.

5. Please include the results of single-subject behavior in the asynchrony task. It is helpful to compare the behavioral pattern between these two tasks. The authors compared the Psame between ownership and asynchrony tasks. Still, they did not directly compare the behavioral results (e.g., the reported proportion of elicited rubber hand illusions and the reported proportion of perceived synchrony).

We agree with the reviewer that several approaches could be used to compare ownership and synchrony detection data. Focusing on a comparison at the computational level as we do means that we handle a “summary” of the datasets, obtained thanks to careful and statistically robust fitting methods. By doing so we avoid limitations related to multiple comparisons and random effects from the participant samples. These limitations could be controlled for in a more traditional approach based on classic inferential statistics, however not without impacting the statistical power of our analyses. Critically, in the present study we use a model-based approach to learn more about the computational principles of body ownership, so we want to use this approach throughout the manuscript. Adding lot of extra comparisons based on descriptive statistics might be distracting to the reader (and we already have lots of results in the current manuscript). Moreover, the two tasks have many differences that make direct statistical comparisons of the sort as the reviewer is recommending somewhat difficult to interpret. Thus, we prefer to focus on a model-based comparison of the two tasks, using a computational approach to pinpoint difference in processes between the tasks more than on specific observations. Additionally, we would like to emphasize that the main findings of the study are the fit of the BCI model to the body ownership data and the modulation of body ownership by the sensory uncertainty. The comparison with the synchrony judgment task, and the finding of the different p_same_ are interesting and novel but can almost be seen as a secondary “bonus finding” of the study; but a finding of interest to researchers in the rubber hand illusion community, we think. Nonetheless, if the reviewer thinks it would be insightful, we could add the plot of the individual responses and corresponding fit presented within Author response image 4 as the Figure3_Supplement4.

**Author response image 4. sa2fig4:** Individual data and BCI model fit. The figure display one plot per participant, the “yes [the rubber hand felt like my own hand]" answers as a function of visuo-tactile asynchrony (dots) and corresponding BCI model fit (curves) are plotted. As in the main text, dark blue, light blue, and cyan correspond to the 0%, 30%, and 50% noise levels, respectively.

6. The analysis of model fitting seems to lack some details. If I understood it correctly, the authors repeated the fitting procedure 100 times (line 502), then averaged all repeats as the final results of each parameter? It is reported that "the same set of estimated parameters at least 31 times for all participants and models". What does this sentence mean? Can the authors show more details about the repeats of the model result?

As the reviewer mentioned, we repeat the fitting procedure 100 times, with different initial values for the parameters; however, we did not average the results but selected the parameter set that maximized the most the log-likelihood of the model. This procedure is a way to find the “best estimation” of the parameters and avoid local minimums (sets of estimated parameters that do not lead to the true minimum negative log-likelihood for a given participant and a given model).

Finding the same set of estimated parameters several times is an argument in favor of the robustness of our fitting procedure: the optimization algorithm leads to the same “best set of estimated parameters” starting from different initial values for the parameters.

We added the following sentence to the parameter estimation section to clarify this procedure:

“The best estimate from either of these two procedures was kept, i.e., the set of estimated parameters that corresponded to the maximal log-likelihood for the models”

7. In Figure 3A, was there an interaction effect between the time disparity and visual noise level?

The interaction between time disparity and visual noise was not significant (F(12, 280) = 1.6, p = 0.097). Note that Figure 3 is now Figure 1.

8. Line 624, the model comparison results suggested that the subjects have the same standard deviation as the true stimulus distribution. I encourage the authors to directly compare the BCI* model predicted uncertainty (σ s) to the true stimulus uncertainty, which will make this conclusion more convincing.

We respectfully disagree that a comparison based on one parameter estimation would be a stronger argument to select the best model between BCI* and BCI than a confidence interval and bootstrap analysis using AIC and BIC since we are dealing here with a parsimony question. Our analysis shows that the BCI* would present a risk of an overfit. Since the estimated value of one parameter is not independent of the evaluation of the other parameters, an analysis based on parameter estimation in the case of an overfitted model does not make sense to us. Hence, we find it more relevant to use a model comparison method that is based on the overall fit of the models and not just one parameter value.

9. How did the authors fit the BCI model to the combined dataset from both ownership and synchrony tasks? What is the cost function when fitting the combined dataset?

We have added the information about the extension analysis the reviewer is asking about to the appendix:

When fitting the BCI to the combined dataset, we used the same expression of the log-likelihood to be maximized as when fitting only one dataset:log⁡L(θ)=∑i, j [n1ijlog⁡p(C^=1|sj, θ)+n0ijlog⁡(1−p(C^=1|sj, θ))]

where n1ij and n0ij are the observed data, i.e., the numbers of times the participant reported “yes” and “no”, respectively, in the (i, j)th condition.

When one dataset was considered, there were 21 conditions (7 asynchronies and 3 levels of visual noise). When the datasets were combined, the conditions were doubled since all experimental conditions were tested in the body ownership task and in the synchrony detection task. Most parameters were fitted to all the data, the only exception being when a different value of psame was estimated for each type of judgment. In this case, psame is affected by only half of the data. As a result, the parameters to be fitted (θ) depended on the type of extension analysis:

θ=[psame,σ0,σ30, σ50, λ], when both tasks shared all parameters.

θ=[psame,ownership,psame, synchrony, σ0,σ30, σ50, λ], when a different value of *p*_same_ was estimated for each type of judgment

10. As shown in the supplementary figures, the variations of ownership between the three visual noise levels varied widely among subjects and the predicted visual sensory sigmas (Appendix 1 – Table 2). The ownership in the three visual noise levels correlated with the individual difference of visual uncertainty?

We are not sure we fully understand the reviewer's concern here. Indeed, the Appendix 1 – Table 2 presents the results of the parameter recovery analysis; therefore, it is based on simulation and not actual participants. The variability is therefore greater than in the actual dataset. Moreover, in our study, because all the parameters are fitted individually, the variations in all parameters explain the variations among individuals, and all contribute to the goodness of the final fit.

11. The statements of the supplementary figures are confusing. For example, it is hard to determine which one is the "Supplementary File 1. A" in line 249?

We apologize for this lack of clarity. We initially created a supplementary material document containing supplementary figures, files, and notes. However, the submission format to *eLife* does not allow supplementary notes. In accepted manuscripts, the supplementary figures would be directly linked to the main figures they supplement (e.g., Figure 1_Supplement1) while the supplementary files are meant to be independent, which should help clarify the referencing of the supplement material. For more clarity, we transform the supplementary file 2 into appendix 1 – Section 4 and 5 and supplementary file 1 into figure 4 – supplement 1 and 2.

12. Line 1040, it is hard to follow how to arrive at this equation from the previous formulas. Please give some more details and explanations.

Good point. We have added the following equations to make the different steps leading to the expression of the decision criterion easier to follow.xtrial22(1σ2−1σ2+σs2)< log(psame1−psame)+12log⁡(σ2+σs2σ2)xtrial2< σ2(σ2+σs2)σs2 (2log(psame1−psame)+log(σ2+σs2σ2))

Reviewer #3 (Recommendations for the authors):This study investigated the computational mechanisms underlying the rubber hand illusion. Combining a detection-like task with the rubber hand illusion paradigm and Bayesian modelling, the authors show that human behaviour regarding body ownership can be best explained by a model based on Bayesian causal inference which takes into account the trial-by-trial fluctuations in sensory evidence and adjusts its predictions accordingly. This is in contrast with previous models which use a fixed criterion and do not take trial-by-trial fluctuations in sensory evidence into account.The main goal of the study was to test whether body ownership is governed by a probabilistic process based on Bayesian causal inference (BCI) of a common cause. The secondary aim was to compare the body ownership task with a more traditional multisensory synchrony judgement task within the same probabilistic framework.The objective and main question of the study is timely and interesting. The authors developed a new version of the rubber hand illusion task in which participants reported their perceived body ownership over the rubber hand on each trial. With the manipulation of visual uncertainty through augmented reality glasses they were able to assess whether trial-by-trial fluctuation in sensory uncertainty affects body ownership – a key prediction of the BCI model.This behavioural paradigm opens up the intriguing possibility of testing the BCI model for body ownership at a neural level with fMRI or EEG (e.g., as in Rohe and Noppeney (2015, 2016) and Aller and Noppeney (2019)).I was impressed by the methodological rigour, modelling and statistical methods of the paper. I was especially glad to see the modelling code validated by parameter recovery. This greatly increases one's confidence that good coding practices were followed. It would be even more reassuring if the analysis code were made publicly available.The data and analyses presented in the paper support the key claims. The results represent a relevant contribution to our understanding of the computational mechanisms of body ownership. The results are adequately discussed in light of a broader body of literature. Figures are well designed and informative.

Thank you for this positive evaluation of our work. We are especially pleased to read that the reviewer thinks that our modeling approach is sound and that our data and analyses support the key claims. We were also delighted to see that you highlighted that the current approach offers new perspectives, including the possibility of investigating the neural basis of causal inference of body ownership. Indeed, this is something we are currently pursuing in a new fMRI study (see the attached abstract, the corresponding manuscript is currently under review; see also our responses to reviewer 1).

Main points:1. Line 298: It is not clear if all 5 locations were stimulated in each 12 s stimulation phase or they were changed only between stimulation phases. Please clarify.

The clarification has been added to the manuscript. “All five locations were stimulated at least once in each 12 s trial and the order of stimulation sites randomly varied from trial to trial.” Indeed, a 12s stimulation meant six touches, one on every location and one location was stimulated a second time. The order of stimulation and the location stimulated twice were randomly chosen from one trial to the other (see below an example of the stimulation sequence for one participant).

Example of a stimulation location sequence, the touch location being numbered from 1 to 5:

Trial 1: 4, 2, 3, 1, 2, 5 – Trial 2: 3, 5, 4, 2, 5, 1 – Trial 3: 2, 4, 3, 3, 5, 1 – Trial 4: 5, 2, 1, 3, 4, 2 – ….

2. Line 331: "The 7 levels of asynchrony appeared with equal frequencies in pseudorandom order". I assume this was also true to the noise conditions, i.e., they also appeared in pseudorandom order with equal frequencies and not e.g., blocked. Could you please make this explicit here?

The reviewer is correct; in the new version of the manuscript, we have explicitly explained the pseudo randomization of the noise conditions:

“The three levels of noise also appeared with equal frequencies in pseudorandom order.”

3. Line 348: Was the pilot study based on an independent sample of participants from the main experiment? Please also include standard demographics data (mean+/-SD age, sex) from the pilot study.

The pilot participants were different from the ones participating to the main experiment. We added the demographics of the pilot sample to the manuscript:

“(3 males, aged 27.0 ± 4 years, different than the main experiment sample)”.

4. Line 406: From the standpoint of understanding the inference process at a high level, the crucial step of how prior probabilities are combined with sensory evidence to compute posterior probabilities is missing from the equations. More precisely it is not exactly missing, but it is buried inside the definition of K (line 416) if I understand correctly. I think it would make it easier for non-experts to follow the thought process if Equation 5 from Supplementary material would be included here.

We thank the reviewer for pointing out this lack of clarity. We tried to make our procedure as transparent as possible without penalizing the readability of our manuscript. Following the reviewer’s suggestion, we added the equation showing how prior probability and sensory evidence are combined (Equation 5 in the appendix) when presenting the inference step of the model in the main manuscript:

“… This equation can be written as a sum of the log prior ratio and the log-likelihood ratio:d=log(psame1−psame)+log(p(xtrial|C=1)p(xtrial|C=2))

5. Line 511: There are different formulations of BIC, could you please state explicitly the formula you used to compute it? Please also state the formula for AIC.

The BIC and AIC formula, originally reported only in the appendix, are now added to the main text as well.

“We calculated AIC and BIC values for each model and participant according to the following equations:AIC=2npar−2log⁡L∗BIC=ntriallog⁡npar−2log⁡L∗

where L∗ is the maximized value of the likelihood, npar the number of free parameters, and ntrial the number of trials.”

6. Line 512: "Badness of fit": Interesting choice of word, I completely understand why it is chosen here, however perhaps I would use "goodness of fit" instead to avoid confusion and for the sake of consistency with the rest of the paper.

We followed the reviewer’s suggestion and used the term “goodness of fit” to avoid any confusion.

7. Figure 4: I think the title could be improved here, e.g., "Model predictions of behavioural results for body ownership" or something similar. Details in the current title (mean +/- sem etc.) could go in the figure legend text.I am a bit confused about why the shaded overlays from the model fits are shaped as oblique polygons? This depiction hints that there is a continuous increase in the proportion of "yes" answers in the neighbourhood of each noise level. Aren't these model predictions based on a single noise level value?The mean model predictions are not indicated in the figure only the +/- SEM ranges marked by the shaded areas.

We followed the reviewer’s recommendation to improve the figure’s title and renamed Figure 4, now Figure 2: “Observed and predicted detection responses for body ownership in the rubber hand illusion”. Moreover, we thank the reviewer for pointing out a mistake in the code used to plot the subplot A and C, where continuous predictions were used instead of being pooled by noise level. The figure has been edited accordingly. Finally, we created a Figure2_Supplement5 that displays the predicted proportion of “yes” responses (mean+/- SEM) by our main models.

Line 261: Given that participants' right hand was used consistently, I am wondering if it makes any difference if the dominant or non-dominant hand is used to elicit the rubber hand illusion? If this is a potential source of variability it would be useful to include information on the handedness of the participants in the methods section.

To the best of our our knowledge, no significant effect of handedness has been observed in the rubber hand illusion (Smit M, Kooistra DI, van der Ham IJM, Dijkerman HC (2017) Laterality and body ownership: Effect of handedness on experience of the rubber hand illusion. Laterality 22:703–724.). The illusion seems to work equally well on the right hand and the left hand, and in the literature, one finds that many studies use the left hand (as Botvinick and Cohen 1998) and that many use the right hand (which we typically do because most neuroimaging studies has studied the right hand, so we know more about the central representation of the right upper limb). However, we included only right-handed participants to eliminate any putative effect of handedness in the current data. This clarification has been added to the method section.

Line 314: Please use either stimulation "phase" or "period" consistently across the manuscript

We thank the reviewer for pointing out this lack of consistency. We now use ‘stimulation period’ instead of phase.

Line 422: The study by Körding et al., (2007) is about audiovisual spatial localization, not synchrony judgment as is referenced currently. Please correct.

We corrected our mistake.

Line 529: Perhaps the better paper to cite here would be Rigoux et al., (2014) as this is an improvement over Stephan et al., (2009) and also this is the paper that introduces the protected exceedance probability which is used here.

We agree with the reviewer, the reference to Stephan et al.’s study was not the most relevant. We are now citing Rigoux et al., (2014) instead.

Figure 3 and potentially elsewhere: Please overlay individual data points on bar graphs. There is plenty of space to include these on bar graphs and would provide valuable additional information on the distribution of data.

Individual data points have been added to the bar plot on figure 1 (former Figure 3). The bar plots on figure 2 (former figure 4) display the same data with the addition of the model predictions. To avoid overloading the latter figure, we did not add the individual data points (that would be the same as in figure 1.B) but that can be done if needed. Individual data points have also been added to Figure3_Supplement3.

Figure 5A: Please consider increasing the size of markers and their labels for better visibility (i.e., similar size as in panels B and C).

Markers’ size has been increased.

Line 608,611, 639-640, and potentially elsewhere: Please indicate what values are stated in the form XX +/- YY. I assume they represent mean +/- SEM, but this must be indicated consistently throughout the manuscript.

The reviewer is correct; for consistency, we always used mean +/- SEM; this is now indicated across the Results section.

*[Editors’ note: further revisions were suggested prior to acceptance, as described below.]*

In particular, there was a consensus that the potential contribution of demand characteristics should be discussed, rather than dismissed. We also ask that you discuss the potential utility of a more simple model (signal detection theory). Please see more details below.

Thank you for your feedback and the time you spent on our manuscript. We now discuss the potential contribution of demand characteristics without dismissing the concern. We also explicitly mention the signal detection theory in the text.

(1) The authors added a paragraph detailing why the minimised (in their opinion) the contributions of demand characteristics. They argue that the theory that subjects respond to demand characteristics "cannot explain the specifically shaped psychometric curves with respect to the subtle stepwise asynchrony manipulations and the widening of this curve by the noise manipulation as predicted by the causal inference model." We were not fully convinced by this argument and wonder why you would want to categorically rule this possibility out.Reviewer 1 wrote back: This claim is false. The authors should spell out the alternative theory in terms of subjects using general purpose Bayesian inference to infer what is required. Subjects do not need to know how general purpose inference works; they just need to use it. Does the fact that information was delivered trial by trial over many trials rule out general purpose Bayesian inference? On the contrary, it supports it. Bayesian updating is particularly suited to trial by trial situations (e.g. see general learning and reasoning models by Cristoph Mathys), as illustrated by the author's own model. The authors' model could be a general purpose model of Bayesian inference, shown by its applicability to the asynchrony judgment task. The fact that the number of asynchrony levels may not have been noticed by subjects is likewise irrelevant; subjects do not need to know this. Indeed, the authors point out that the asynchrony judgment task was easier than the RHI task, so that the authors needed to use a smaller asynchrony range for this task than the RHI one. That is, subjects' ability to discriminate asynchronies is shown by the authors' data to be more than enough to allow task-appropriate RHI judgments. (Even if successive differences between asynchrony levels were below a jnd, one would still of course still get a well formed psychophysical function over an appropriate span of asynchronies; so subjects could still use asynchrony in lawful ways as part of general reasoning, i.e. apart from any specific self module.)(In their cover letter the authors bring up other points that are covered by the fact that subjects do not need to know how e.g. inference and imagination work in order to use them. It is well established that response to imaginative suggestions involves the neurophysiology underlying the corresponding subjective experience (e.g https://psyarxiv.com/4zw6g/ for review); imagining a state of affairs will create appropriate fMRI or SCR responses without the subject knowing how either fMRI or SCRs work.) In sum, none of the arguments raised by the authors actually count against the alternative theory of general inference in response to demand characteristics (followed by imaginative absorption), which remains a simple alternative explanation.Following a discussion we ask that to address the Reviewer's perspective, you acknowledge the possibility that demand characteristics are contributing to participants performance.

In the new version of the manuscript, we now acknowledge that demand characteristics can influence the participants' judgments; moreover, we point out how within our computational framework (influencing p_same_).

However, we prefer not to speculate about unknown cognitive factors that may contribute to demand characteristics in the current study. We find reviewer 1’s proposal about (unconscious) Bayesian general inference in response to demand characteristics followed by imaginative absorption very speculative and improbable. The participants received no experimental feedback, and we are not aware of any published study that have reported (unconscious) Bayesian general inference in a paradigm like ours; the articles the reviewer referred to in law and medicine are very different. Note further that in control experiments with similar asynchrony manipulation, we find little evidence of general cognitive inference https://psyarxiv.com/uw8gh. Regarding “imaginative absorption” and “hypnotic hallucinations” we are not aware of any previous study that have reported perceptual or imaging results showing such effects to occur in normal, not highly hypnotizable subjects, and in the absence of active hypnotic suggestions (the classic studies cited in the Dienes and Lush preprint, e.g., Kosslyn et al., 2000 *Am J Psychiatry*, used highly hypnotizable subjects and hypnotic induction). Also, as we pointed out in our previous response, in Lush et al., 2021, there was *no* relationship between hypnotic suggestibility and the subjective strength of the rubber hand illusion when considering the difference between synchronous and asynchronous conditions; and if no changes were seen in such a simple design we do not understand how hypnotic hallucinations or “imaginative absorption” could explain the current detailed psychometric curves as a function of asynchrony and noise level. Indeed, strong claims like the ones the reviewer is making would require further formalization of this theory, especially quantifiable – hence testable – predictions about the expected contribution of demand characteristics’ influence on the participants’ behavior when confronted to variable degrees of asynchronous stimulations. Because of this lack of specific predictions and based on the reasons we outlined in the previous rebuttal letter, we think that considering demand characteristics and associated cognitive factors as a major explanation for the current major findings is very speculative. But as said, we now acknowledge the possibility of demand characteristics in the text, have taken the reviewer’s perspective into account, and added the following to the discussion (pages 20, lines 584-588):

“While it seems plausible that *p*_same_ reflects the real-world prior probability of a common cause of the visual and somatosensory signals, it could also be influenced by experimental properties of the task, demand characteristics (participants forming beliefs based on cues present in a testing situation, Weber et al., 1972; Corneille and Lush, 2022, Slater and Ehrsson, 2022), and other cognitive biases.”

(2) "How the a priori probabilities of a common cause under different perceptive contexts are formed remains an open question."Reviewer 1: One plausible possibility is that in a task emphasizing discrimination (asynchrony task) vs emphasizing integration (RHI) subjects infer different criteria are appropriate; namely in the former, one says "same" less readily.

We agree that we cannot exclude that experimental properties of the tasks may influence the responses. Thus, in response to the reviewers’ comment, we edited the following sentence in the paragraph discussing what factors may explain the observed difference in p_same_ between the two tasks, (pages 20, lines 582-586):

“While it seems plausible that *p*_same_ reflects the real-world prior probability of a common cause of the visual and somatosensory signals, it could also be influenced by experimental properties of the task, demand characteristics (participants forming beliefs based on cues present in a testing situation, Weber et al., 1972; Corneille and Lush, 2022, Slater and Ehrsson, 2022), and other cognitive biases.”

and(3) "temporally correlated visuotactile signals are a key driving factor behind the emergence of the rubber hand illusion"Reviewer 1: On the theory that the RH effect is produced by whatever manipulation appears relevant to subjects, of course in this study, where asynchrony was clearly manipulated, asynchrony would come up as relevant. So the question is, are there studies where visual-tactile asynchrony is not manipulated, but something else is, so subjects become responsive to something else? And the answer is yes. Guterstam et al., obtained a clear RH ownership effect and proprioceptive drift for brushes stroking the air i.e. not touching the rubber hand Durgin et al., obtained a RH effect with laser pointers, i.e. no touch involved either. The authors may think the latter effect will not replicate; but potentially challenging results still need to be cited.Here we do not ask that you provide an extensive literature review. Instead, we simply ask you that you acknowledge in the discussion that task differences might influence participants performance (similar to our request above).

Following the Reviewer and Editors suggestion, we rephrased the part of the sentence under discussion to emphasize that other types of factors can influence the RHI and body ownership judgments. Together with the other changes we have implemented in the discussion (see above), we think that the role of visuotactile temporal congruence has now been better contextualized and cognitive factors acknowledged. We added the following sentence (page 16, lines 458-462):

“Even though the present study focuses on temporal visuotactile congruence, spatial congruence (Fang et al., 2019; Samad et al., 2015) and other types of multisensory congruences (e.g., Ehrsson et al., 2005; Tsakiris et al., 2010; Ide 2013; Crucianelli and Ehrsson, 2022) would naturally fit within the same computational framework (Körding et al., 2007, Sato et al., 2007).”

(4) On noise effects.Reviewer 1: If visual noise increases until 100% of pixels are turned white, the ratio of likelihoods for C=1 vs C=2 must go to 1 (as there is no evidence for degree of asynchrony) so the probability of saying "yes" goes to p_same, no matter the actual asynchrony (which by assumption cannot be detected at all in this case). p-same is estimated as.8 in the RHI condition. Yet as noise increases, p(yes) actually increases higher than 0.8 in the -150 to +150 asynchrony range (Figure 2). Could an explanation be given of why noise increases p(yes) other than the apparent explanation I just gave (that p(yes) moves towards p(same) as the relative evidence for the different causal processes reduces)?The deeper issue is that it seems as visual noise increases, the probability that subjects say the rubber hand is their own increases and becomes less sensitive to asynchrony. In the limit it means if one had very little visual information, one just knew a rubber hand had been placed near you on the table, you would be very likely to say it feels like your own hand (maybe around the level of p-same), just because you felt some stroking on your own hand. But if the reported feeling of the rubber hand were the output of a special self processing system, the prior probability of a rubber hand slapped down on the table being self must be close to 0; and it must remain close to zero even if you felt some stroking of your hand and saw visual noise. But if the tendency to say "it felt like my own hand" was an experience constructed by realizing the paradigm called for this, then a high baseline probability of saying a rubber hand is self could well be high – even in the presence of a lot of visual noise.Please consider that this effect may bear on the two different explanations.

We first would like to point out a misunderstanding here regarding how the causal scenario is selected: the observer is not sampling from the distribution of p(C=1|s)/p(C=2|s) but maximizing it; that’s why p(yes)>p_same_ when the noise increases. In a hypothetical situation p_same_ is slightly above.5 and visual noise is very high, the participant would indeed rely mostly on prior and this would lead to a majority of “yes”.

Then, we are concerned that the theoretical case presented here, with a 100% visual noise, participants would not see anything, it would be like a unisensory stimulation condition. Recall that we ask our participants “did the hand you saw felt like it was your own hand?”. If the signal is undiscernible from the noise, participants won’t see a hand, and therefore there’s no measurement of visuotactile asynchrony, and thus our model is not applicable. Moreover, in this “extreme case scenario” mentioned by the reviewer, the participant is aware of a hand being put in front of them while observing an extremely noisy visual scene, and the reviewer reasons that this would lead the participant to report having the illusion. We argue that having your hand stroked while seeing a hand near yours not being touch could be assimilated to a strong incongruence from a visuotactile perspective (asynchrony > 12 seconds) which would still lead to a “no” answer, despite a noisy visual signal, as long as the participant is able to measure a visual signal (i.e., not noise only).

Finally, we would like to remind the reviewer that our data are much richer than merely a baseline probability of reporting the illusion. Namely, we measure and model detailed psychometric curves as a function of asynchrony and noise level. Thus, we do not believe that demand characteristics would provide a complete alternative explanation. Nevertheless, we have acknowledged that p_same_ could be influenced by demand characteristic in the discussion (see our response to the first point).

(5) The authors reject an STD model in the cover letter on the grounds subjects would not know where to place their criterion on a trial by trial basis taking into account sensory uncertainty.Reviewer 1: Why could not subjects attempt to roughly keep p(yes) the same across uncertainties? If the authors say this is asking subjects to keep track of too much, note in the Bayesian model the subjects need an estimate of a variance for each uncertainty to work out the corresponding K. That seems to be asking for even more from subjects. The authors should acknowledge the possibility of a simple STD model and what if anything hangs on using these different modelling frameworks.We feel that a brief mention of this possibility will benefit the community when considering how to leverage your interesting work in future studies.

The subjects do not seem to attempt to roughly keep p(yes) the same across uncertainties since the number of yes answers increases significantly across our visual noise conditions as mentioned in the Results section (page 8: “regardless of asynchrony, the participants perceived the illusion more often when the level of visual noise increased (F(2, 28) = 22.35, p <.001; Holmes’ post hoc test: noise level 0 versus noise level 30: p = .018, davg = 0.4; noise level 30 versus noise level 50: p = .005, davg = 0.5; noise level 0 versus noise level 50: p <.001, davg = 1, Figure 1B)”). Nonetheless, the risk of a learned threshold for each uncertainty is worth considering. We believe this is unlikely because we used multiple interleaved levels of noise while withholding any form of experimental feedback. We added this information to our discussion (page 18, lines 513 – 518).

“While we have argued that people take into account trial-to-trial uncertainty when making their body ownership and synchrony judgments, it is also possible that they learn a criterion at each noise level (Ma and Jazayeri, 2014), as one might predict in standard signal detection theory. We believe this is unlikely because we used multiple interleaved levels of noise while withholding any form of experimental feedback.”

(6) A few proofing notes that have been picked up by Reviewer 3 (these are not comprehensive, so please read over the manuscript again more carefully):1. Main points 1 and 3: The changes in response to these points as indicated in the response to reviewers are not exactly incorporated in the main manuscript file. Could you please correct?

Correction done.

2. Main point 4: in the main manuscript file there is an unnecessary '#' symbol at the end of the equation, please remove.

Correction done.

3. Main point 7: the title for figure 2 in the updated manuscript does not match the title indicated in the response to reviewers. I think the latter would be a better choice.

Correction done.

4. Supplements for figures 2, 3, 4: It seems that after re-numbering these figures, the figure legends for their supplement figures have not been updated and they still show the original numbering. Could you please update?

Correction done.